# Fault-mediated magma propagation and triggered seismicity revealed by the 2022 São Jorge Azores unrest

Stephen P. Hicks [1,17] ✉, Pablo J. Gonzalez[2,17], Anthony Lomax[3], Ana M. G. Ferreira [1], Ricardo S. Ramalho [4,5], Neil C. Mitchell [6], Graça Silveira [5,7], Nuno Afonso Dias[5,7], João Fontiela [8], Rui Fernandes[9], Susana Custódio [5], Maria Tsekhmistrenko[1], Virgilio Mendes[5], Adriano Pimentel [10,11], Rita Silva [10,11], Gonçalo Prates [12,13], William Sturgeon[1], Augustin Marignier[1,16], Fernando Carrilho[14], Rui Marques [10,11], Miguel Miranda[5,15] & Arturo M. Garcia[10,11]

Understanding failed volcanic eruptions is key to mapping magma plumbing and forecasting hazards. Faults and fractures guide magma, but their mechanisms remain unclear due to the lack of precise earthquake locations and limited 3-D fault mapping in volcanic regions. The triple-junction setting of the Azores Archipelago, where volcanic systems and seismogenic faults coexist, offers a natural laboratory to study fault–magma interactions. We analysed ~18,000 earthquakes relocated to high precision using onshore and ocean-bottom seismometers, combined with geodetic data and seismic autocorrelation imaging, during a failed 2022 eruption on São Jorge Island. A magmatic dike ascended rapidly and mostly aseismically from the upper mantle, intruding a crustal fault before stalling ~1,600 m below the surface. Seismicity indicates that magma branching and lateral fluid escape along the fault triggered an intense, months-long swarm with rotated focal mechanisms. This study demonstrates the dual role of faults in facilitating and arresting magma ascent.

Magma that ascends to the Earth's surface causes volcanic eruptions, but it can also stall at various depths, including the shallow crust, leading to failed eruptions[1]. Understanding the mechanisms of magmatic ascent is essential for interpreting volcanic unrest and crustal formation. The final fate of magma depends on several factors, like its pressure relative to the crustal rock stress state[2], fracture toughness at the dike tip[3], and internal magma dynamics (e.g. density, resupply of magma, degassing/devolatilisation[4] and crystallisation[1]).

Pre-existing faults and fractures play a crucial but underexplored role in magma dynamics. Faults can capture and guide melt[5], providing

[1]Department of Earth Sciences, University College London, London, United Kingdom. [2]Estación Volcanológica de Canarias, Department of Life and Earth Sciences, Consejo Superior de Investigaciones Científicas (IPNA-CSIC), La Laguna, Spain. [3]ALomax Scientific, Mouans Sartoux, France. [4]School of Earth and Environmental Sciences, Cardiff University, Cardiff, United Kingdom. [5]Instituto Dom Luiz (IDL), Faculdade de Ciências, Universidade de Lisboa, Lisboa, Portugal. [6]Department of Earth and Environmental Sciences, University of Manchester, Manchester, United Kingdom. [7]Instituto Superior de Engenharia de Lisboa, Instituto Politécnico de Lisboa, Rua Conselheiro Emídio Navarro 1, Lisboa, Portugal. [8]Institute of Earth Sciences, University of Évora, Évora, Portugal. [9]Instituto Dom Luiz, University of Beira Interior, Covilhã, Portugal. [10]Instituto de Investigação em Vulcanologia e Avaliação de Riscos (IVAR), Universidade dos Açores, Azores, Portugal. [11]Centro de Informação e Vigilância Sismovulcânica dos Açores (CIVISA), Azores, Portugal. [12]Instituto Superior de Engenharia, University of Algarve, Faro, Portugal. [13]Centro de Estudos Geográficos, IGOT, University of Lisbon, Lisbon, Portugal. [14]Divisão de Geofísica, Instituto Português do Mar e da Atmosfera, Lisbon, Portugal. [15]AIR Centre, Azores, Portugal. [16]Present address: Department of Earth Sciences, University of Oxford, Oxford, United Kingdom. [17]These authors contributed equally: Stephen P. Hicks, Pablo J. Gonzalez. ✉e-mail: stephen.hicks@ucl.ac.uk

energetically favourable pathways[6–8] for it to reach the surface[7,9]. Alternatively, magma may stall due to fluid circulation[10], or if the fault is misoriented for opening[9,11]. While fault architecture has been shown to influence fluid flow and seismogenesis[12], the mechanical link between faults and magmatism remains less well understood[13–15]. Spatio-temporal patterns of volcano-tectonic (VT) seismicity can reveal magma pathways[16,17]. However, a key challenge in disentangling the role of faults and fractures proximal to magma movement is obtaining sufficiently accurate and precise earthquake locations relative to fault length-scales[18], a particularly challenging task in ocean island settings, where island and inter-island geography limit seismometer network coverage and geodetic observations.

In contrast to classical orthogonal rifts (e.g., Afar, East Africa), transtensional environments provide insight into the interaction between pre-existing faulting structures and magmatism, and how they accommodate plate motion. In SW Iceland, dikes strike perpendicular to tectonic extension[7], triggering seismicity along oblique faults that accommodate the strike-slip component of plate motion[19]. However, such cases still somewhat obscure the more direct role of faults in magma ascent, leaving a knowledge gap about how larger, proximal faults influence magma movement[20] and the feedback mechanisms between tectonics, magmatism and seismogenesis.

Unlike Iceland, the coexistence of active magmatic systems and seismogenic crustal-scale faults in the Azores hotspot and triple-junction region offers a clearer view of the interplay between tectonic structures and magmatism[21] (Fig. 1a). The Azores is shaped by the diffuse boundary between the Eurasian and Nubian plates, with the Terceira Rift (Fig. 1a) as the main spreading system, where slow (~4.5 mm/yr), WSW-ENE-oriented relative motion[22–25], produces right-lateral transtension[26], which is consistent with the mechanisms of past earthquakes in the region[27–29] that exhibit rift-parallel normal faulting and off-rift strike-slip faulting (Fig. 1a). Deformation across the Azores is highly distributed, with fissure systems, off-rift transforms, grabens, and extension-oblique volcanic ridges[21,22,26,30–36], such as that expressed by São Jorge Island, located ~40 km SW of the Terceira Rift axis (Fig. 1b).

São Jorge is a narrow island, which is 55 km long and up to just 7 km wide (Fig. 1b, c). Erupted basalts show signs of a weak mantle plume[37], and fissure eruptions along well-defined scoria cones have generally migrated 5 cm/yr westwards over 750 kyr[22]. These volcanic alignments lie along strike-slip fault zones, reflecting magmatic-tectonic interplay[22,24]. In the west of São Jorge, two major fault zones (Fig. 1c and Fig. 2), Picos (P-FZ; striking WNW-ESE) and Pico do Carvão (PdC-FZ; striking almost E-W) feature scoria/spatter cones and craters[35]. The PdC-FZ hosted eruptions in 1580, 1808, and a possible submarine eruption in 1964[38,39]. Historically, the island hosted one of the largest earthquakes in the Azores, estimated at $M$ 7.5, in 1757[40–43]. Paleoseismology shows that the right-laterally offset PdC-FZ (Fig. 2) is currently the fastest-slipping fault (2.6–3.4 cm/yr) on São Jorge[44] and can generate $M$ ~7 earthquakes[35,44].

Starting on 19 March 2022, São Jorge experienced a highly active seismic swarm[45], with microseismicity continuing for at least 2 years. In this work, high-precision hypocentre locations from onshore and off-shore seismic data, reconciled with modelling of surface deformation observations, reveal that a crustal fault can serve as both a pathway and a barrier to magma ascent.

## Results
### Geodetic observations and dike-opening model
Daily GNSS solutions from nine stations in the Azores (Fig. 1b; see Methods) reveal intense deformation beginning on 19 March 2022, when the seismic swarm commenced (Fig. 3a and Supplementary Fig. 1). Stations on Pico (PIED; AZTP) and Graciosa (AZGR; ENAO) moved horizontally away from São Jorge by up to 18 mm (Fig. 1b). Faial (HORT) and Terceira (PAGU; TERC) show negligible motion. Stations on São

Jorge (QEMD, VLAZ) moved roughly eastward by up to 10 mm. QEMD also subsided by 5 mm. Some stations exhibit rapid deformation over a single day, whereas others show a slightly slower three-day-long deformation transient (Supplementary Fig. 1). Aside from long-term interplate strain accumulation, no substantial deformation occurred before or after the onset of seismicity on 2022-03-19 (Fig. 3a and Supplementary Figs. 1 and 2).

Similarly, analysis of processed Sentinel-1A interferograms (see Methods) reveals no significant deformation signals before or after the onset of seismicity (Supplementary Figs. 3–5). During 15–21 March 2022, independent interferograms spanning 2022-02-13 to 2022-03-21 (ascending) and 2022-03-15 to 2022-03-27 (descending) detected up to +6 cm line-of-sight deformation on central São Jorge (Fig. 1c and Supplementary Figs. 4, 6). Both ascending and descending passes show a similar uplift pattern, symmetrical across the island's flanks, with lobes of maximum uplift at the coastlines ('L1' and 'L2' in Fig. 1c) and negligible deformation along an 8–10 km saddle sub-parallel to the island's axis. All deformation detected by InSAR occurred between 15 and 21 March 2022 (Supplementary Figs. 3–5), consistent with the GNSS signals during the onset of seismicity.

InSAR fringe patterns and GNSS displacements north and south of São Jorge (Fig. 1b, c) indicate substantial opening, and thus a zone of tensile opening striking slightly oblique to the island's long axis. Joint inversion of three-component GNSS and both InSAR tracks with a Bayesian 3D mixed boundary element method (BEM) inversion (see Methods) yields a quadrangular dike-opening solution. This model is illustrated in Fig. 4, with uncertainties in the model parameters presented in Table S1 and Supplementary Figs. 7–9, and which are illustrated by the fuzzy pink shading in Fig. 4b–d, based on an ensemble of 1000 acceptable solutions. The modelled dike comprises a ~6-km-long by ~25-km-deep, WNW-ESE (285° strike), near-vertical (83° ± 7° dip) geometry beneath the western-central island, just east of the seismicity (Fig. 4). The dike has a maximum opening of 72 cm with a total volume of $79 \times 10^6$ m³. The top of the dike is located ~1.65 ± 0.60 km below the surface, as required by both ascending and descending InSAR tracks, and the near-field GNSS displacements on São Jorge (QEMD, VLAZ). The inversion yields a bottom depth of the dike at ~26 km below sea level (bsl), within the lithospheric mantle[46,47]. The vertically extensive nature of the modelled dike is required by the far-field GNSS stations on adjacent islands; however, the bottom depth is one of the most uncertain aspects of the model (formal uncertainty of ± 8 km; Supplementary Figs. 7–9 and Table S1), with the best-fit model at the deeper end of the ensemble solutions, but most models reaching beyond 20 km depth bsl (Fig. 4b). The quadrangular model's top-length (4300 ± 2500 m) and its shear angle (−11° ± 11°; i.e., slants to the west) are reasonably well constrained, although the uncertainties in these parameters translate to a substantial variation in its along-strike areal extent (fuzzy pink shading in Fig. 4d). Although far-field GNSS vertical time series are noisy, our model predicts well the observed vertical and horizontal GNSS displacements on São Jorge and adjacent islands (Supplementary Fig. 10), as well as both ascending and descending interferograms (Supplementary Fig. 11).

To test the robustness of the quadrangle geometry parameterisation, we conducted an extensive exploration of the model space. These tests included evaluating a wide range of source geometries, such as single and multiple vertical, inclined, and curved dikes, spherical and ellipsoidal pressure sources, faults with both right- and left-lateral shear, and combinations of these (Text S1). Though physically plausible, none of these alternative configurations improve fits to both GNSS and InSAR data (Table S2). Our sub-vertical quadrangular dike model emerged as the preferred solution because it is the only configuration that reconciles all datasets with small residuals and physically plausible model parameters.

Previous analysis of GNSS signals suggests a 2-day-long, up to 4 cm, precursory uplift signal at QEMD starting on 2022-03-16[45]. However,

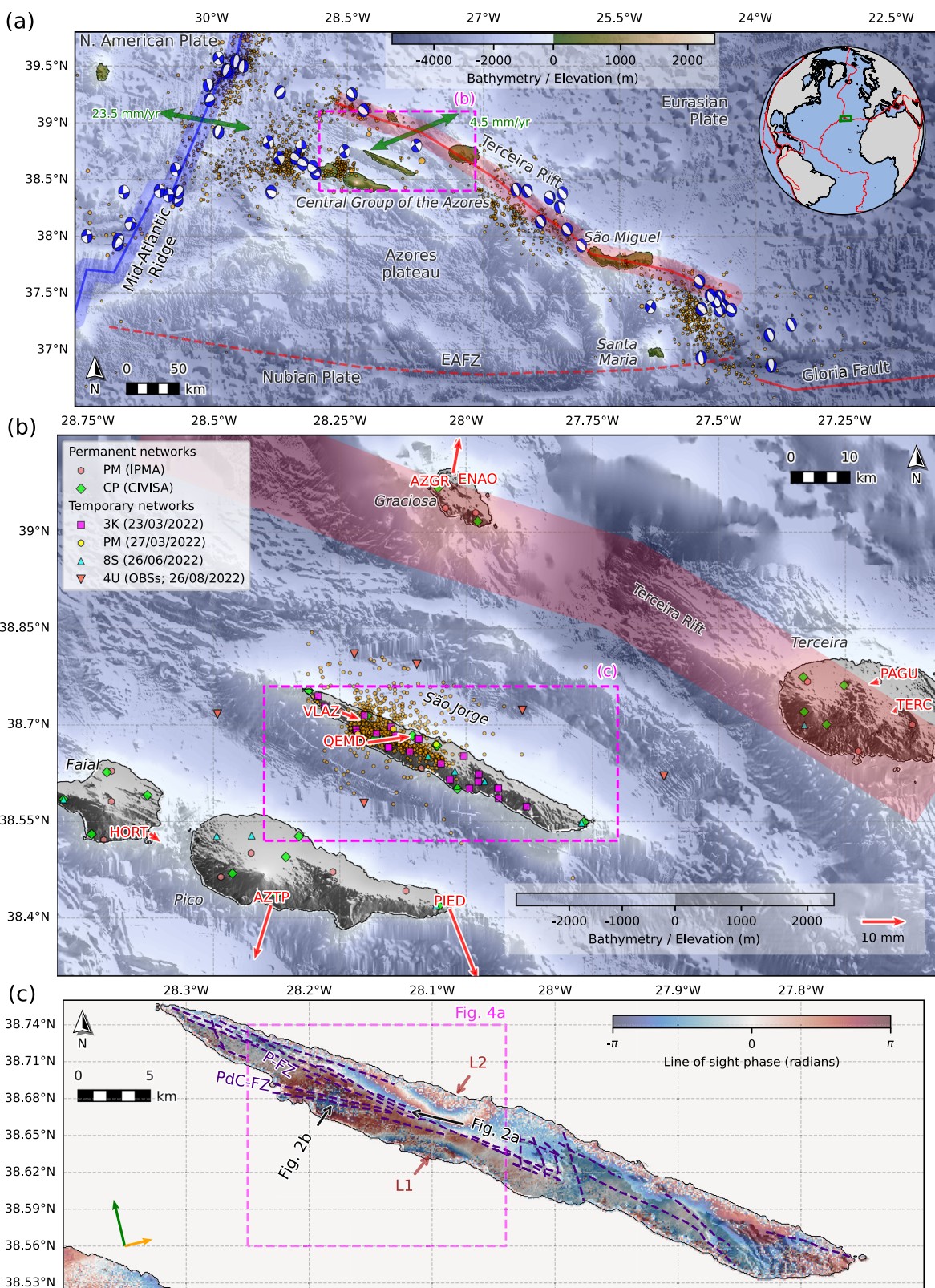

**Fig. 1 | Seismotectonic context, stations, and surface deformation. a** Tectonic configuration of the Azores region, showing pre-2022 M > 3.5 background seismicity (orange circles)[140]. Double-couple components are plotted for moment tensors[28]. Green arrows show relative plate motions[23]. EAFZ East Azores Fracture Zone. **b** Map of the Central Group islands of the Azores showing locations of permanent and temporary seismic stations. Red arrows show horizontal GNSS solution displacement vectors for the onset of the seismic swarm (17/03/2022–22/03/2022), with station names labelled in red. Orange circles show the routine

seismic locations of the 2022 seismic swarm from Centro de Informação e Vigilância Sismovulcânica dos Açores (CIVISA). **c** Wrapped ascending interferogram from the Sentinel-1A satellite spanning 2022/02/13– 2022/03/21. Supplementary Fig. 6 shows the unwrapped interferogram. Green and orange arrows indicate the satellite azimuth and look direction, respectively. The brown annotations indicate the two uplift lobes in the interferogram. Dark blue dashed lines indicate mapped faults and volcanic alignments on São Jorge[35,44]. Black arrows show the aerial imagery in Fig. 2.

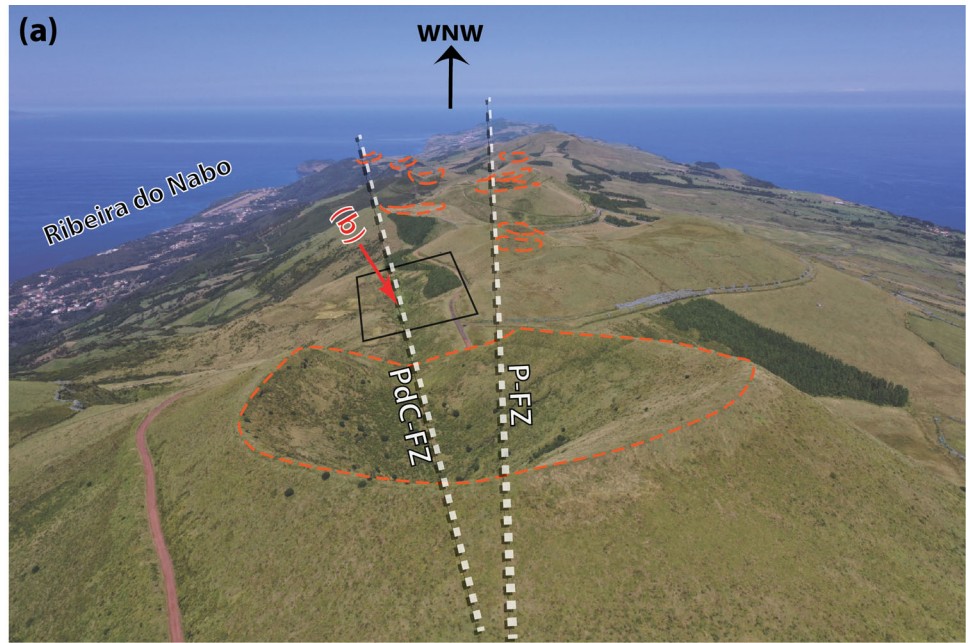

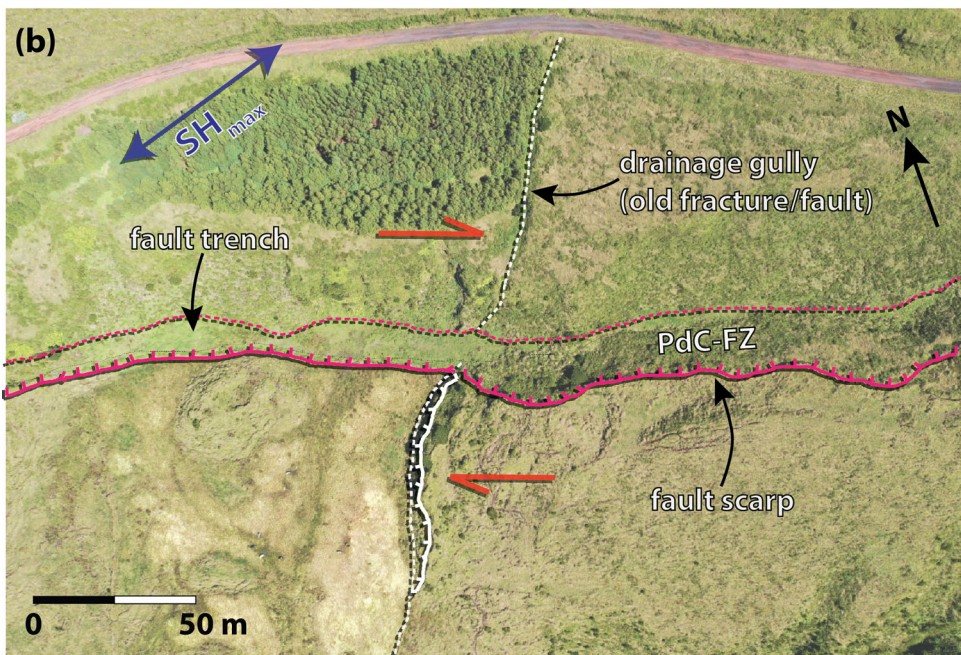

**Fig. 2 | Drone imagery showing surface faulting on São Jorge near the 2022 unrest area.** P-FZ Picos fault zone; PdC-FZ Pico do Carvão fault zone. **a** View looking west-north-west (see Fig. 1c for camera position and direction). Dotted orange lines highlight the vents of past fissure eruptions. **b** Evidence of right-lateral offsets along the PdC-FZ (see Fig. 1c and Fig. 3a for location).

given vertical-component noise levels and our daily solution uncertainties, we cannot confidently identify such a signal in our Precise Point Positioning (PPP) solution, nor does our independent network-based GNSS solution corroborate it (Text S2; Supplementary Fig. 2). The previously reported solutions[45] from QEMD show multiple similar-sized anomalies at other times, which were left uninterpreted. Furthermore, we observe no such signal at the nearby GNSS station VLAZ (Supplementary Fig. 1). Nevertheless, we conducted forward modelling of the proposed sill source at the depth of the main seismicity cluster[45] (Text S2). Such a magmatic source would have produced cm-scale signals at VLAZ, which we do not observe. Therefore, we suggest that any precursory, rapid transient signals are likely GNSS processing artefacts; or, if the precursory uplift signal at QEMD is real, it is too small, uncertain, and localised to be confidently quantified or modelled.

## Seismicity distribution and its temporal evolution

Seismic data come from permanent seismometer networks, that were complemented and densified by temporary stations installed in response to the seismic swarm (Fig. 1b and Supplementary Fig. 12); (see Methods and Data availability). Six temporary ocean-bottom seismometers (OBS) offshore São Jorge enhance the inherently poor coverage due to the islands' geographic distribution and the narrow, elongate geometry of São Jorge (Fig. 1b). Using data from 83 stations, our automated workflow (see Methods) yields high-precision relative relocations for 18,049 events. All events have a maximum azimuthal gap of <240°, a maximum RMS residual of 0.4 s, and a maximum semi-major axis length of 1500 m. The median relative depth uncertainty of all events in the catalogue is 46 m.

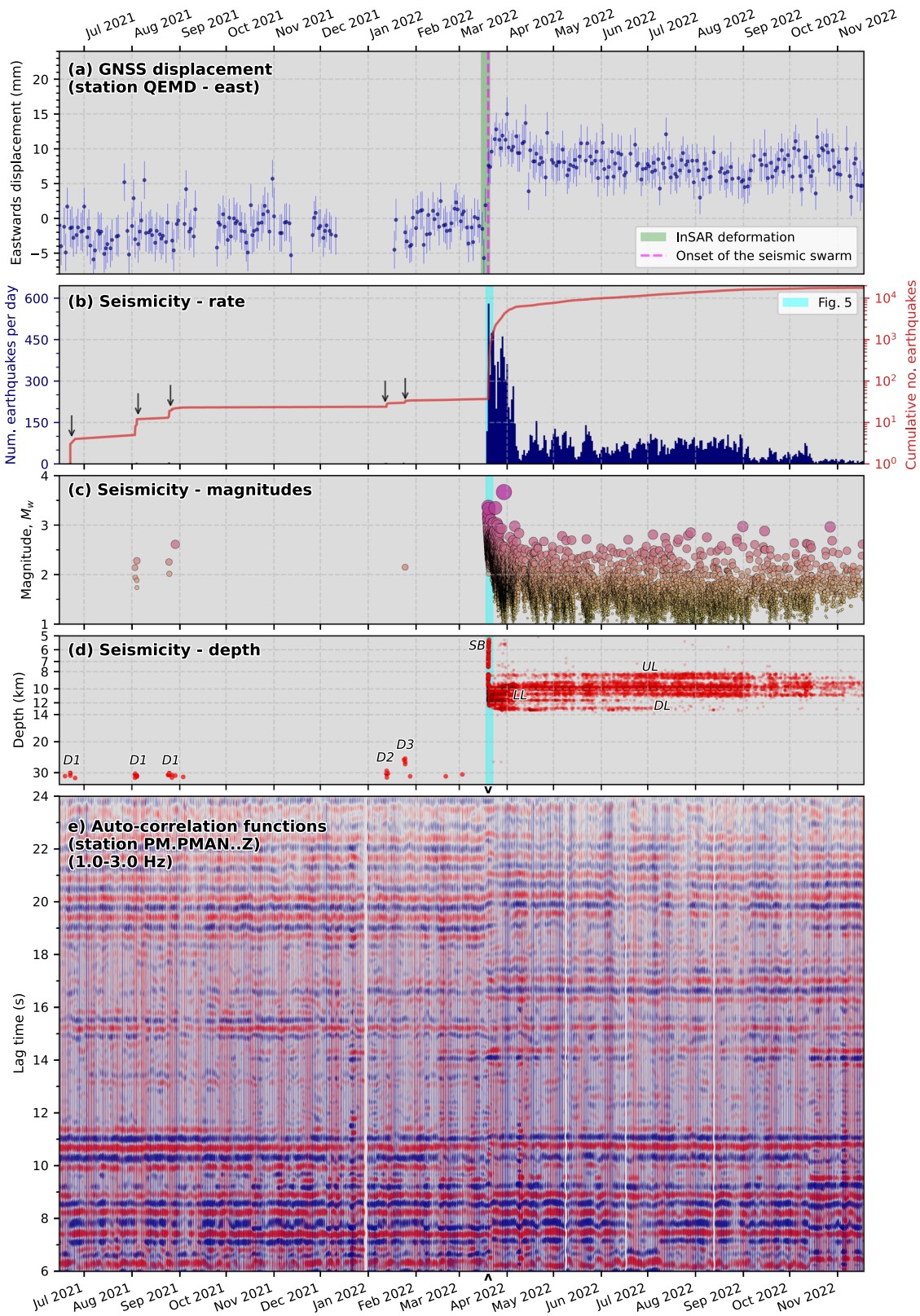

From June 2021 (9 months before the onset of unrest), we identified 36 events at -30 km depth bsl south of São Jorge (Figs. 3b–d and 4). The largest of these earthquakes has a moment magnitude ($M_w$) of 2.6, which was detected by nine stations. Our high-precision relative relocations show that these events, forming three main clusters, progressively migrated northward (labelled Deep, D1-3 in Figs. 3b–d and 4), starting beneath the Canal de São Jorge (São Jorge channel, south of the island), to -2 km inland of the coastline, gradually shallowing from -32 km to -25 km depth (Fig. 4b). Waveform template matching[48] with these well-located events finds a greater number of events (150) that could not be accurately located (Fig. 3c), affirming short, migrating bursts of seismicity (Fig. 3b–d) separated by more extended aseismic periods. Based on our results, the two weeks immediately before the onset of the seismic swarm were aseismic.

**Fig. 3 | Evolution of the 2022 São Jorge unrest and precursory seismicity.**
**a** Processed daily eastwards GNSS displacements at station QEMD (location shown in Fig. 1b) with error bars, and the inferred period of InSAR detectable deformation labelled (Fig. 1c and Supplementary Figs. 3–5). Displacements are given within a global reference frame, with the secular trend removed. Time series for other GNSS stations are shown in Supplementary Fig. 1. Panels (**b**–**d**) show the time evolution of seismicity rate, moment magnitude, and depth, respectively. The blue vertical stripe denotes the time window shown in Fig. 5. Downward-pointing black arrows in (**b**) denote bursts of deep precursory seismicity. In (**c**), circle sizes are scaled by magnitude. Note the logarithmic depth axis in (**d**). Labels denote the main seismicity clusters shown in Figs. 4–6 and described in the main text. **e** Autocorrelation function (ACF) results from the vertical component of station PM.PMAN (location shown in Fig. 4) show changes in subsurface properties at the onset of the seismic swarm. Red and blue colours represent positive and negative amplitudes, respectively, in the ACF waveforms. At the onset of the seismic-volcanic unrest (indicated by the small black arrowheads above and below the time axis), the ACFs at lag times of <8 s and >18 s become less coherent and shifted in time.

The seismic swarm started abruptly at 17:02 UTC on 19 March 2022, coinciding with the onset of geodetically observed deformation (Fig. 3a–b). We find many earthquake-like signals that begin at least 3 hours earlier and accelerate in rate, although they are too small to be located. During the first 9 hours, we find 120 events, of which nine have $M_w > 3.0$, with a very high rate of seismicity (up to 50 earthquakes per hour) persisting over the first 24 h (Fig. 5a). We detected two phases of tremor-like bursts (see Methods) during this time (Fig. 5b), the first coinciding with the start of the swarm; the other ~6 h later both marking the onset of accelerating seismicity rate. The first locatable earthquake in the swarm occurred at 9 km depth within the vicinity of the modelled dike, with seismicity then migrating upwards to ~5 km depth and spreading bilaterally, forming two shallow branches of rapidly-upward migrating seismicity at ~400 m/hour ('SB' in Figs. 4 and 5). Almost concurrently, seismicity then migrates ~10 km westward and ~5 km downward at ~400–800 m/hour to depths of 8–13 km, where the main zone of seismicity develops and remains exclusively for subsequent months.

Subsequent westward and downward migration formed the main zone of seismicity at 8–13 km depth, residing near the estimated crust-mantle transition beneath São Jorge based on our inverted 1D velocity model (Fig. 5) and independent estimates of Moho depth[47]. These epicentres, including the earlier shallow branches of seismicity (SB), form a narrow (<500 m width), WNW-ESE lineation along the mapped surface trace of the PdC-FZ (Fig. 4a), slightly oblique to the island's long axis. Along a profile perpendicular to the PdC-FZ (Fig. 4b, c), events at 8–12 km form a near-vertical, NNE-dipping structure directly west of the modelled dike, comprising distinct sub-streaks. On a PdC-FZ-parallel cross-section (Fig. 4d), most events align along two gentle (~10°) west-dipping lineations: one at 8–11 km depth (labelled upper lineation, 'UL' in Fig. 4d) with substantial internal complexity showing many sub-vertical filaments (Fig. 6), and the other at 11–12 km with sub-horizontal streaks (lower lineation, 'LL' in Fig. 5d). These are separated by a ~1.5-km-wide aseismic region ('AS' in Fig. 5d). Additional seismic clusters and streaks occur even deeper, at 12–14 km ('DL'). Based on clustering of waveform pair cross-correlations[49], we find 22 distinct clusters comprising at least 50 events (Fig. 6a). Clusters in UL appear to show a slow upward migration with time (Movie 1), whose front is consistent with a fluid diffusivity of ~0.014 m²/s (Fig. 6b, c).

Following the early intense period of high-rate seismicity and tremor, the remaining seismicity appears as high-frequency signals with impulsive onsets (Fig. 6d), typical of VT events. After the first nine hours of the swarm, we found no additional tremor-like signals. The seismic swarm overall has a remarkably high $b$-value of $2.4 \pm 0.1$ (Supplementary Fig. 13) based on $M_w$ estimates from spectral inversion (see Methods; Supplementary Figs. 14 and 15). Well-constrained focal mechanisms (see Methods; Supplementary Fig. 16) show strike-slip faulting, with one nodal plane parallel to the main lineation of seismicity and the PdC-FZ, indicating left-lateral faulting (Fig. 4a). Dense coverage of P-wave polarities across the focal sphere for ~40 earthquakes indicates double-couple failure (Supplementary Fig. 16).

### Subsurface velocity changes

To assess possible subsurface structural changes before, during, and after the seismic swarm, we analysed ambient-noise phase autocorrelation functions[50] (ACFs) from 1–3 Hz-filtered vertical-component waveform data to yield subsurface P-wave reflection responses (see Methods). Given that only three permanent stations on São Jorge captured the onset of the unrest, we obtained high-quality, stable ACF results from PM.PMAN and PM.ROSA (station locations shown in Fig. 4a, d). Pre-unrest reference ACFs from May to September 2021 show coherent arrivals at lag times of ~6–24 s (Fig. 3e and Supplementary Fig. 17). When the seismic swarm and surface deformation began on 19 March 2022, PM.PMAN's ACFs showed a sharper shift to higher lags, especially in the 8–11 s and 18–24 s ranges (Fig. 3e), indicating velocity reductions at <10 km and >15 km depth, respectively, based on our 1D velocity model (Fig. 4b and Supplementary Fig. 18). Stronger reflections also appear at 16–18 s on 2022-03-19. In contrast, PM.ROSA's ACF shows less stable, less systematic variations at the onset of unrest (Supplementary Fig. 17). These two stations are ~16 km apart, and given the waveform frequency and inferred depth of subsurface changes, we infer that PM.PMAN sensitivity kernel includes the geodetically imaged main dike that ascended vertically, whereas PM.ROSA is too far west of the dike (Fig. 4).

## Discussion

Using our high-precision seismicity catalogue, focal mechanisms, geodetic model, and seismic autocorrelation function imaging, we have developed a model for the 2022 São Jorge seismic-volcanic unrest, illustrated in Fig. 7, and detailed below.

Our geodetic dike-opening model indicates magma intrusion from 1 to 26 km bsl (Fig. 4), with no significant inflation preceding it (Supplementary Figs. 1 and 2). The precursory earthquakes in the upper mantle (25–35 km depth) correlate well with the modelled dike's base (Fig. 4). Even though our best-fitting model dips slightly to the north, many acceptable models in the ensemble have slight southerly dips (Fig. 4b and Supplementary Figs. 7–9) and even directly reach the final phase of deep seismicity that was active just ~2 months before the main seismic swarm (D3)[51,52]. Such deep precursory seismicity[18,53–55] is commonly interpreted as reflecting magma accumulation, destabilisation, or migration between deep reservoirs[55–57], with a depth of 25–35 km consistent with an upper mantle storage region inferred from fluid inclusion barometry of São Jorge lavas[52].

Our geodetic observations and dike model imply rapid, stealthy magma ascent, possibly involving near-simultaneous dike-opening in the crust and upper mantle. Vertically extensive and rapid melt ascent is also supported by our ambient noise ACF results, which show rapid structural changes at PM.PMAN, the station located closest to the ascending dike (Figs. 3d and 4). Our seismicity catalogue has a higher completeness magnitude during the first few days of the swarm (Fig. 3c), during dike emplacement. This issue likely arises from the initial high rate of seismicity (Figs. 3b and 5a), and is further compounded by the fact that there were only three operational seismic stations on São Jorge at the onset of the crisis, with the network densified a few days later (Supplementary Fig. 12). Still, we consider that our seismicity catalogue provides a clear overall picture of the processes during and after dike emplacement.

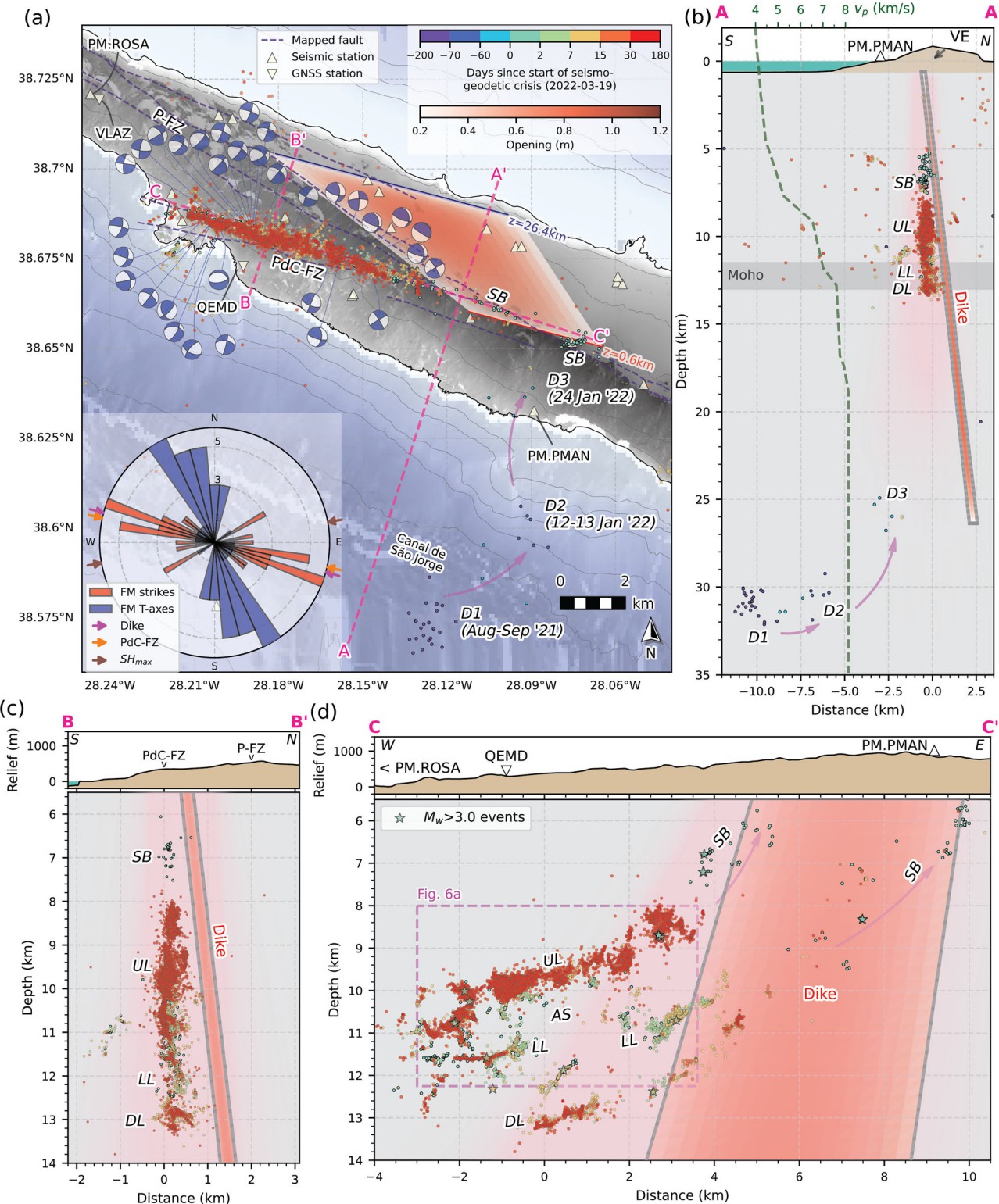

**Fig. 4 | High-precision seismicity relocations and joint GNSS-InSAR dike-opening model. a** Map view, with seismic events coloured by time relative to the onset of the seismic swarm. The red-orange shaded area represents our geodetic quadrangular dike-opening model. Purple dashed lines are surface fault traces and volcanic alignments[24,44]. Magenta arrows show the migration of precursory and early seismicity. White triangles are seismic stations. Elevation contours are plotted at 250 m intervals. The inset rose diagram shows focal-mechanism strikes relative to the dike, and the mean strike of mapped faults, along with a comparison of the inferred regional SHmax direction (Fig. 1a) with focal-mechanism T-axes.

**b** Seismicity-perpendicular cross-section, A-A'. IE island edifice. The dashed green line shows the P-wave velocity from our 1D model (Supplementary Fig. 6).
**c** Seismicity-parallel cross-section, with a focus on the main clusters of seismicity.
**d** Along-strike cross-section, C-C'. Labelled clusters of seismicity are discussed in the text. Note the break in range of the vertical axes between the relief and seismicity cross-sections in (**c**, **d**). In all cross-sections, the fuzzy pink shading illustrates model uncertainty, based on an ensemble of the 1000 lowest-misfit geodetic diking solutions. These models represent the range of plausible dike geometries consistent with the observed deformation (InSAR and GNSS).

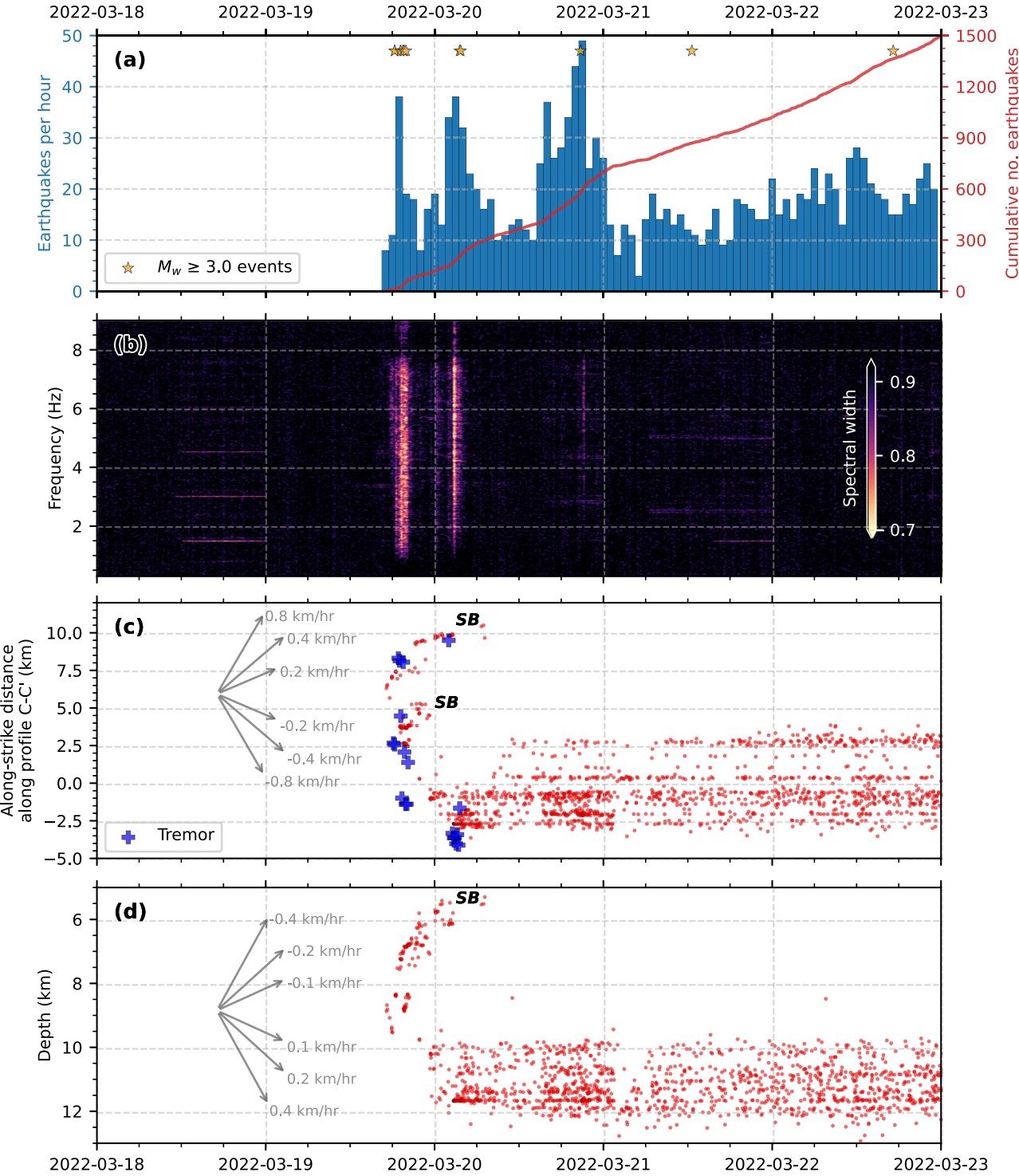

**Fig. 5 | Onset of the seismic swarm. a** Earthquake rate per hour (blue bars), cumulative number of events (red line), and $M_w > 3$ events (orange stars). **b** Spectral width of the network covariance matrix computed using vertical-component waveforms from permanent seismic stations. A narrow spectral width indicates a coherent, spatially localised source, such as tremor. The lower two panels show the temporal variation of seismicity position **c** along strike (WNW-ESE position; Profile C-C' in Fig. 5, and **d** with depth. Grey arrows show indicative lateral and depth migration velocities. Lateral positions of tremor-like bursts are denoted by blue plus symbols in (**c**) based on maximum likelihood positions from back-projection of smoothed cross-correlation envelopes. 'SB' represents the early, upward migrating branches of shallow seismicity.

The deep (>9 km) and shallow (<5 km) portions of melt ascent were aseismic (Fig. 4). Only two moderately dipping branches of seismicity ('SB' in Figs. 3–5), plus associated tremor, appear to be the only seismogenic response of the ascending intrusion; these initiated at ~8–9 km depth, and rapidly migrated upward to ~5.5 km depth during the first

few hours of the seismic swarm. Given that both of these shallow seismicity branches became inactive within a few hours, and considering their time-space correlation with the geodetic dike model, along with its inherent uncertainty, these shallow earthquakes and tremor-like bursts (Fig. 5b, c) likely mark the flanks of the ascending dike.

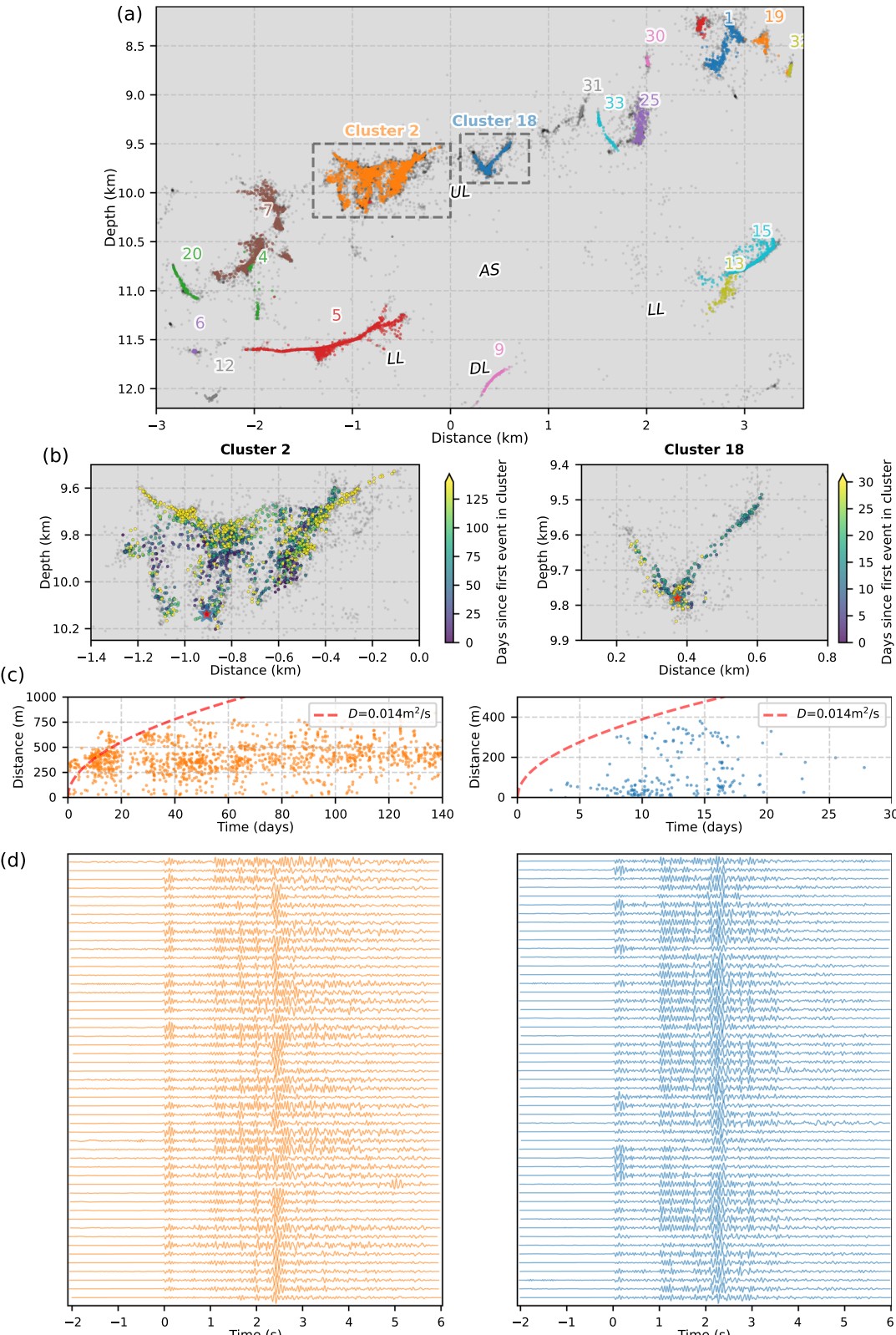

**Fig. 6 | Seismic event clustering. a** Enlarged view of the main zone of seismicity along the strike-parallel cross-section (Fig. 4d), with events coloured by their waveform correlation-derived cluster. Only clusters with at least 50 events are shown. The panels below show the details of two distinct clusters: 2 (left) and 18 (right). Black dots show the unclustered seismicity. **b** Zoomed-in view of these two seismicity clusters, with events coloured by their time relative to the first event in each cluster (red star). **c** Events in each cluster are plotted as a function of time and distance relative to the first event in each cluster. The red dashed line represents an approximate fit for the diffusivity of the migrating seismicity front. **d** Examples of highly similar waveforms (P-wave cross-correlation coefficient >0.87 with the first earliest in each cluster) in the frequency band 2–40 Hz, with a maximum of 50 events shown for visualisation.

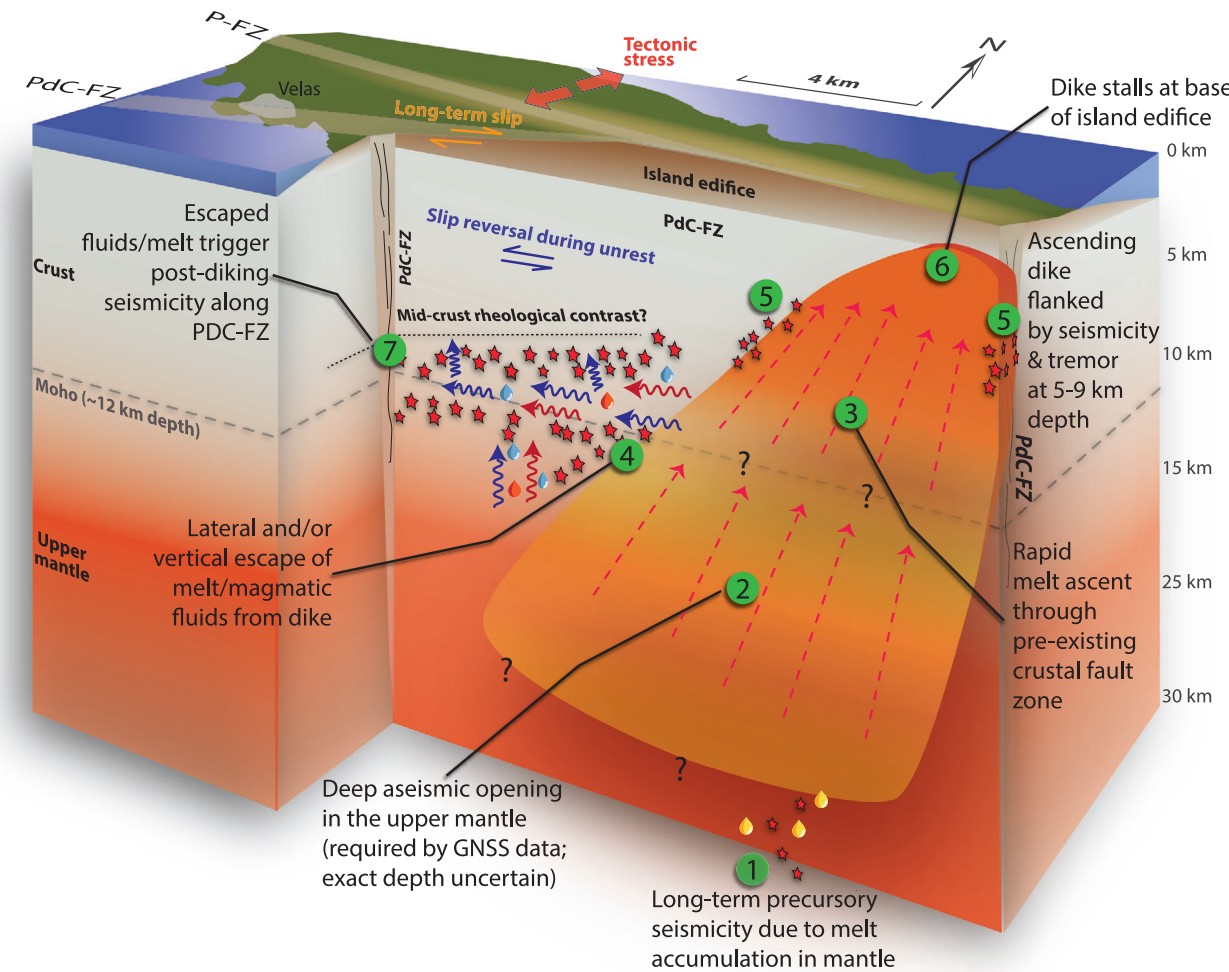

**Fig. 7 | Interpretation. a** 3D schematic view cut along the strike of the PdC-FZ and the intruded near-vertical dike showing the processes driving the São Jorge seismic-volcanic unrest in 2022. Note the exaggerated width of the dike. Our quadrangular model is uncertain about the exact depth of the dike. The broken interpreted flow arrows in the dike represent a possible scenario of vertical segmentation of diking dynamics.

Our geodetic model shows that magma stalled ~1600 m below the surface, near the estimated base of the island edifice, based on surrounding bathymetry (Fig. 4). We are not aware of any monitoring infrastructure, data, or reports (peer-reviewed or otherwise) of gas geochemical anomalies observed during the dike intrusion and start of the seismic swarm. The short-lived nature of the intrusion and the different permeability of the volcanic island edifice may have inhibited efficient gas escape to the surface. Magma propagation is governed by a combination of fracture toughness, overpressure, buoyancy, and viscosity[3], so a primary explanation for stalling at this shallow depth is that the volume of magma, and hence the resulting overpressure, may have been insufficient to overcome the lithostatic and edifice-related stresses at shallow depths. A dynamic increase in magma viscosity, such as due to devolatilisation[4], may also have inhibited magma ascent. However, exsolution of volatiles and an increase in viscosity are unlikely to occur on such a short timescale, given the rapid ascent of the dike and the subsequent triggered seismicity in the adjacent region.

The dike intruded parallel to the PdC-FZ and main zone of VT seismicity streaks and lineations at ~8–12 km depth, east of the surface-mapped fault zone, where the fault is likely buried by lavas and scoria cones (Fig. 4). The fault therefore provides a direct structural and hydraulic connection between the dike and seismicity. Unlike typical dike intrusions, where seismicity surrounds the dike in a 'dogbone' pattern[16,58–65], the 2022 São Jorge seismic-volcanic unrest shows the main zone of seismicity confined to only one flank of the geodetically modelled dike, with no activity within or to the east of the dike. This pattern highlights a discrepancy between interpretations derived from seismicity or geodesy alone during volcanic unrest. Few similarly oriented faults have been mapped on the eastern side of the island (Fig. 1c), and the long-lived segmentation of three volcanic complexes across the island[24] might have controlled the localisation of triggered seismicity. The PdC-FZ and seismicity lie in a zone positively stressed by the modelled dike intrusion, both for left- and right-lateral strike-slip failure on receiver faults (>50 kPa; Supplementary Figs. 19 and 20). However, similarly oriented and stressed nearby structures, like the P-FZ, remained aseismic. Such asymmetrically localised seismicity is thus likely due to fluid-triggered seismicity, with the PdC-FZ providing a direct hydraulic connection between the westward-tilting intruded dike and earthquakes, thereby enabling fluid-driven triggering. The PdC-FZ is a major mantle-rooted structure in the central Azores that accommodates regional transtension[24,38,44,52], and appears to have been the locus of recent magmatic and seismogenic activity on the island, given its recent eruptive history and its capability to generate $M_w$ ~7 earthquakes[44], likely involving complete crustal rupture. Such events, along with long-term tectonics, may have pre-loaded the fault in the lead-up to the 2022 seismic-volcanic unrest.

The highly active swarm had an unusually high *b*-value of ~2.4 (Supplementary Fig. 13), compared to the global volcanic average of

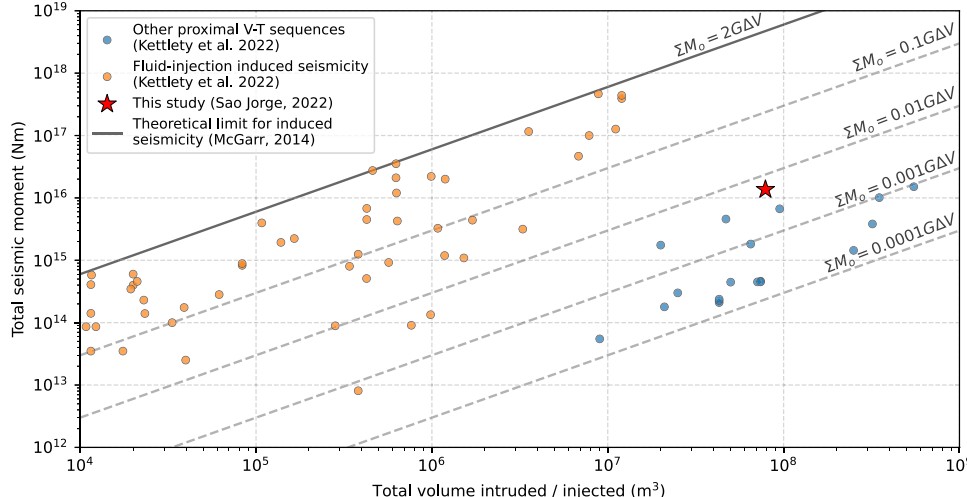

**Fig. 8 | Global context.** Comparison of intrusion volume versus total seismic moment with other proximal volcano-tectonic (V-T) sequences and fluid injection-induced seismicity[66]. Grey dashed lines indicate orders-of-magnitude smaller scaling of the McGarr[141] relationship to account for comparatively lower seismic efficiency in volcanic settings[66]. These relationships are depicted with a shear modulus of G = 30 GPa.

-1.7[66–68]. Moreover, the total seismic moment released (-$10^{16}$ Nm, or $M_w$ 4.7) is large relative to the intrusion volume (-$10^8$ m³) compared to other dike-related VT sequences[66,68] (Fig. 8). These abnormal characteristics are likely due to a fluid-rich environment[69] along the PdC-FZ in the lower crust, with the $M_w$ to intrusion volume scaling slightly tending toward seismicity caused by fluid injection, which has greater seismic efficiency than most volcanic sequences[66] (Fig. 8). The rapid westward and downward migration of seismicity (Fig. 4), which left a -2 km thick aseismic zone ('AS') between three seismic lineations ('UL', 'LL', and 'DL'; Fig. 4), may indicate that, following the main dike stalling due to the compressive load of the island edifice, a laterally propagating branch of magmatic fluids may have then initiated at the western edge of the vertically-stalled intrusion, at 9–10 km depth[63,70,71], and escaping into the permeable PdC-FZ. Forward modelling (see Methods; Supplementary Fig. 21), however, shows that such melt branches would need to be small (<$10^2$ m³ total volume) to avoid inducing any observable surface deformation. Such thin branches of melt might be consistent with the very thin lineaments of seismicity that we observe (Figs. 4d and 6a). Destabilisation of melt at the western edge of the main dike may be dynamically favourable given the dike's tendency to lean to the west in our ensemble of geodetic solutions (Fig. 4 and Supplementary Figs. 7–9). We cannot rule out these branches of melt intermixing with fluids, nor can we exclude vertical fluid ascent from the westward-tilting deeper parts of the dike (Fig. 7), or a combination of both lateral and vertical transport. Some filament-like seismicity clusters, such as Clusters 2, 7, and 18 (Fig. 6 and Movie 1), show complex fracture-like geometries and an upward migrating front over weeks to months with slow diffusivity (-0.001–0.002 m²/s), suggesting long-lived upward fluid migration along fractures[72] in a likely distributed damage zone of the PdC-FZ.

Overall, lacking evidence for a substantial laterally propagating melt intrusion in AS, between LL and UL, we propose that fluid pore pressure increases, driven by thermal pressurisation[73], small channels of melt, and magma devolatilisation[74–77], triggered the main seismicity. The PdC-FZ likely facilitated fluid channelling[78] through the lowermost crust near the crust-mantle boundary at -12 km. Lateral melt branching, exsolution, and subsequent devolatilisation likely increased magma viscosity[4,79,80], helping the dike to stall -1.6 km below the surface. The resulting change in static stress produced the observed focal mechanisms, which indicate left-lateral seismogenic shear along the PdC-FZ, indicating a 90° rotation of stress relative to the long-term right-lateral motion[21,22,26,30–33,35,44] (Figs. 1a, 2, 4a). Such stress rotations

due to dike intrusions have been reported previously, with a possible mechanism of shear dilatancy driven by overpressured magma in a relatively weak background stress field[13,15,74,81].

Although strongly supported by the geodetic data and ACF results, a near-vertical dike-opening in a matter of days, from 26 to 1.6 km depth, is an unusual aspect of the 2022 São Jorge unrest. However, magmatic intrusions propagating rapidly at -0.1–1.0 m/s[58,82], with partial or total aseismicity, especially at mantle depths, are not without precedence[57,82]. We infer that because the dike intruded into a mature fault zone, few new fractures had to be created, leading to aseismic opening, possibly accompanied by devolatilisation and shear dilatancy at the dike tip. Alternatively, the longer-term GNSS deformation transients at far-field stations (e.g., AZTP on Pico, ENAO, Graciosa, and to a lesser extent QEMD on São Jorge; Supplementary Fig. 1) may indicate that the shallower and deeper parts of the modelled dike deformed on different timescales. In this case, we speculate that the mantle segment deformed more slowly due to its inherently different mechanical properties. Such a vertically segmented dike may be consistent with the ACF imaging results, which imply a bimodal depth distribution in subsurface changes, with clear changes at lag times <8 s and >16 s, but without any apparent changes at intermediate depths. However, we cannot formally quantify this hypothesis given the available time resolution and accuracy of geodetic observations. Regardless of the exact behaviour at depth, such a stealthy and rapid ascent of magma presents a challenge for tracking dike intrusions and forecasting their eruptive potential.

Another remaining issue is that residual horizontal GNSS vectors from our preferred quadrangular diking model at stations on islands to the south (AZTP and PIED on Pico; HORT on Faial) show a consistent eastward trend (Supplementary Fig. 10), which could indicate an aseismic source of slow left-lateral slip[83]. This trend is particularly intriguing given the rotated left-lateral slip of the swarm seismicity. Therefore, we inverted the geodetic data for a sheared dike model (Text S1.1). However, we found that the resulting model unsatisfactorily increases the GNSS and InSAR residuals (Table S2), and results in a physically implausible near-horizontal shear dike located too far north of the seismicity. The observed left-lateral shear residual may reflect transient stress perturbations during magma emplacement, possibly associated with shear dilatancy or local stress rotations, which may be physically linked to the rotated focal mechanisms. Alternatively, the shear residual may be due to the differential timescale of opening between the mantle and crustal intrusion segments, as described

above. However, the sparse GNSS dataset cannot provide confident constraints on such sources.

Overall, our findings highlight clear volcano-tectonic interactions, most evident in transtensional environments. The dike intruded obliquely to the minimum compressive stress (Fig. 4a) but was guided by the PdC-FZ, which acts as a major transtensional rift in the central Azores. The magma intrusion likely targeted a weaker fault segment near the PdC-FZ and P-FZ junction (Fig. 4a). The PdC-FZ, capable of generating $M$-7[26,35,44] earthquakes, likely exhibits along-strike variations in strength and coupling[84,85]. The seismic swarm, confined to the lowermost crust (Figs. 4 and 6a), possibly marks a rheological transition separating the locked shallow fault from the creeping upper mantle. Although the dike increased PdC-FZ stress for both left- and right-lateral slip modes (Supplementary Figs. 19 and 20), the seismic swarm released only $M_w$ 4.7 of left-lateral slip, leaving the likelihood of a future large earthquake, which would presumably release regional tectonically accumulated right-lateral stress, no less likely. Such major crustal faults in the Azores may result from repeated dike intrusions over time[31,86,87].

Integrating our geodetically modelled dike intrusion, the relocated seismicity to one side of the dike, and our knowledge of the fault structure on São Jorge, we suggest that pre-existing faults can have opposing effects on magma propagation. The PdC-FZ facilitated rapid vertical melt ascent from the upper mantle, but may also have caused magma to stall by allowing lateral devolatilisation and melt escape, thereby increasing viscosity and reducing pressure within the main intrusion. The Azores thus offer valuable insights into the interplay between magmatism and seismic cycles.

## Methods

### GNSS data and analysis

The daily positions of each station were estimated using the GipsyX software package[88], employing the precise point positioning (PPP) strategy[89]. This approach enables independent computation of station positions by relying on fixed satellite orbit and clock parameters provided by the Jet Propulsion Laboratory (JPL). To ensure alignment with the latest realisation of the International Terrestrial Reference System, the ITRF2020 reference frame, daily transformation parameters estimated by JPL were applied. For earlier solutions for which ITRF2020 transformation parameters were unavailable, existing ITRF2014 solutions were converted to ITRF2020 using global parameters provided by the International Terrestrial Reference Frame service (https://itrf.ign.fr/).

The long-term motion of each GNSS station was analysed using the Hector software package[90], specifically designed for time series analysis of geodetic data. We modelled the trajectory of each station as a combination of a linear trend (secular motion), seasonal signals (annual and semi-annual variations), and step offsets caused by geophysical or instrumental changes.

Outlier detection and removal were performed as an initial step using an automated approach[91]. To ensure realistic uncertainty estimates for the derived parameters, Hector accounts for temporal correlations in the GNSS time series. A noise model comprising a power-law component and white noise was employed, as it has been shown to accurately represent the noise characteristics of geodetic data. The software estimates not only the linear velocities but also the amplitudes and phase lags of seasonal signals.

Offsets in the data, arising from events such as equipment changes, antenna replacements, or geophysical phenomena, were systematically incorporated into the model to ensure accurate motion estimates for all stations. To quantify the deformation caused by the dike intrusion, we computed offsets to characterise the sudden displacements observed at the start of the swarm on 19 March 2022. To isolate these displacements, 1 week of GNSS data around this epoch was excluded from the analysis for all stations. This exclusion minimised the influence of short-term noise and enabled precise computation of the deformation signals associated with the dike intrusion.

### Satellite radar interferometry (InSAR) processing

InSAR displacement measurements are obtained by analysing phase differences between radar images acquired by the Sentinel-1A satellite, which are then utilised to generate displacement maps, also known as interferograms[92]. At São Jorge, we processed descending (track orbit number 082) and ascending (track orbit number 002) interferograms using the Hybrid Pluggable Processing Pipeline[93]. We analysed all Sentinel-1A data from 2022-01-01 to 2022-07-31, approximately 3 months before and after the onset of the unrest. Sentinel-1A images were processed at a multilook factor of ten pixels in range and two pixels in azimuth, resulting in a pixel spacing of about 40 m. Topography phase contributions were removed using the COPERNICUS GLO-30 Global Digital Elevation Model (https://portal.opentopography.org/raster?opentopoID=OTSDEM.032021.4326.3). Residual differential phase interferograms were spatially filtered using a Goldstein filter with 0.6 strength to reduce the impact of decorrelation during phase unwrapping. Additionally, to minimise negative impacts during phase unwrapping, pixels over the sea were masked to zero using a fine-resolution coastline. Unwrapping was performed using a minimum-cost flow algorithm. All interferograms were cropped to a common area and georeferenced to the UTM zone 26S coordinate system.

### Seismicity data and analysis

There are two permanent seismic networks in the Azores (codes: PM, CP), operated by IPMA[94] and CIVISA[95], respectively, with just three stations on São Jorge before the start of the 2022 unrest. On 23 March 2022, just four days after the start of the seismic swarm, 15 temporary short-period seismic stations were installed (code: 3K)[96]. Three temporary broadband stations were installed 4 days later (code: PM). An additional ten broadband stations were installed on São Jorge and adjacent islands in June 2022 (code: 8S)[97]. Finally, at the end of August 2022, we deployed six short-period ocean-bottom seismometers (OBSs) offshore São Jorge (code: 4U)[98]. In this study, we analyse available seismic data until 18 November 2022, covering the first eight months of the seismic swarm. A timeline of stations is shown in Supplementary Fig. 12.

We developed a fully automated workflow to detect, pick, and associate seismic arrivals. For most of the study period, we used the PhaseNet convolutional neural network model[99] implemented in SeisBench[100], to detect and pick P- and S-wave arrivals. The association step was performed using PyOcto[101]. For the OBS deployment period, we found that an existing neural network trained on OBSs[102] yielded numerous false picks, leading to spurious events. Therefore, for this latter period, we used QuakeMigrate[103], which back-projects characteristic functions of the seismic waveforms to a coherent source, thus implicitly associating picked arrival times with events.

This workflow yielded 18,049 well-located events (maximum azimuthal gap ≤220°; ≥8 arrival times) relocated within an initial velocity model based on Rayleigh-wave ellipticities recorded at PM.ROSA in the west of São Jorge[47] (Supplementary Fig. 18). We generated 5000 random realisations of this starting model, perturbing velocities (up to 2 km/s) and layer thicknesses (ensuring a minimum thickness of 0.5 km and maximum thickness of 4 km), and jointly inverting for velocity structure and hypocentres[104–106]. We chose the best-fitting output model as our preferred, minimum 1D velocity structure (Supplementary Fig. 18).

Initial locations using our inverted 1D velocity model (Supplementary Fig. 18) reveal a highly clustered swarm; however, the relatively large uncertainty in the absolute hypocentre positions (median depth uncertainty of 430 m) indicates that greater precision is needed.

Moreover, the evolving station density during the 2022 São Jorge seismic swarm, with sparse station coverage for the first few days, necessitates a joint earthquake relocation approach that couples station corrections and relative locations between the later and earlier periods. We therefore used a new, multi-scale, high-precision method, NLL-SSST-coherence[107–110], which combines source-specific station travel-time corrections (SSSTs) to account for 3D velocity heterogeneity with the stacking of hypocentre probability density functions based on inter-event waveform coherence to improve small-scale relative location. We computed inter-event waveform coherency using a 2–40 Hz bandpass filter.

The workflow of NLL-SSST-coherence comprises a two-step process. First, SSSTs to our best-fitting 1D velocity model (Supplementary Fig. 18) are developed iteratively over collapsing length scales, using well-constrained events with at least 15 arrival times. Next, assuming that highly similar waveforms for two events recorded at the same station imply the events are nearly co-located, we measure inter-event waveform coherency at multiple stations as a weight to combine and stack the location probability density functions from the NLL-SSST relocations. This approach effectively reduces stochastic noise in travel-time data, thereby improving the precision of target-event locations. The practical advantages of this approach are that it requires waveforms from only a few stations and is computationally faster than other higher-precision relative location algorithms, such as hypoDD and GrowClust, because differential arrival-time measurements need not be computed. NLL-SSST-coherence yields high location precision across multiple scales, leading to a different interpretation of seismicity patterns than with methods that focus on fine-scale precision[107]. NLL-SSST-coherence has been applied to volcano-tectonic seismicity in other areas[108].

We computed a new magnitude scale including station corrections with observed amplitudes of the São Jorge seismicity using least-squares inversion[111]. This computation yielded the following ML equation:

$$M_L = \log(A) + 0.656 \log(r/17) + 0.00948 \log(r - 17) + 2 + C$$

where $A$ is the Wood-Anderson simulated horizontal-component amplitude in mm, $r$ is the hypocentral distance in km, and $C$ represents inverted station corrections.

We also computed $M_w$ using $S_H$-wave spectra[112,113]. We found that compared with $P$-wave spectra, $M_w$ estimates from $S_H$-wave spectra yielded far more observations per station (Supplementary Fig. 14), especially at lower magnitudes, resulting in network-averaged $M_w$ values with higher stability. The comparison between $M_w$ and $M_L$ is shown in Supplementary Fig. 15 ($M_w = 0.63\, M_L + 0.91$).

Caution is needed when interpreting $b$-values from frequency-magnitude distributions using local magnitude ($M_L$) scales of low-magnitude seismic sequences[114–119]. These studies show that moment magnitude ($M_w$) gives more accurate b-value estimates. We computed $b$-values for both $M_w$ and $M_L$, using magnitude of completeness ($M_c$) values for each one, approximated using the boundary-value-stability method[119–122]. Our results show a similar discrepancy between $M_w$ and $M_L$ (Supplementary Fig. 13). Therefore, we prefer to use $b$-values from moment magnitudes.

We computed focal mechanisms using Bayesian inversion of first-motion polarity data[123]. All solutions were best fit by double-couple mechanisms. An example solution is shown in Supplementary Fig. 16.

## Mechanical modelling of surface-displacement observations

The surface displacements observed by the GNSS network and Sentinel-1A interferograms show evidence of a major island axis elongated dilatation source in the central-west area of São Jorge. This pattern is characteristic of magmatic intrusions, typically a nearly vertical dike. To quantify the dimensions and magma volume associated with this intrusive pattern, we utilised a Bayesian 3D mixed boundary element method (BEM) inversion[124]. The model assumes that the island edifice and upper crust are homogeneous, isotropic and behave elastically with a Young's modulus of 5 GPa and a Poisson's ratio of 0.25. The 3D BEM method is a flexible numerical approach accounting for topography/bathymetry and non-planar dilatant cracks (e.g., quadrangles). The weights of the three types of observations were normalised in the inversion. This weighting approach prevents biasing the inversion toward the denser InSAR data, ensuring that each dataset contributes equally to the model solution[125].

To account for a realistic topography, we created a blended topography and bathymetry model at 100 m spatial resolution, combining the GLO-30 COPERNICUS model for the island's topography and the 2023 version of the EMODnet bathymetry model for the Azores (EMODnet DTM 2022).

We explore models of planar quadrangular intrusions that allow tilted (inward or outward) crack boundaries along depth. First, the inversion explores nonlinear geometric parameters, followed by solving for the opening using linear inversion to satisfy constant overpressure. Note that a single magma overpressure parameter governs the dike-opening pattern. This physically realistic assumption yields spatially smooth opening distributions that are independent of the chosen crack-mesh discretisation and avoids the need to specify a linear-inversion regularisation smoothing parameter (typical for kinematic inversions).

The inversion iteratively explores a large set of model parameters describing the dike intrusion position, orientation and dimensions by generating forward models optimised by a neighbourhood inversion algorithm[126,127]. See Table S1 for a full description of these geometric parameters. These models minimise the misfit with the observed displacements between 15 and 21 March 2022. To carry out the inversions efficiently, the spatial resolution of the surface-displacement maps was reduced using a quadtree partition approach, with a minimum average quadrant size of eight pixels and a maximum of 128 pixels. To subdivide the quadrants, two conditions were specified: displacement variance and displacement thresholds, both of 0.03 m. These parameters yield 698 and 702 points for the descending and ascending interferograms, respectively. We used all three components of the GNSS vectors. During inversion, the weights of the three observation types were normalised to prevent a single dataset from dominating the results.

Convergence of the inversion was assessed by monitoring the evolution of model misfit and the spread of accepted solutions. The inversion was terminated after 250 iterations, when the model cost fell below 750 and was stable for the last 100 iterations, indicating a stable inversion (Supplementary Figs. 9 and 22). The posterior distributions of model parameters (Supplementary Figs. 7–S9) show well-constrained values for dike strike, dip, and opening, with broader uncertainty in depth extent. This convergence behaviour supports the robustness of the inferred quadrangular intrusion geometry.

We also explored a wide range of other models, and the results are detailed in Text S1.

To explore the scenario of a laterally migrating dike, we forward-modelled intrusion scenarios across the seismicity region using a range of dike-opening models, with random (20,000 samples) geometries/sizes, and parameters drawn from the solution space of the geodetic inversion. We found that most models produce a surface-displacement signal that would be resolvable using the GNSS network on São Jorge or InSAR data, but we do not observe this (Supplementary Fig. 21). Only very small intrusions, with <100 to 1000 m³ of magma, would have been missed by GNSS (<1 mm) and InSAR (<1 cm). Therefore, we can likely rule out laterally propagating dike fingers hundreds of metres to ~2 km wide in depth.

## Tremor detection

To detect tremor-like bursts of seismicity, we used the covseisnet software package[128,129]. This method exploits the coherence of tremor signals across the network, which is computed as an array covariance matrix. Long-duration sources of coherent seismic energy are given by a low spectral width value of the array covariance matrix (Fig. 5b). We pre-processed vertical component data from permanent seismic stations (PM.PMAN, PM.ROSA, PM.SRBC, PM.PID, PM.HOR, PM.PCED, PM.PGRA, PM.PICO, PM.PCAN, PM.PPNO, CP.PAMA and PM.PSCM) with spectral whitening in 50-s windows followed by a temporal normalisation to minimise the impact of large, individual earthquakes. We average the network covariance matrix over 20-min windows, with a 50% overlap. In each window, cross-correlation functions are filtered in the 0.5–9 Hz band of interest. Then, a grid search determines the set of inter-station time delays that maximise the stacked correlation function envelopes at zero lag. To determine the tremor source locations (Fig. 5c), we back-project these cross-correlation time shifts using the S-wave source-specific travel-time models derived from the fine-scale earthquake relocation workflow.

## Autocorrelation imaging

The ACF approach has been successfully used to image temporal variations in subsurface structure beneath volcanic regions[130–132]. The phase autocorrelation method is not biased by high-amplitude features such as earthquakes and thus does not require pre-processing (e.g., time-domain normalisation and spectral whitening) to remove these features[133–135]. We tested different frequency bands and found that the 1.0–3.0 Hz band gave the most stable results, and is lower than most of the radiation from events of the seismic swarm. We also tested three different stacking windows, with and without overlap, and obtained consistent results. The ACFs were computed as 3-day linear stacks with a 2-day overlap.

## Data availability

Seismic waveform data from the PM network[94] are available from the IPMA FDSN webservices client at https://ceida.ipma.pt/. Waveform data from temporary networks 8S[97] and 4U[98] are archived at the EarthScope Data Management Centre (http://ds.iris.edu).

Our seismicity catalogue and 1D layered velocity model are available from a Zenodo repository[136].

The Sentinel-1A IW SLCs and satellite orbits files used in this study are provided by the European Space Agency (ESA). Files are publicly available through the Alaska Satellite Facility (ASF) Data Search Vertex (https://search.asf.alaska.edu). Bathymetry data around the Azores from the EMODnet Digital Bathymetry[137], while topography on land was obtained from the COPERNICUS GLO-30 Global Digital Elevation Model (https://portal.opentopography.org/raster?opentopoID=OTSDEM.032021.4326.3).

The GNSS daily files for the stations used in this study are available at the Portuguese GNSS National Repository managed by the Collaboratory for Geosciences (https://glass.c4g-pt.eu/), maintained with the support of EPOS (European Plate Observing System) (https://www.epos-eu.org/). The analysed stations belong to different networks: QEMD, HORT, PIED, AZTP, AZGR, and TERC are part of the REPRAA network (https://repraa.azores.gov.pt/); ENAO is part of the IGS network (https://igs.org); and VLAZ and PAGU are part of the C4G network (https://glass.c4g-pt.eu/).

## Code availability

Geodetic inverse models were performed using the DefVolc software, available online at http://opgc.fr/defvolc. Displacements, strains and stresses from triangular dislocations were computed using cutde (https://github.com/tbenthompson/cutde). Movie 1 was produced using the Sparrow software within the Pyrocko package.

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

## Acknowledgements

This work was supported by a Natural Environment Research Council (NERC) Urgency Grant (NE/X006298/1 to A.M.G.F., R.S.R. and N.C.M.), the European Research Council (101001601 to A.M.G.F.), a NERC Independent Research Fellowship (UKRI184; NE/B000184/1 to S.P.H.), the Portuguese FCT (PTDC/CTA-GEO/2083/2021 [https://doi.org/10.54499/PTDC/CTA-GEO/2083/2021] to R.F., G.S., J.F., N.A.D., P.J.G., R.S.R., M.M., and A.M.G.F.; PTDC/CTA-GEF/6674/2020 [https://doi.org/10.54499/PTDC/CTA-GEF/6674/2020] to S.C., G.S., N.A.D., V.M. and F.C.; UIDB/04683/2020). This work was also funded by FCT, I.P./MCTES through national funds (PIDDAC); (LA/P/0068/2020 [https://doi.org/10.54499/LA/P/0068/2020]; UID/50019/2025 [https://doi.org/10.54499/UID/50019/2025]; UID/PRR/50019/2025- [https://doi.org/10.54499/UID/PRR/50019/2025]; UID/PRR2/50019/2025 to S.C., G.S., R.F., N.A.D., V.M, and M.M.), and Spanish Agencia Estatal de Investigación (MICIU), VolcaMotion project (proyecto PID2022-139159NB-I00 financiado por MICIU/AEI/10.13039/501100011033 y por FEDER, UE, to P.J.G.). A.P. acknowledges the CEEC Institutional contract funded by FCT (https://doi.org/10.54499/CEECINST/00024/2021/CP2780/CT0003). We are grateful to the UK Ocean Bottom Instrument Consortium (OBIC)[138] (https://obs.ac.uk) and SEIS-UK[139] (https://seis-uk.le.ac.uk/; Loan: GEF-1153) teams for providing the instrumentation and installation services. The Portuguese Navy (Marinha Portuguesa) is acknowledged for providing critical support during the deployment and recovery of the OBS network; we particularly thank the captains and crews of the NRPs *António Enes* and *Sines*. The Azores Government, through its Fundo Regional para a Ciência, is also acknowledged for its financial support to harbour operations during the deployment and recovery of the OBS network. We thank Carlos Corela, Gonçalo Henriques, Paula Lourinho, Filipe Porteiro, and Octávio Melo from OKEANOS, University of the Azores, Sandra Sequeira from Observatório Príncipe Alberto do Mónaco (IPMA) in Faial, José Silva (Serviço de Ambiente e Ação Climática de São Jorge/Parque Natural de São Jorge), Miguel Mendonça (Municipality of Angra do Heroísmo), and Câmara Municipal das Velas, for crucial help and logistics support with the seismic deployments. We thank the CAD team from CIVISA for collecting and processing seismic data. We thank Luis Matias (IDL and the University of Lisbon) and Thomas Boulesteix (IPNA-CSIC) for fruitful discussions.

## Author contributions

Conceptualisation: S.P.H., P.J.G., A.M.G.F., R.S.R. and N.C.M. Methodology: S.P.H., P.J.G., A.L., A.M.G.F. and R.S.R. Software: S.P.H., P.J.G., A.L. and G.S. Validation: S.P.H., P.J.G. and G.S. Formal analysis: S.P.H., P.J.G., A.L. and R.S.R. Investigation: S.P.H., P.J.G., A.L., A.M.G.F., R.S.R., N.C.M., S.C., G.S., N.A.D., J.F., R.F., A.P., M.T., V.M., R.S., W.S., A.M., F.C., R.M., M.M., G.P. and A.M.G. Data curation: S.P.H., A.M.G.F., N.A.D., R.F., M.T., V.M., R.S., A.P., W.S., A.M., F.C. and R.M. Writing—original draft: S.P.H., P.J.G., A.M.G.F., R.S.R. and N.C.M. Writing—review and editing: S.P.H., P.J.G., A.L., A.M.G.F., R.S.R., N.C.M., G.S., N.A.D., J.F., R.F., S.C., V.M., A.P., W.S., R.M., G.P. and R.M. Visualisation: S.P.H., P.J.G., A.L., R.S.R. and G.S. Supervision: A.M.G.F. Project administration: S.P.H., P.J.G. and A.M.G.F. Funding acquisition: A.M.G.F., R.S.R., N.C.M., R.F., P.J.G., R.M., S.C. and N.A.D.

## Competing interests

The authors declare no competing interests.
