## [Transparent Peer Review file · Nature Communications]

Fault-mediated magma propagation and triggered seismicity revealed by the 2022 São Jorge Azores unrest

Corresponding Author: Dr Stephen Hicks

Version 0:

Reviewer comments:

Reviewer #1

(Remarks to the Author)
Dear Editor, Dear authors,

Please find enclosed my review of the manuscript NCOMMS-25-06043-T entitled "Leaky faults modulated magma ascent and seismicity during the 2022 São Jorge (Azores) volcanic unrest" by Hicks, Gonzalez et al.

I have divided this review into 2 main parts: Overall impressions and minor details.

I hope my review contributes to improving your great manuscript.

Kind regards,

Summary

This manuscript analyzes the São Jorge volcanic seismic crisis using a wide range of geophysical observables. On the one hand, seismic data from both land-based and OBS stations are analyzed using deep-learning techniques (EQTransformer). This approach yields nearly 12000 localized earthquakes with an average uncertainty of 40 m, due to the application of NLL-SSST-Coherence. The seismic analysis is further complemented by a frequency-magnitude study, focal mechanisms, and calculation of Auto-Correlation Functions (ACFs).

On the other hand, the processing of data from permanent GNSS stations allows the authors to obtain time series, which, in combination with InSAR images, leads to the modeling of a 25 km deep dike extending from the top mantle to the island's basement to the top of the mantle, aligned with the main axis of the island.

Based on these results, the authors interpret this process as a failed eruption, in which magma ascended approximately 25 km within a matter of days—mostly aseismically—before stalling at a depth of 1.6 km below the surface. This stagnation is attributed mainly to the island's compressive stress and a lateral devolatilization, which reduced the magma's buoyancy. The study highlights the interaction of magma ascents and fault structures which can influence the migration and evolution of unrest processes in tectonic environments.

Overall Impressions

This manuscript presents a highly interesting dataset within a multidisciplinary framework, making a valuable contribution to the field and worthy of publication. However, in my opinion, several aspects require clarification, particularly regarding the interpretation of the results and the study's limitations.

My main concerns with the manuscript lie in the interpretation of the phenomena and the limitations of the results, which in turn, affect the robustness of the interpretation.

Limitations:

Throughout the manuscript, the authors provide a great analysis of both seismological and geodetic data. One of the study's strongest aspects is the remarkable dataset they are working with—not only in comparison to previous studies on this seismovolcanic unrest (which includes data from more than 80 seismic stations, including OBS), but also in terms of the impressive multidisciplinary approach employed (the combination of InSAR-GNSS data)

However, I did not find any discussion of potential biases or limitations in their analysis and/or interpretation, which, in my opinion, is a significant drawback.

For example, while EQTransformer is a powerful tool for automatically analyzing seismic sequences, cultural, ambient noise or huge seismic activity can limit its detectability, potentially obscuring many earthquakes, particularly in dense seismic swarms. Therefore, many earthquakes during the first hours of the seismic unrest (when the seismic activity was denser) may be left undetected / analyzed. This concern is reinforced by the high completeness magnitude reported by the authors (~2.4), which is quite high. In this regard, for the same seismic unrest, Suárez et al. (2023) reported a migration from 10 km to 4 km during the early phase of the sequence on March 19 (from 18:00 to 01:00). However, due to the low magnitudes and overlapping earthquakes, they stated that thousands of events occurred, many of which remained undetected or were not located. Could the limitations or biases of the method, combined with the overlapping seismicity, have limited the detection of this hypocenter migration toward the surface? Please discuss this aspect further.

According to Figure 1A, the CIVISA network includes at least 2 additional stations located near the proposed dike. However, the manuscript only presents ACFs for two stations: PM.PMAN (Figure 4.b in the main text) and PM.ROSA (Figure S13). The changes in PM.PMAN are from 6-12s and from 18-23s but also from 16-18s. These changes seem to be comparable with changes in late-May and mid-June at the same time-lag ranges? What made them different from the others? Quantitative information needs to be provided to support this reasoning.

PM.ROSA plot is limited to a time window of 16–24 seconds and displays a different x-axis scale compared to Figure 4.b, making it difficult to draw any clear conclusions. Could the authors clarify why the ACFs from the PM.ROSA station do not show the same clear changes as those from PM.PMAN? What limitations might be influencing these observations? Perhaps including a dv/v plot could offer additional insights and better relate these changes to the interpretations discussed by the authors.

In my opinion, the results derived from the ACFs alone do not provide sufficient evidence to support the authors' statement (2.3 Subsurface velocity changes – First Paragraph) “indicating velocity reductions at <10 km and >15 km depth, respectively,” given that these findings are based on data from just one station out of the 83 used in this study. This is particularly concerning when the authors main interpretation involves a simultaneous dike opening from 26 km to almost 1 km depth. The structural changes inferred by the authors from these ACFs require additional supporting evidence. Perhaps including ACFs, from other stations within the CIVISA network, (e.g., Figure 1b) near the proposed dike location could offer further support for the near-field interpretation (which I agree with). Similarly, if the structural changes observed at deeper levels (26 km) are real, they should be detectable at any station on the island.

Regarding the GNSS results, I may have missed it, but I did not find any mention of the uncertainties. The authors state that no significant inflation preceded the magma intrusion, as they did not observe any vertical deformation. However, earlier studies of the seismic unrest indicated that the QEMD station detected a vertical displacement of 4 cm (Suárez et al., 2023) between March 16 and 18-19, which might be consistent with the 6 cm of deformation reported by the authors via InSAR in this manuscript. Are there any limitations or biases, either in the data itself or in the processing tools applied, that may have prevented the detection of this inflation? The authors must provide some discussion on this regard, specifically on their uncertainties.

Finally, the mechanical modeling of the surface assumes a quadrangular intrusion to fit the dike to the GNSS and InSAR observations. I'm not sure why this geometry is the only one explored, or what are the geophysical / geological constraints considered to determine the upper and lower limits on the space-parameter (i.e. length, shear, top depth, top length, etc.) used to obtain the dike model.

As a result, the authors model a 26 km long intrusion as a single layer, which fits well with the horizontal component of QEMD, showing minimal residuals. However, the horizontal residuals at the GNSS stations to the south indicate non-random direction (all of them points to the east), suggesting that the model could not properly fit the observations. This raises the following concerns: Why do the residuals show this trend? Is it possible that multiple sources of deformation are contributing to a more complex, overlapping deformation that your model does not account for (e.g., deep inflation/deflation combined with a shallower dike intrusion)? Did you include vertical deformation during the modeling process? If not, how could this omission affect your results? These questions directly tie into the next concern related to the interpretation of the phenomena.

Discussion:

The author's interpretation of the volcano-tectonic unrest is mainly based on the results derived from the dike modelling, which, in my opinion, is the major drawback of the manuscript. As I mentioned earlier the authors do not provide any discussion on the limitations of their model. It appears that their interpretation of the process is mainly grounded on the best fitting model despite the possible biases or limitations, even alternative hypothesis.

As far as I understand, and as the authors state in the Methods section (Section 4 – "Mechanical modeling of surface displacement observations"), "...the model assumes that the volcanic island edifice and upper crust are homogeneous and isotropic." While this is a common approach when modeling the crust, their Vp velocity model shows a significant contrast between the physical properties of the crust and the upper mantle (a difference of more than 4 km/s from the surface to 10 km depth). The strong variations observed in the 1D velocity model are consistent with earlier studies on the island's petrology (Zanon et al., 2020 and 2023), which identify distinct transition zones, with the Moho boundary near 10-12 km depth and a second transition around 25 km depth. Furthermore, Supplementary Material Figures S11 and S12 illustrate typical shapes of two overlapping probability distributions in terms of "shear" and "length".

Additionally, despite the authors' statement in Section 2.1 ("Geodetic observations and dike-opening model") that "...no significant deformation occurred before or after the seismicity (Figs. 3a and S1 & S2)," two distinct displacement trends can be clearly observed in the supplementary Figs. S1 and S2, even in Fig. 3a. First, a rapid deformation within a day or two, followed by a slower deformation that can be clearly seen in all the horizontal components of the GNSS stations, lasting more than three weeks.

Therefore, the resulting model, based on the "best-fit model parameters," provides an excellent mathematical fit to the observed data. However, it raises several concerns regarding the feasibility of the geophysical model.

This evidence, along with the results from earlier studies (for example, the ones that the authors reference in the main text), led me to suggest that the results might indicate a more complex deformation pattern than a simple quadrangular plane (i.e., two overlapping sources of deformation, such as a shallow dike and one or multiple deep reservoirs inflating/deflating, or any other combination). This is a common behavior observed in volcanic systems and oceanic volcanic islands, which has been documented in multiple unrest processes across different geodynamic scenarios, as the authors reference in the manuscript (Hudson et al., 2017), or as suggested by petrological studies in the archipelago (Zanon et al., 2020, 2023). Numerous examples can be found in previous studies, but one clear and recent example cited by the authors is the La Palma eruption, which also exhibited a migration of deep reservoirs (~20-30 km depth) years-to-months prior to the volcanotectonic unrest (Torres et al., 2019). In this case, the process was also dominated by regional-to-local stress (Del Fresno et al., 2022). Why did the authors exclude or avoid considering other possible and well-established models for this type of unrest?

Therefore, the authors should discuss (or explore) alternative scenarios/models (which have been entirely overlooked in the manuscript) that could also explain these results. They must also properly describe the limitations and biases associated with their model, as it forms the cornerstone of their interpretation. In my opinion, this discussion and a thorough revision of the proposed model and/or interpretation are necessary.

Finally, despite my concerns related to the dike modelling, about the seismic activity at ~12 km depth, I have multiple concerns:

The authors suggest that a near-vertical dike opens simultaneously from 26 km depth below sea level up to 1.6 km below the surface, with practically no seismic activity during the ascent. As I suggested earlier, a limitation on the methodology might be hindering seismic activity. It is quite difficult to imagine a magma body with enough energy to open a dike quasi-simultaneously from 26 km depth to the surface, facilitated by a "mantle-rooted" structure (PdC-FZ), reaching the basement of the volcanic edifice, stagnating almost aseismically, and yet the same structure being compressed enough to release energy due to lateral volatile transfer. Additionally, if PdC-FZ is a mantle-rooted structure, why is the basement of the dike shifted to the north and not aligned with this structure?

In this model, the authors also state that "...magma stalled ~1,600 m below the surface, near the estimated base of the island edifice," and later attribute the seismic activity to "thermal pressurization and magma devolatilization," which increased its viscosity. However, if magma crossed the upper mantle and crust in less than a day, with enough devolatilization to generate such high seismic activity (including dozens of felt earthquakes), and stalled at these shallow depths, I would expect gas emissions. Even nowadays, some fingerprints of this unrest may still be detectable. Is there any geochemical evidence or reports supporting your interpretation (e.g., groundwater temperature increases, isotopic variations, or diffuse CO₂ emissions)?

At approximately 12 km depth, in the vicinity of an aseismic area, the authors state that there is no evidence of magma presence or injection (a claim with which I am not entirely convinced) and the seismic activity is primarily driven by thermal pressure and lateral fluid migration. What mechanism could prevent triggering during the dike-opening process yet lead to seismic events afterward due to fluid transfer? How do the authors explain the presence of this ~2 km wide aseismic area? Please provide some reasoning for this observation.

Details

This is a minor suggestion that could improve readability: I recommend changing all references to 'seismic unrest' to 'volcano-tectonic unrest,' given that your manuscript focuses on tectonic and volcanic interactions

First, I suggest minor changes / clarifications to the main text and, secondly, to the figures in the main text and supplementary material.

INTRODUCTION:

(Second Paragraph – "... in ocean island setting where seismograph coverage is limited ...") I suggest changing seismograph coverage for seismic network.

RESULTS:

Section 2.1 - Geodetic observations and dike-opening model

(First Paragraph) This section only focuses on the horizontal GNSS components, add some statements on your results of the vertical ones, and the uncertainties.

(First Paragraph - "...no significant deformation occurred before or after the seismicity") What do you mean by "after the seismicity", as I can relate, the seismic activity remains months after the first burst of activity. Maybe referring to certain dates this sentence is clearer.

(Second Paragraph - "All deformations detected by InSAR occurred between 15 and 21 March 2022 Figs. S3-S5, ...") Could you provide the unwrapped InSAR images in this supplementary figure, it would help to the non-InSAR readers specialist to better understand your great results!

Section 2.2 - Precursor seismicity

(First Paragraph – "We find deep clusters of ML...") Which ML equation are you using? Please provide a reference.

(First Paragraph – "The seven weeks immediately before the onset of the seismic swarm was aseismic") Add "based on our results".

Section 2.2 - Seismic Swarm

(Second Paragraph) This paragraph is difficult to interpret without a weekly/monthly histogram of activity. The authors properly describe the location evolution, but the reader does not know "when" and "how many".

(Second Paragraph – "All events after 19.2022 lie between 6-14 depth, most concentrated at 9-14 (Fig.4)") I suggest adding "near to the Moho discontinuity in the Island (Ferreira et al., (2020))"

(Second Paragraph – "...extending upwards to ~6km depth from the eastern edges of UL and LL (labelled Shallow, "SH) ..."). I suggest changing SH for any other label, as SL, to avoid confusion with SHmax in Fig. 4

Section 2.3 - Subsurface velocity changes

(First Paragraph – "...São Jorge, PM.PMAN and PM.ROSA (Fig 4a.) ...") change to PM.PMAN (Fig 4a.) and PM.ROSA (Fig. S13).

(First Paragraph – ... show coherent arrivals at lag times of ~2-24 s) Fig. 4b only displays from 6-24 seconds and Fig. S13, from 16 to 24. Also, figure S13 shows a different timespan (x axis) from Fig. 4b. Please unify and correct where necessary.

DISCUSSION:

(Third Paragraph – "...from 1-26 km depth bsl ..."). Change to "from 26 to 1 km depth bsl"

METHODS:

Section 2 – Frequency magnitude relationships:

(First paragraph- “We found that compared to P-wave spectra, Mw estimates from SH-wave spectra had smaller uncertainties”) If able, add a plot to the supplementary material to support this statement

DATA AND CODE AVAILABILITY:

The Zenodo repository is private, so I cannot access to check the author's results. If possible, please provide the configuration parameters used to analyze the seismic data to help the readers to reproduce your results.

Figures:

Figure 1:

Fig. 1-a panel:

Change “Elevation / bathymetry” for “Bathymetry / elevation” to be consistent with color in the legend.

Remove the 500 and 1500 ticks in elevation legend to be consistent with the scale in bathymetry.

Add a distance scale and arrow pointing to the north.

Fig. 1-b panel:

Due to the color palette, the seismic network is practically indistinguishable from the background, therefore I suggest using a grey palette to the elevation

Add a distance scale and arrow pointing to the north.

Fig. 1-c panel:

If able, providing the unwrapped InSAR image in the supplementary material could help the readers to interpret your results

Figure 2:

In my opinion, this figure could be either removed or moved to the supplementary material.

Figure 3:

Fig. 3-a panel:

Missing uncertainties in the plot. Please plot them. Why is so much missing data from December to February?

Fig. 3-b panel:

The depth cannot be appreciated by this color scale, I suggest adding a subpanel showing the Tp-Ts in ROSA and/or PMAN or any other station in the island, along with the hypocenter-color scale in the plot.

A histogram could help the readers to understand the evolution of the seismic activity through the lapse analyzed

The vertical dashed line seems to be duplicated

Fig. 3-c panel:

A legend to explain the colors of this subplot is needed.

Please highlight the main contributions of this panel with some squares or arrows.

Due to the color palette, the label and the vertical magenta line cannot be appreciated

The y-axis values are from 6-24 s, but in the main text says 2-24 s, please modify accordingly the plot and/or the text (2.3 Subsurface Velocity Changes- First Paragraph: "...from May to September 2021 show coherent arrivals at lag times of ~2-24 s")

Highlight the main changes of this plot, to non-seismologist readers it could be difficult to interpret.

Figure 4:

As a common comment on this plot, please, unify the seismic color palettes. It's quite confusing to see how the seismic activity has 3 different color palettes, Fig. 4a referencing the depth, 4b and 4c the date but with different scales (up to 180 days and up to 200 days from 2023-03-19). Apart from confusing, it does not help the reader with its interpretation. This is a great plot, maybe with some clarifications it could be easier to interpret to the readers

Fig. 4-a panel:

Add a distance scale and arrow pointing to the north.

As I pointed out in Fig. 3-b panel, the color palette used to the topography make the plot quite confusing. As the authors are showing the altimetry as contour lines every 250 m, maybe the green background could be removed.

As shown in Fig. 1-b panel, maybe the QEMD deformation can be added to this plot.

In the main text the authors describe at least 150 deep earthquakes, why can only be seen a dozen of them?

Fig. 4-b panel:

In my opinion, the "VEN arrow" does not provide any information, at first sight I thought it was some deformation data.

Please align the Vp major ticks with the distance grid, it is quite confusing with the current display.

Fig. 4-c and 4-d panels:

I suggest changing the near-surface earthquakes "SH" label to "SL", may be confusing, as long as, you are plotting in 4-a panel the "Regional SHmax".

Figure 5:

Fig 5-a panel:

From Fig. 4-a, it seems to me, like some seismicity is missing. If you are only plotting those seismic activity clustered, maybe you can add the "unclustered" seismicity with grey dots in the background?

Fig 5-b,c and d panels:

I suggest changing the order, Cluster 4 on the left, Cluster 19 on the right, to be consistent with their real location.

If the legend expands from 0 to more than 150 days in Fig. 5-b, why the x-axis in Fig, 5-c only expand to 140?

In Fig. 5-c, does the first location (star) compare with affects to this plot? Because I see a slight migration in Cluster 19; but seeing cluster 4, it looks like if I change the position of this first point maybe some migration could be seen.

In Fig.5-d, maybe change the waveform colors from clusters 4, and 19 to brown and green as they are shown in Fig.5-a and Fig.5-c

Figure 6:

Fig. 6-a:

I really like this plot; it is super intuitive!

Fig. 6-b:

Move it to the supplementary material? To me it does not fit with the model in the same figure

Fig. S1:

Please align the y-major ticks (and the grid) with the daily displacements in each panel. It will help to highlight the displacements.

Fig. S2:

Same as figure S1.

Fig. S3:

Please provide a color-scale to the values of the interferograms. Comparing this plot with the Fig. 1c, it seems like the color-scale has different values, please unify them.

Fig. S4:

Same as Fig S3. Additionally, if unwrapped InSAR images could be provided will be helpful, with an scale in [cm].

Fig. S8.

May stations codes are overlapped, please fix it.

Fig. S10:

Same as Fig S3

Fig. S11:

It seems as the quality of the image is too low, please provide it with higher resolution.

Fig. S12:

Same as Fig S11

Fig. S13:

As I stated before, this plot is inconsistent with the plot in the main text Fig. 3c, please unify them

Reviewer #2

(Remarks to the Author)

The paper is a detailed analysis of the seismicity and deformation of a failed eruption that took place in 2022 in São Jorge island, Azores, an island with fairly rare eruptions as the previous one took place in 1808. The event is well documented by onshore and offshore seismic recordings from permanent and temporary networks, including OBS which improved events locations. At the time of the intrusion, nine permanent GNSS stations were operating on São Jorge and the neighbouring islands, and these data were completed by InSAR acquisitions, from ascending and descending orbits of the Sentinel-1 satellite. Seismic and displacement data are analysed with up to date methods, showing that a dike stalling at 1.6 km depths was responsible for the unrest. From magnitudes of EQ in swarms west of the dike, the b-value is computed. Hypocentres are relocated, the velocity model is improved, and temporal variations in velocity structure are imaged using autocorrelation. Analysis show that both the seismic swarm and the dike are located along a well-know fault (Pico do Carvao, PdC-FZ). These most remarkable feature shown is that Fault Plane Solutions (FPS) on seismic events that took place during and after the dike emplacement are consistent with left lateral fault displacement t , when paleo-seismology, long term GPS solutions, and the tectonics indicate that long term slip along the fault is right lateral. The discussion concludes that the left lateral focal mechanisms are related to the dike opening, despite the right lateral long term slip. The low-b value of the post-intrusion seismic swarm, and well as the large seismic moment relative to the intruded volume are interpreted as consequences of the percolation of magmatic fluids released by magma devolatilation. This devolatilation is assumed to have increased the viscosity and made the dike stall at depth. The paper addresses a well documented and rare event, with relevant methods. It is clearly and logically organized with relevant references. However, I have a few major concerns regarding the main conclusions, which are not discussed thoroughly enough to be fully convincing. Mainly, reason for the inconsistency of the focal mechanism with the long term slip is not sufficiently explained and, secondly evidences, for the fluid percolation being responsible for the swarm to the west, the increased viscosity and the failed eruption are too slim. Below are suggestions of elements to discuss in more details in order to make the conclusions more robust.

Major comment

- My major concern is the lack of discussion around the left lateral mechanism of seismic events in the swarms west of the main dike, when the tectonics indicates that the sense of slip is right lateral. How can the authors explain that the dike intrusion induced a Coulomb Stress Changes (CSC) large enough to overcome the tectonic loading? Does it mean that this tectonic loading was null at the level of the swarm because it had been released by previous diking events west of the present dike intrusion? Is this possible regarding slip accumulated since the previous dike intrusion west of the dike? It would be convincing to provide some orders of magnitude of this long term slip, and compare them to those potentially induced by the 2022 event.
 - It is indicated that the PdC-FZ was probably more pre-stressed than the P-FZ. Which magmatic or tectonic event could have pre-stressed the PdC-FZ ?
 - What would the CSC be on the conjugate fault indicated by the focal mechanism ?
 - How do the authors explain that no event took place east of the dike, in a region of increased Coulomb stress as evidenced by Fig. S14?
 - At the bottom of page 13, it is indicated that "the dike increased stress on PdC-FZ for both left and right lateral modes", but only CSC on left lateral faults are shown in Fig. S14. To support this assertion, CSC for right lateral slip should also be shown.
 - I also have concerns about the statement that "the seismic swarm released only Mw 4.7, leaving the likelihood of a future large earthquake uncertain". This is very vague. Is the likelihood of a future large earthquake increased or decreased ? I believe that if the sentence concerns tectonics induced earthquakes consistent with the right lateral sense of slip, this likelihood should be decreased, but this should be more clearly indicated.
- Considering that the dike intruded oblique to the minimum principal stress, coeval lateral displacement should have taken place making the dike a sheared intrusion, otherwise the model is inconsistent with the conclusion. Could the displacement data fit be improved by considering such a mechanism? The GNSS residual in Fig. S9, show that shear displacements following a left lateral sense of slip could provide a better fit of the observation. I agree that such a slip direction is opposite to the tectonic slip direction, and it is surprising. It is also likely that when conducting an inversion considering a sheared intrusion, the other dike parameters (such as the strike) determined by the inversion will be different. This deserves to be further explored.
- My second major concern is about the seismic swarm and dike arrest triggered by fluids leaking at depth west of the dike. The occurrence of a seismic swarm west of the dike characterized by a low b-value, the high Mw magnitude relative to the dike volume, together with the intrusion not leading to an eruption support this hypothesis. This bundle of indications are in my opinion too weak to be part of the title and one of the main conclusions. I find it doubtful that magmatic fluids would migrate downward from the top of the dike to the LL and UL locations, instead of being released at the top of the dike which had previously formed. Therefore I would recommend being more prudent and I would keep this hypothesis as a likely scenario and not as a major conclusion :
 - As these seismic events are located beneath the emerged part of the island, I wonder whether gas release could be confirmed by field or remote sensing observations, such as those provided by measurements from satellites, similarly as evidences at Ambrym by Shreve et al., 2019? If not what could be the reason?
 - What could be the explanation for fluids not leaking from the top of the dike, but instead moving to reach LL and UL locations? fluids tend to move upward and not downward so the sequence of seismic events from SH to UL and LL, deeper

than the top of the dike is surprising.

- Is there any remote sensing or field observations of an increase in the volume of gases released at the top of the intruded dike? Gases tend to collect above intrusions (See for Instance Menand and Tait, 2002) and usually take the shortest pathway to leak in the atmosphere, so that the region at the top of the dike, also located along the PdC-FZ fault, is the most likely places for fluids to be released. If not, how can this be explained?
- Is the speed at which fluids migrate in the fault compatible with what is known of fluid migration along faults?
- An alternative hypothesis, for the LL, and UL swarms is that that magma first intruded the area of the SH swarm and then tried to intrude west of this dike in the area of swarms LL and UL, but was further arrested by the high normal stress inherited from a previous historical intrusion. The involved volume could be low enough not to induce any deformation (with volumes consistent with S15).
- Magma propagation is controlled by fracture toughness, magma overpressure, magma buoyancy and magma viscosity (Rivalta et al., 2015). Alternative mechanisms could explain that magma stalled at depth. For instance, the injected magma volume and hence the dike overpressure, or alternatively the buoyancy could be insufficient for further upward propagation.

Minor comments

- A timeline in the supplementary material, with the number of stations as a function of time would help tracking the seismicity coverage;
- Similarly, it would be useful to have the cumulated seismicity as a function of time in Fig. 3b, not only before the intrusion but also after;
- The boundary element method and inversion used for analysis of displacement data allows for curvatures. A model with an along dip curvature could connect the deep earthquake swarm to the fracture at shallower depth. Would such a model provide a better or as good fit of the displacement data, according to the Akaike Information Criteria?
- The inversion parameters should be described in a figure. Parameter "Depth" should perhaps be called "Height"?
- What is the convergence of the inversion solution? This indication is important to show readers how the solution has converged (spread of last computed models, whether iterations stopped because the standard deviated on computed solutions was low enough or because the maximum number of iterations decided at the beginning was reached).
- It would also indicate where the best fit solution is located with respect to the search limits.
- The plots showing the 1D and 2D PPD should be shown with better resolution.
- Considering that the inversion were conducted giving equal weight to the three data set, you could perhaps quote Dumont et al., 2021 who showed the relevance of such an approach.
- The film showing the migration of events is too long (more than five minutes). Perhaps snapshots at irregular time, along with the cumulated seismicity, would be more informative and could be joined in a plot. The colorcode for the events is missing and sometimes the colors reset to red, for an unknown reason.
- The order of the supplementary figures should follow the order of the main text. For instance, on page 4, before the seismicity distribution section, reference to the supplementary material jumps from S5 to S9. There are other such places where supplementary figures should be reordered;
- Page 13, line 7: "hydrothermal fluids or melt or volatile..". In this sentence a word (an "if") might be missing "hydrothermal fluids or **if** melt or volatile..".
- Page 13, line 9: before the end of the page "observed transtensional environments", a word is missing "observed **in** transtensional environments"

Figures

- On Fig. 4: In subplot a. the deep swarm is not visible enough (dots are too small and colorcode is different from Fig 4b).
- No date colorcode is indicated in fig 4a. In a, b, and c colorcodes for dates since 03-19-2022 are different. I suggest unifying colorscales of the different subplots for clarity.
- Fig S6, referred to before paragraph 2.3, does not seem to be the right one.

References

- Dumont, Q., Cayol, V., & Froger, J. L. (2021). Mitigating bias in inversion of InSAR data resulting from radar viewing geometries. *Geophysical Journal International*, 227(1), 483-495.
- Menand, T., & Tait, S. R. (2002). The propagation of a buoyant liquid-filled fissure from a source under constant pressure: An experimental approach. *Journal of Geophysical Research: Solid Earth*, 107(B11), ECV-16.
- Rivalta, E., Taisne, B., Bungler, A. P., & Katz, R. F. (2015). A review of mechanical models of dike propagation: Schools of thought, results and future directions. *Tectonophysics*, 638, 1-42.
- Shreve, T., Grandin, R., Boichu, M., Garaebiti, E., Moussallam, Y., Ballu, V., ... & Pelletier, B. (2019). From prodigious volcanic degassing to caldera subsidence and quiescence at Ambrym (Vanuatu): The influence of regional tectonics. *Scientific Reports*, 9(1), 18868.

Valerie Cayol

Reviewer #3

(Remarks to the Author)

The manuscript presents an analysis of the 2022 magma intrusion event on the island of São Jorge, within the Azores

archipelago. This event offers a valuable opportunity to advance our understanding of magmatic processes in tectonically active oceanic settings. By combining InSAR and GNSS ground deformation inversions with seismic analyses of earthquake distributions and source characteristics, the work provides an integrated perspective on both the magmatic and tectonic aspects of the event. I particularly like the analysis of the induced seismicity, and I agree that the patterns highlighted are not typical of seismicity directly induced by a dike. This is a very unusual and interesting observation.

A key point of the manuscript lies in the interpretation of the intrusion as a "leaky fault" phenomenon, with the proposed role of pre-existing fault zones in both enabling and contributing to the arrest of magma ascent. The manuscript is generally well-written and clear in its exposition. The language is precise, and the structure supports the logical flow of the argument. Overall, this is a valuable and promising manuscript that has the potential to make a significant contribution to our understanding of magma-tectonic interactions in rifted oceanic environments. However, there are two key issues that, if addressed, could significantly strengthen the clarity and impact of the conclusions:

1) Conceptual clarity regarding leaky faults and dikes: At several points, the manuscript appears to conflate or contrast the concepts of leaky faults and "regular" dikes without explicitly defining or distinguishing them. It would greatly benefit the manuscript and advance the debate in the general literature on magma intrusions to clearly articulate the theoretical and observational criteria that separate these two mechanisms and to frame the interpretation of the São Jorge event within this context. A more systematic discussion would help clarify whether the observations more strongly support a leaky fault model, a dike process, or some hybrid scenario such as a sheared intrusion.

2) Missing or incomplete supporting information: Some of the key data and analyses needed to evaluate the proposed mechanism are currently absent or insufficiently detailed in the text, figures, or tables. Addressing these gaps, as detailed below, will improve the transparency and reproducibility of the interpretations.

Major comments:

a) Conceptual clarity regarding leaky faults and dikes. Some differences between "regular" dikes and faults can be summarized in the following way:

"Regular" dikes are hydraulic fractures (Pollard, Gudmundsson, Anderson, Rubin, and many others):

- 1) Predominantly opening component on the dislocation modelling the deformation (i.e. displacement is roughly perpendicular to the dislocation wall);
- 2) Orientation roughly coincides with most compressive + intermediate stress axis, perpendicular to least compressive stress axis (usually close, ± 10 degrees) \rightarrow strike is parallel to strike of local normal faulting mechanisms
- 3) Dikes are usually formed without the need of any pre-existing structure, i.e. the rock is actively fractured at the intrusion tip due to the extremely intense stress concentration forming there due to the dilation of the intrusion walls.

Leaky faults (my take on this):

I understand it is a tectonic fault zone that allows magma or fluids to ascend through it, effectively leaking material from depth toward the surface or upper crust. I understand a leaky fault is not a dike, though it may host magma like a dike would. Rather, magma exploits a pre-existing fault or fracture, rather than creating its own new pathway (like a dike does). The intrusion occurs on the fault plane, rather than perpendicular to the least principal stress, which is typical for dikes.

I would expect:

- 1) Predominantly shearing component on the dislocation modelling the deformation (i.e. displacement is predominantly parallel to the dislocation wall). This is because usually faults have accumulated shear stress on them and as soon as they are lubricated by any fluid intruding them, they would also shear as well as open to accommodate the magma;
- 2) Orientation strikes between most compressive and least compressive stress axis (usually 30 to degrees) (Anderson's theory of faulting)
- 3) Slip usually occurs mostly on pre-existing faults or on pre-existing fault segments potentially linked by brief segments of pure opening.

Since the main point of the article is clarify the specific mechanism behind the intrusion, I suggest to be really clear about what is intended every time a dike is mentioned. For example, in the introduction, references 5 to 8 (as well as some others further below) on dike intrusions are listed as studies on magma leaking through pre-existing faults. These are incorrectly listed citations.

b) As for the São Jorge event, is the evidence supporting a "regular" dike or magma intruding a pre-existing fault (these two are very different concepts, as explained in point a, and cannot be conflated)? It is not very clear from the manuscript. The inversion results in the text suggest pure opening, as no shearing component of the displacement is mentioned anywhere (or at least I could not find it). The method used is declared "3D mixed BEM". Does this mean that a shearing component on the dislocations is allowed and used? If the argument is that of a leaky fault, this possibility should at the very least be explored. As it is now, the inversion seems to assume a purely opening dike, contradicting one of the main discussion arguments of the manuscript.

Moreover, the intrusion is parallel to the strike of the closest normal faulting events, it is therefore roughly perpendicular to the axis of least compression. The authors do not list any evidence of a vertical pre-existing fault rooted at 25 km depth (the fractures mapped on the island are all broken and can be hardly linked to a consistent major strike slip fault that could leak a fluid as viscous as magma). Based on the evidence presented, the "regular" dike interpretation would seem far more logical. I may have misunderstood some of the assumptions or evidence: please clarify.

c) Incomplete information:

- 1) In the ground deformation section the maximum opening is listed but not the average opening or, alternatively, the total volume. The total volume (or the average opening, which would allow me to infer the volume) is an important parameter and should be included. Was the pressure in the BEM model inverted for or fixed? This is not clear. Was a simple rectangular dislocation model used first to fit the data, or only the BEM model? I believe it is always useful to include the simplest model possible, beside the complex model.
- 2) Please add the vertical GNSS displacements to Fig. S9. Without them, I am not able to check the calculations nor it is possible to the reader to reproduce the results. In the caption of fig. S9, please add the time interval to which the displacements refer to (is it 15 to 21 March 2022 as mentioned in the main text at the bottom of pag. 18?). "We used all three component GNSS vectors" -> "We used all three components of the GNSS vectors"?
- 3) The resolution of some of the figures in the supplement is far too low to read them, please increase resolution.
- 4) The inversion procedure is described well but please include in the table caption a definition for the parameters inverted for. For example, what is Bot. length? The length of the dike quadrangle at its bottom? What is shear angle?? How much is the total volume?

d) Further comments:

- 1) The bottom of the intrusion is very deep. Please state what specific data, among those used, require this very unusual bottom depth. Please discuss how robust this is.
- 2) The southern GNSS horizontal displacements are not really well-fit, fig S9. Clearly the data are not sufficient for a strong discussion on this, but what is your interpretation of this misfit?

Good luck with your study.

Version 1:

Reviewer comments:

Reviewer #2

(Remarks to the Author)

think overall the authors have addressed the comment appropriately. Thank you for the very clear and detailed answer provided to all reviewers. The different points of the answer were numbered. I will address the answer to my comments in the order they were answered.

R2.1. The discussion on the left lateral mechanism of micro-seismicity in a context of right lateral long term slip is providing some more details. I agree, that we need to assume that the background stress is weak. The more convincing element is provided by the Coulomb stress changes.

R2.2. Thank you for clarifying the origin of the pre-stress on the PdC-FZ. However, I still do not understand why the pre-stress, as indicated in the introduction (l. 110-111) is for right lateral displacement, when failure took place in left lateral mode. For failure to take place for a given failure mechanism, the pre-stress has to be in the same direction. A possibility is that anterior slip (like that associated to a M 7 EQ) released background stress and that the Coulomb stress change induced by the dike, associated to the fluid release, was sufficient to lead to failure. This would mean that the recent seismic activity did not pre-load but unloaded the fault, if this previous failure was west of the dike. A Mohr circle, given in the appendix, would perhaps help understanding how fluids may have brought a weakly loaded fault closer to failure.

R2.3 : perfect answer.

R2.4 : perfect answer, but Fig S19 has replace S14.

R2.5 : Perfect.

R2.6 : I do not understand the answer. I think the fact that M4.7 EQ were released make the release of regional tectonically accumulated right-lateral stress less likely.

R2.7 : My comment on the possibility of coeval sheared intrusion to better fit distant E-W displacements is addressed, but it would have been more satisfactory to conduct the shear stress inversion with the same dike or a dike very close as the best-fit solution determined by the boundary element inversion (DefVolc) or by the dislocation solutions (GBIS). Using GBIS Model, a horizontal source, very different from the best fitting solution is determined. This indicates that the parameter space is probably too large and not sample enough. Consequently, the data fit is decreased with respect to both the simple rectangular solution (GBIS) or the more complex quadrangle (DefVolc). Because the discrepancy between the data and the DefVolc best fit model is small, it would have been better to search a best fit model in the vicinity of the model already determined, somehow using it as an a-priori for the search. This approach could have been conducted with both the GBIS or the DefVolc inversions.

R2.8 Leaky faults

- I think the new title is more appropriate, as the evidences in favor of a fluid leak are too weak.
- I fully agree with the new conclusion, moreover fluid migration from the dike provides an explanation for the micro EQ being triggered at low stress change (see reply to comment R2.2).
- I wonder if there is no error in the interpretation in Fig 7. Indeed Fig 4 shows a North dipping dike, when Fig. 7 shows a south dipping dike.

- The new text version and Fig. 8 comparing b-value in different context brings a factual element to the possible role of fluids in the seismicity trigger.
- Could you unify the cluster names ? There might not be a need for additional lineation names as lineations are indicated by clusters. In the text, clusters are called cluster 2, ..., 18. On figures 4 and 6, letters appear (LL, UL), as well as a mix of letters and numbers (Fig. 6). This is slightly confusing. Moreover the numbers (2, 7 and 18) described in the text are not found in any of these figures.

R2.9 and R2.12 : good addition, thank you.

R2.10 and R2.14: Good explanation.

R2.13 : Fig 6. There are inconsistencies between the numbering of the cluster in the figure, in the caption and in the text.

R2.14 (two answers are named R2.14 and R2.11 is missing) : I agree with the modifications. They improve clarity.

R2.15 : Great it was useful.

R2.16 : I do not see the new panel with the daily rate of seismicity in Fig 4.

R2.17 : Two dikes models have many more parameters (12 parameters + 4 position parameters = 16 parameters) than a single dike with a curvature (10 parameters). Hence, such models may lead to local minima, which might explain why the 2 dikes model did not give any satisfactory results. The parameter space might have been again too large, as two shallow sub-horizontal dikes were determined which fit the two deformation patterns, which is unlikely.

R2.18 to R2.27: perfect. Thank you.

Valerie Cayol

Reviewer #4

(Remarks to the Author)

I enjoyed reading the extensive paper of Stephen P. Hicks and Pablo J. Gonzalez et al. "Faults modulate magma propagation and triggered seismicity: the 2022 São Jorge (Azores) volcanic unrest". It is likely to further our understanding of magmatic and tectonic interaction, and it has an impressive multidisciplinary approach. The paper has already undergone insightful reviews by three reviewers, that have clearly helped improve the manuscript. I recommend it to be published in Nature Communications after some minor reviews.

Please find my comments and suggestions in the attached word file.
 Þorbjörg Ágústsdóttir

Reviewer #5

(Remarks to the Author)

I have just finished reading the paper, "Faults modulate magma propagation and triggered seismicity: the 2022 São Jorge (Azores) volcanic unrest," by Hicks and coauthors. The manuscript has already undergone its first round of revision, and as requested by the editor I have carefully reviewed this new version with particular focus on the comments from Referee#1. The manuscript addresses the seismic crisis and volcanic unrest on São Jorge Island (Azores) using an impressive multidisciplinary approach incorporating seismic and geodetic data. Following careful analysis, the authors propose a stealthy dike intrusion as the main driver of the seismic unrest. This work is outstanding for several reasons: 1) The impressive seismic catalog retrieved via state-of-the-art machine learning techniques, which allowed for a careful analysis of magma-fault interaction. 2) The number of geodetic models employed to constrain the deformation source, even despite sparse data coverage. 3) The proposed very fast dike intrusion that propagated within a fault zone and eventually stalled beneath the volcanic edifice—a rare case observed with modern instrumentation.

I have carefully reviewed the rebuttal letter, focusing especially on the excellent comments provided by Reviewer #1. I believe the authors have addressed all points in a punctual and satisfactory manner, including the required additional analysis. This has resulted in a much stronger manuscript compared to the previous version. The authors should be commended for the additional work performed to address in an excellent manner the referee#1 criticisms, which I agree with in the first place. As a result, I strongly support the consideration of this paper for publication in this journal.

Below I report a few thoughts I had while reading the manuscript. I hope those can be of any help to further strengthen the presented work. I noted some typos and errors, especially in the supplementary figures. For example, the caption for Fig. S19 describes a panel (d) that is not displayed. The authors should carefully proofread the entire manuscript and supplement. I also have a few minor comments concerning the interpretation of the results, particularly the dike-fault interaction and subsequent seismicity. The modeling component (seismic and geodetic) appears to be sufficiently and carefully handled/presented given the data availability. The authors have the choice whether (or not) to include these

suggestions in the manuscript, except for Comment number 1), which requires specific attention and must be addressed and mentioned within the manuscript.

1) Fig. 3c shows a clear difference of completeness magnitude throughout the seismic sequence and especially from the beginning of the seismic crises (~5 days) and later during the complex swarm activity. The b-value analysis (given the method used for the Gutenberg-Richter fit) results in a high $M_c \sim 2.5$. Now the problem is that the interpretation of early dynamics of seismicity can be biased from the fact that smaller events go undetected. Conversely small earthquakes are used to interpret later phases of the magma-fault dynamics and to discuss further intricacies in the development of the seismicity along the Pico do Carvao Fault zone.

I do not think that the lack of smaller events would change the overall picture depicted by the authors regarding the interpretation of the dike-earthquake interaction. However, I ask the authors to add one or two sentences in the discussion section reporting the absence of smaller earthquakes during the early phase of dike propagation/emplacement, noting that this may have hindered the detection of finer scale processes discussed for later phases.

2) Aseismic seismicity zone and left-lateral residuals in the GNSS data: I would suggest considering the hypothesis that the dike emplacement could have triggered aseismic slip (creep or slow slip) along the PdC-FZ, a possibility not considered/discussed by the authors. Creep or slow-slip transients induced by dike emplacement have been documented, (see for example Cattania et al., 2014 JGR <https://doi.org/10.1002/2016JB013722>, Xu et al., 2016 <https://doi.org/10.1002/2015JB012505>, and many examples of aseismic graben forming at dike intrusion during rifting episode). I know the authors attempted to consider models including a fault, but could it be that aseismic left-lateral slip is the cause and/or can explain the left-lateral residuals in your model and reversal of focal mechanisms?

3) The heterogeneities of the focal mechanisms (oblique e normal), and the across fault distribution of seismicity (from the video) seem to point out at the activation of a distributed fault/damage zone rather than a simple planar strike slip structure. In this case, the peculiar streak-like distribution of seismicity may come from a complexity of the fault system. Would it be this an additional possibility to add to the discussion to interpret the lineaments in Fig. 6? The complexity can be a complementary explanation of seismicity induced by magmatic/fluid fingers occurred after the dike emplacement in such a complex fault zone.

4) The dike arrest below the volcanic edifice/island should be primarily due to the increasing compression due to the edifice load on the dike upper tip approaching the surface in the first place. To emplace, 20 km dike intrusion in only few days, magma should be primitive (mafic and low viscosity) with little gas content. The intrusion is so fast that exsolution of volatiles should not be occurring in this short time span, while volatiles would start to exsolve only later after the dike get arrested and at its final length. Due to the compression of the edifice the magmatic fluid can escape along the fault zone rather than upward explaining the peculiar spatial-temporal distribution of seismicity (streaks) and the absence of degassing at the surface. The authors seem to give equal weight to the mechanical arrest due to edifice load and increase in magma viscosity. I just would like to remark that the two processes may occur on slightly different time scales.

5) Page 8, the sentence: "whose front is consistent with a fluid diffusivity of $\sim 0.001-0.002$ m²/s (Fig. 6b-c)." Here the authors report a very low value of diffusivity while in Fig.6c it is reported a much higher value.

6) Page 18 sentence: "The resulting static stress change led to the observed left-lateral seismogenic shear along the PdC-FZ, rotated by 90° with respect to the background right-lateral motion". Is this 90 or 180 degree rotation.

7) At Page 18 the sentence "We infer that because the dike intruded into a mature fault zone, few new fractures had to be created, leading to aseismic Mode-I opening, possibly accompanied by devolatilisation and shear dilatancy at the dike tip" reads odd. Dikes always open in Mode I, if the authors here refer to the dike-induced fractures opening in Mode I, then they need to clarify this concept.

Best Regards,
Luigi Passarelli

Version 2:

Reviewer comments:

Reviewer #2

(Remarks to the Author)

I think my comment were addressed appropriately. The different points of the answer were numbered. I will address the answer to my comments in the order they were answered.

R2.7 : Sheared intrusion or not ? I am not really sure of the direction of lateral slip that was taken into account. From the figure enclosed in the response, it seems to me that the addition of a lateral slip makes the solution worse, and corresponds to a right lateral displacement, when a left lateral movement is needed. From the posterior PPDs shown in the answer, it

seems to me that the parameter range correspond to 0-10m and not -10-10m. Could a wrong slip direction explain that the solution is not improved and the PPD is uniform ? I may be wrong and misunderstanding the outputs from GBIS.

R2.8 Leaky faults C: thank you for the clarification in the caption, the problem was indeed the perspective.

R2.17 : Perfect thank you ! The Akaike Information Criteria AIC are a good way to find the best compromise between fitting the data and having a large number of parameters. In your case you do not even need AIC, as more complexity degrades the best fit model.

As indicated in your paper, the Moho is determined from seismic velocities at a depth of 11.5-13 km, while the first clusters (D1, D2, D3) are in fact at depths of 33-25 kilometers, in the mantle. This is also the storage depths indicated by microthermometry (25.5 km Zanon et al., 2023).

This mantle source might be worth emphasizing as it is not that common.

Response to Reviewers

Reviewer 1

Overall comments

This manuscript analyzes the São Jorge volcanic seismic crisis using a wide range of geophysical observables. On the one hand, seismic data from both land-based and OBS stations are analyzed using deep-learning techniques (EQTransformer). This approach yields nearly 12000 localized earthquakes with an average uncertainty of 40 m, due to the application of NLL-SSST-Coherence. The seismic analysis is further complemented by a frequency-magnitude study, focal mechanisms, and calculation of Auto-Correlation Functions (ACFs).

On the other hand, the processing of data from permanent GNSS stations allows the authors to obtain time series, which, in combination with InSAR images, leads to the modeling of a 25 km deep dike extending from the top mantle to the island's basement to the top of the mantle, aligned with the main axis of the island.

Based on these results, the authors interpret this process as a failed eruption, in which magma ascended approximately 25 km within a matter of days—mostly aseismically—before stalling at a depth of 1.6 km below the surface. This stagnation is attributed mainly to the island's compressive stress and a lateral devolatilization, which reduced the magma's buoyancy. The study highlights the interaction of magma ascents and fault structures which can influence the migration and evolution of unrest processes in tectonic environments.

This manuscript presents a highly interesting dataset within a multidisciplinary framework, making a valuable contribution to the field and worthy of publication. However, in my opinion, several aspects require clarification, particularly regarding the interpretation of the results and the study's limitations.

My main concerns with the manuscript lie in the interpretation of the phenomena and the limitations of the results, which in turn, affect the robustness of the interpretation.

Throughout the manuscript, the authors provide a great analysis of both seismological and geodetic data. One of the study's strongest aspects is the remarkable dataset they are working with—not only in comparison to previous studies on this seismovolcanic unrest (which includes data from more than 80 seismic stations, including OBS), but also in terms of the impressive multidisciplinary approach employed (the combination of InSAR-GNSS data)

However, I did not find any discussion of potential biases or limitations in their analysis and/or interpretation, which, in my opinion, is a significant drawback.

The author's interpretation of the volcano-tectonic unrest is mainly based on the results derived from the dike modelling, which, in my opinion, is the major drawback of the manuscript. As I mentioned earlier the authors do not provide any discussion on the limitations of their model. It appears that their interpretation of the process is mainly grounded on the best fitting model despite the possible biases or limitations, even alternative hypothesis. Therefore, the resulting model, based on the "best-fit model parameters," provides an excellent mathematical fit to the observed data. However, it raises several concerns regarding the feasibility of the geophysical model.

Therefore, the authors should discuss (or explore) alternative scenarios/models (which have been entirely overlooked in the manuscript) that could also explain these results. They must also properly describe the limitations and biases associated with their model, as it forms the cornerstone of their interpretation. In my opinion, this discussion and a thorough revision of the proposed model and/or interpretation are necessary.

General author response

We truly appreciate the time Reviewer 1 has given to comment thoroughly on our manuscript, which has improved our study. We do, however, disagree with their comment that “[the] interpretation of the volcano-tectonic unrest is mainly based on the results derived from the dike modelling, which, in my opinion, is the major drawback of the manuscript”. Given that a previous study (Suarez et al., 2024) uses the seismicity and geodetic data to present a conceptual model of the cause of the unrest, we feel that a more formal, quantified, and physical model of the dike intrusion is much needed. Although the seismicity patterns (e.g., streaks) and their space-time migration are fascinating, they cannot be easily interpreted without the framework of a broader geodetic-based model. We would also like to emphasise that our model and interpretation are not based solely on a single best-fitting model. Rather, we employed a Bayesian inversion framework that explores the full parameter space and provides a probabilistic assessment of model parameters. This approach allows us to quantify uncertainties and avoid over-reliance on a single deterministic solution. We admit that these data uncertainties, model uncertainties, and exploration of the model space were not effectively presented in the previous, short-format manuscript, so we have now taken great care to present these issues openly (see e.g., new supplementary texts; for details, see individual responses below). We believe these additions strengthen the manuscript by providing a more balanced interpretation and by transparently communicating the assumptions and constraints of our modelling.

R1-1	Reviewer comment	While EQTransformer is a powerful tool for automatically analyzing seismic sequences, cultural, ambient noise or huge seismic activity can limit its detectability, potentially obscuring many earthquakes, particularly in dense seismic swarms. Therefore, many earthquakes during the first hours of the seismic unrest (when the seismic activity was denser) may be left undetected / analyzed. This concern is reinforced by the high completeness magnitude reported by the authors (~2.4), which is quite high. In this regard, for the same seismic unrest, Suárez et al. (2023) reported a migration from 10 km to 4 km during the early phase of the sequence on March 19 (from 18:00 to 01:00). However, due to the low magnitudes and overlapping earthquakes, they stated that thousands of events occurred, many of which remained undetected or were not located. Could the limitations or biases of the method, combined with the overlapping seismicity, have limited the detection of this hypocenter migration toward the surface? Please discuss this aspect further.
-------------	-------------------------	---

Author response	Thank you for this detailed and important comment. Since our original manuscript, we have re-picked the data using a lower probability threshold in the machine learning picker, and this has resulted in ~twice as many events (~18,000) compared to the previous version of our catalogue. First of all, we would like to say that we also reported the shallow, upward migrating seismicity at the onset of the seismic swarm in our original manuscript, but we perhaps didn't focus on it enough. Therefore, we have made a new figure, Fig. 4, which shows the temporal evolution of this early seismicity to greater detail. With our new catalogue, we find 120 locatable events during the first 9 hours of the swarm. For the same time period and depth range, Suarez et al. (2024) find 101 events with an RMS residual < 0.2 s. Therefore, the two catalogues, at least for this period of time, are comparable, but overall, our catalogue has nearly four times as many high-precision relocated events as Suarez et al. (2024). We also agree that many events could not be located. In particular, visual analysis of waveforms shows that the seismic swarm possibly began a few hours before the first event in our catalogue, with an increasing rate of 'drumbeat'-like activity, but the arrival times of these events could only be picked on a maximum of just one or two stations. Another advance we have made since submitting the original manuscript is to detect tremor-like bursts. We used a network covariance matrix analysis to detect and locate the tremor. We find two bursts of tremor at the onset of seismicity and another ~8 hours later (new Fig. 4a). The onset of high-rate, drumbeat-like activity at the start of the swarm can then be regarded as volcanic tremor. We found no tremor in the days-to-weeks-to-months before the start of the swarm. We located the tremor and found that it matches well the eastward, upward and westward downward migrating seismicity associated with the dike intrusion (Fig. 4c-d). It is also worth noting that our earthquake catalogue uses arrival times and waveforms from CIVISA stations, which Suarez et al. (2024) did not have access to. This network includes an extra permanent station on São Jorge, PM.PAMA, which was located very close to the seismic swarm. Regarding magnitudes and completeness magnitude, we would like to emphasise that we are choosing moment magnitude as our preferred scale, rather than local magnitude. We find that the b-value using M_w is much higher (2.4) than with M_L (1.5); see new Fig S13, even with a locally calibrated M_L scale. Similarly, the completeness magnitude (M_c) is higher with M_w (2.6) than with M_L (2.3). Our M_L b-value and M_c are similar to those of Suarez et al. (2024), who only analysed M_L in their catalogue. Therefore, our preferred M_c using M_w cannot be seen as worse-performing than, or directly compared to, Suarez et al. (2024); instead, we believe the b-value computed using M_w is more robust. Overall, therefore, we feel that our catalogue is by no means incomplete or biased, given the data we have available. The machine learning picker performs well in our study. As mentioned above, even though there are some events that occurred before the first locatable event, these could not be located, even with manual analyses. Moreover, our seismic phase associator, PyOcto, has been shown to perform well at times of very high seismicity rates (Münchmeyer et al., 2024, doi: 10.26443/seismica.v3i1.1130). We find that 67% of $M_w \geq 2.0$ events occurred during the first two weeks of the seismic swarm (Fig. 3), when few temporary stations were installed (Fig. S12). Overall, along the lines of what the reviewer says, the M_c is likely due to several factors: the highest rate of seismicity and large earthquakes occurring during the early part of the seismic swarm, the temporally evolving station coverage, the background noise level on a small ocean island, and the attenuation characteristics of the subsurface.
---

	Changes made	 • We have updated the manuscript and figures based on our catalogue of ~18,000 events. • We have updated the main text to describe in more detail the migration of seismicity during the first few hours of the seismic swarm (Section 2.2), and we have updated the interpretation of dike ascent as a result (Section 3). • We have added new Fig. 4, which shows in detail the spatial-temporal evolution of seismicity at the onset of the swarm. • We have updated and included extra supplementary plots (Figs. S14 & S15) comparing our two magnitude estimates. • We have added the following text to Section 2.2: "We find many earthquake-like signals starting several hours earlier, and accelerating in rate, although they are too small to be located".
R1-2	Reviewer comment	According to Figure 1A, the CIVISA network includes at least two additional stations located near the proposed dike. However, the manuscript only presents ACFs for two stations: PM.PMAN (Figure 4.b in the main text) and PM.ROSA (Figure S13). The changes in PM.PMAN are from 6-12s and from 18-23s, but also from 16-18s. These changes seem to be comparable with changes in late-May and mid-June at the same time-lag ranges? What made them different from the others? Quantitative information needs to be provided to support this reasoning.
	Author response	Many thanks for this comment. Actually, station CP.PAMA is the only permanent station on São Jorge from the CIVISA that was operational during the onset of the seismic crisis. Station CP.PSS01 is a temporary station that was installed a few days later. Please see our new Fig. S12 that shows a station timeline, which was specifically requested in R2-15. CP.PAMA is a noisy analogue station, and we were unable to convert recorded counts to realistic ground motion units. Nevertheless, we still tried to compute ACFs, but they contained numerous artefacts (see figure below; the y axis shows Julian days in 2022). Therefore, we excluded CP.PAMA from our formal ACF analysis in the manuscript.  PAMAZ_2022_3days_1.0-3.0-Linear  We have reanalysed the ACFs from PM.PMAN at the time periods stated by the reviewer (late May and mid-June), but we do not see any obvious change in coherence or time lag shift at these times; certainly nothing on the order of the changes that we see on 19 March 2022.
	Changes made	We realise that there were some minor discrepancies in the text when we quoted lag times (as noted by R1-23 below), and we have now fixed these.

R1-3	Reviewer comment	PM.ROSA plot is limited to a time window of 16–24 seconds and displays a different x-axis scale compared to Figure 4.b, making it difficult to draw any clear conclusions. Could the authors clarify why the ACFs from the PM.ROSA station do not show the same clear changes as those from PM.PMAN? What limitations might be influencing these observations? Perhaps including a dv/v plot could offer additional insights and better relate these changes to the interpretations discussed by the authors.
	Author response	Many thanks for noticing that the PM.ROSA plot covered different dates and lag times - good catch. The sensitivity of each station to subsurface changes can be estimated with a Fresnel zone argument. If we compute the first fresnel zone radius for a reflector at a depth of 20 km (as indicated by the 16-24 s lag times), a corresponding P-wave velocity (since ACFs from vertical component data give a P-wave reflectivity response) at this depth from our 1-D velocity model of 8.0 km/s (Fig. 5), and at a frequency of 2 Hz (the middle of the bandpass used in our ACF analysis), we estimate a radius of 6.3 km. The plot below shows this radius around PM.PMAN and PM.ROSA, compared with the location of the modelled dike:  We can see that the first Fresnel zone for PMAN overlaps with the modelled dike, especially at its deeper end. ROSA, on the other hand, just misses out on being sensitive to the dike. Therefore, we believe the position of ROSA relative to the dike is the main factor limiting a positive observation of a temporal ACF change at the onset of the seismo-volcanic unrest. As requested by the reviewer, we also computed dv/v changes using the MSNoise workflow (Lecocq et al., 2014, doi: 10.1785/0220130073). We computed dv/v for both 2021 to get a baseline and for 2022, for both PMAN and ROSA. We find that the dv/v are more complicated and harder to explain than the ACF results. Even though for lag times of 2.0-26.0 s, PMAN shows a $\sim 0.3\%$ reduction in dv/v at the onset of the seismic swarm:  There are similar orders of magnitude changes at other times in the year, and in 2021: 
ROSA also shows similarly complex changes:

We therefore speculate that these dv/v results might be overly sensitive to shallow subsurface changes, and possibly therefore related to e.g. rainfall or hydrogeological factors. Therefore, we find that these dv/v results require further dedicated analysis along with other seismic imaging (i.e., local earthquake tomography; coda interferometry) to better understand the subsurface structure and its temporal variation during the 2022 dike intrusion.

Changes made

We have fixed the lag time and date scale in the PM.ROSA plot, Fig. S17.

We have added the following sentence to Section 2.3: *“These two stations are ~16 km apart, and given the waveform frequency and depth of subsurface changes, we infer that PM.PMAN sensitivity kernel includes the geodetically-imaged main dike, whereas PM.ROSA is too far west of the dike (Fig. 4).”*

R1-4	Reviewer comment	In my opinion, the results derived from the ACFs alone do not provide sufficient evidence to support the authors' statement (2.3 Subsurface velocity changes – First Paragraph) “indicating velocity reductions at <10 km and >15 km depth, respectively,” given that these findings are based on data from just one station out of the 83 used in this study. This is particularly concerning when the author's main interpretation involves a simultaneous dike opening from 26 km to almost 1 km depth. The structural changes inferred by the authors from these ACFs require additional supporting evidence. Perhaps including ACFs, from other stations within the CIVISA network, (e.g., Figure 1b) near the proposed dike location could offer further support for the near-field interpretation (which I agree with). Similarly, if the structural changes observed at deeper levels (26 km) are real, they should be detectable at any station on the island.
	Author response	We thank the reviewer for this comment, but we disagree that the ACFs “do not provide sufficient evidence” for several reasons. First, there were only three permanent stations on São Jorge that captured the onset of the seismic crisis and main dike intrusion. We carried out ACF analysis for stations PM.PMAN and PM.ROSA, but were unable to yield stable ACFs at CP.PAMA (see our response to R1-2 above). Therefore, our ACF analysis uses the best and only data available at the time. Even though the reviewer mentions the total combined network of 83 stations, most of these stations were installed well after the onset of the seismic swarm and the main dike intrusion (see our new Fig. S12 for a station timeline). Based on our Fresnel zone arguments, it is understandable that PM.ROSA did not record any subsurface changes associated with the main dike (see our response to R1-3 above). ACF analysis is a robust single-station method, and we have only interpreted substantial changes relative to our baseline analysis from data in 2021. Given the timing of the ACFs changes correlates with the dike intrusion, we believe it is a worthy observation that indicates changes to the subsurface over a wide range of depths. Nevertheless, we understand the reviewer’s concern about the interpretation of simultaneous opening in the crust and mantle, which we have somewhat tempered and revised, particularly in light of other reviewer comments (R1-9 & R3-4) and other reanalyses of the geodetic data and the uncertainties/limitations of our dike model.
	Changes made	We have written the following paragraph in the Discussion: “Although required by the geodetic data and ACF results, a near-vertical dike opening in a matter of days from 26 km depth up to 1.6 km appears to be an unusual aspect of this event. However, magmatic intrusions propagating rapidly at ~0.1-1.0 m/s⁵⁹⁻⁶², with partial or total aseismicity especially at mantle depths, is not without precedence^{57,59,63}. We infer that because the dike intruded into a mature fault zone, no new fractures had to be created, leading to aseismic Mode-I opening, possibly accompanied by devolatilisation and shear dilatancy at the dike tip^{62,63}. Alternatively, the longer-term GNSS deformation transients at far-field stations (e.g., AZTP on Pico, ENAO, Graciosa, and to a lesser extent QEMD on Sao Jorge; Figs. S1-S2) may indicate that the shallower and deeper parts of the modelled dike acted deformed on different timescales. In this case, we speculate whether the mantle segment deformed more slowly due to different mechanical properties in the mantle. Such a depth segmentation may be consistent with the ACF imaging results, which imply a bimodal depth distribution in subsurface changes, with clear changes at lag times <8 s and >18 s, but without any obvious change at intermediate depths. However, we cannot formally quantify such a hypothesis with the available time resolution of geodetic observations. Regardless of the exact behaviour at depth, such a stealthy and rapid magma ascent presents a challenge for tracking dike intrusions and forecasting their eruptive potential.”

R1-5	Reviewer comment	Regarding the GNSS results, I may have missed it, but I did not find any mention of the uncertainties. The authors state that no significant inflation preceded the magma intrusion, as they did not observe any vertical deformation. However, earlier studies of the seismic unrest indicated that the QEMD station detected a vertical displacement of 4 cm (Suárez et al., 2023) between March 16 and 18-19, which might be consistent with the 6 cm of deformation reported by the authors via InSAR in this manuscript. Are there any limitations or biases, either in the data itself or in the processing tools applied, that may have prevented the detection of this inflation? The authors must provide some discussion on this regard, specifically on their uncertainties.
	Author response	We would like to clarify that the QEMD station is not located in the area of InSAR deformation. The InSAR uplift signal of ~6 cm is located east-south-east of QEMD, along the saddle zone of minimal InSAR. Station VLAZ station, further to the northwest, also shows no significant vertical motion during this period. However, the uplift was followed by a nearly symmetric subsidence of similar magnitude, yielding a net-zero vertical displacement, which could explain why it was not captured in the interferogram. Second, we note that the ~3 cm uplift at QEMD occurred between March 16–18/19, but no equivalent signal was recorded in the nearby VLAZ station. We stress that Suarez et al.'s model is conceptual, with no formal quantification or inversion of data. We therefore carried out some forward modelling of such a proposed source (see new Text S2 and Fig. S33) and found that such a localised and short-lived transient is inconsistent with a deep or laterally extensive magmatic source. Instead, the signal may reflect a shallow (< 5km depth), unidentified localized process rather than widespread inflation precursory to the dyke upwards propagation. However, we believe that most likely the ~3 cm uplift signal reported by Suarez et al. (2024) might come from unmodelled tropospheric signals and with more optimistic uncertainties compared to our solutions. Figure 4e-g of their paper shows other signals with a similar amplitude at other times, which are left uninterpreted. Similar potentially spurious signals are also seen on the horizontal components of their filtered time series. We computed our original GNSS solutions independently on a station-by-station basis. Any correlation (i.e., common mode) can only be due to the orbits and not due to the processing, so one station does not affect others due to the processing. To more robustly check the presence of this signal, we compare the GNSS time series processed by an independent, network-based method, which minimises common-mode noise effects, and does not show any precursory signal (Fig. S2).
	Changes made	 • We have now included error bars to the GNSS solutions in Figs. 3a & S1-S2. • We have written an extensive appraisal of the conceptual model of Suarez et al. (2024) in Text S2, which includes an analysis of the GNSS signals and a forward model test. • We have added the following text to the end of the first paragraph of Section 2.1: “Previous analysis of GNSS signals suggests a 2-day-long, up to 4 cm, precursory uplift signal at QEMD starting on 2022-03-16. However, given vertical-component noise levels and solution daily uncertainties, we cannot confidently observe such a signal in our Precise Point Positioning (PPP) solution. The previously reported solutions show multiple similar-sized anomalies at other times, which were left uninterpreted. We do not see a precursory signal in an independent network-based processing solution (Text S2; Fig. S2). Furthermore, we observe no such signal at the nearby GNSS station VLAZ (Fig. S2). We conducted forward modelling of a sill source matching the 4-cm of uplift with minor horizontal motion (Text S2). Such a magmatic source, if located at the depth of the locus of intense seismicity, would have instead produced cm-scale signals at VLAZ. Therefore, we consider that such rapid transient signals are likely GNSS processing artefacts; moreover, if the precursory uplift signal at QEMD were real, it is too small, uncertain, and localised to be quantified and modelled.”

R1-6	Reviewer comment	Finally, the mechanical modeling of the surface assumes a quadrangular intrusion to fit the dike to the GNSS and InSAR observations. I'm not sure why this geometry is the only one explored, or what are the geophysical / geological constraints considered to determine the upper and lower limits on the space-parameter (i.e. length, shear, top depth, top length, etc.) used to obtain the dike model. As a result, the authors model a 26 km long intrusion as a single layer, which fits well with the horizontal component of QEMD, showing minimal residuals. However, the horizontal residuals at the GNSS stations to the south indicate non-random direction (all of them points to the east), suggesting that the model could not properly fit the observations. This raises the following concerns: Why do the residuals show this trend? Is it possible that multiple sources of deformation are contributing to a more complex, overlapping deformation that your model does not account for (e.g., deep inflation/deflation combined with a shallower dike intrusion)? Did you include vertical deformation during the modeling process? If not, how could this omission affect your results? These questions directly tie into the next concern related to the interpretation of the phenomena.
	Author response	(1) "Finally, the mechanical modeling of the surface assumes a quadrangular intrusion to fit the dike to the GNSS and InSAR observations. I'm not sure why this geometry is the only one explored..." We appreciate this observation. While the original manuscript presented only the final model - a tilted quadrangular dike - we want to clarify that we conducted an extensive exploration of the model space in the early phases of the study. This included testing a wide range of source geometries, such as:  - Single and multiple vertical and inclined dikes - Spherical and ellipsoidal pressure sources - Faults with both right- and left-lateral shear - Combinations of the above (e.g., dike + fault, dike + spherical source) None of these alternative configurations provided a satisfactory fit to both GNSS and InSAR data (both ascending and descending tracks). The sub-vertical dike model was the only one that reconciled all datasets with minimal residuals and physically plausible parameters. (2) "...what are the geophysical / geological constraints considered to determine the upper and lower limits on the space-parameter..." To avoid biasing our conclusions, during the inversion, the dyke parameter bounds were not constrained to be narrow values (see the actual range for each parameter in Table S1). However, we used parameter ranges informed by regional geological and geophysical knowledge, including: a) the known depth range of the brittle-ductile transition in the area (volcanics on top of oceanic crust with seismologically constrained moho depths), b) dike intrusions and their geometries as from geological mapping of São Jorge (orientation - strike angle); c) surface deformation patterns observed above dyke elsewhere (i.e., typical two lobes of uplift and perpendicular horizontal displacements separated by a saddle of contracting and relative subsidence above the dyke intrusion); d) the spatial distribution and sensitivity of the GNSS and InSAR datasets. These constraints were used to define realistic priors for model parameters such as top depth, length, dip, and opening. (3) "...the authors model a 26 km long intrusion as a single layer..." We acknowledge the simplification inherent in modeling the intrusion as a single planar feature. However, given the sparse GNSS coverage and the need to balance model complexity with data resolution, a single-source model is more parsimonious, reasonable, and supported by the geodetic observations. The pattern of displacements is incompatible with an axisymmetric source (e.g., spherical or ellipsoidal sources), which will cause an outward radial pattern that is not observed in the pre-, co-, or post-seismic unrest periods. A deep dike source is needed to fit the far-field displacements, while the shallow top-depth fits the InSAR data and GNSS stations on Sao Jorge. However, the down-dip width of the dike is one of the most uncertain parameters. (4) "...horizontal residuals at the GNSS stations to the south indicate non-random

		direction..." We agree that the eastward trend in residuals at southern GNSS stations is noteworthy. This pattern may indeed suggest additional deformation sources not captured by the current model. We considered this possibility and tested models incorporating deeper inflation/deflation and slow-slip fault sources in combination with the dike. However, these models did not improve the fit significantly and introduced trade-offs in other areas (e.g., increased residuals in InSAR or reversing the observed near-field GNSS stations). This feature remains unexplained with the current observational dataset. (5) "Did you include vertical deformation during the modeling process?" Yes, vertical deformation was included in the modeling and inversion process. Both GNSS vertical components and InSAR line-of-sight displacements were used to constrain the model. However, due to the lower signal-to-noise ratio in vertical GNSS data, the horizontal components had the strongest influence on the inversion outcome. Also, lack of notable vertical inflation or deflation (see our answer to R1-5 above, and text S2) shows that there were no substantially large spherical/ellipsoidal pressure change sources. Please note that the co-dyking subsidence signal (~4 mm at QEMD) is well-predicted by our preferred dyke model (see the attached figure showing observed, modelled and residual GNSS vertical displacements). We acknowledge that we overlooked the formal reporting of the extensive tests carried out during the 2 years of manuscript preparation. We are very confident that our preferred quadrangular model is the most sensible and parsimonious of all the models tested. None of the alternative models has a better misfit and complexity. However, as with any geophysical inversion, we can only confirm that the proposed model is consistent with most of the observations, despite the small but noticeable residuals affecting the three GNSS stations in Faial and Pico islands.
	Changes made	 • We have added the following paragraph to Section 2.1: "To test the robustness of our a priori quadrangle geometry parameterisation, we conducted an extensive exploration of the model space. These tests included testing a wide range of source geometries, such as single and multiple vertical, inclined, and curved dikes, spherical and ellipsoidal pressure sources, faults with both right- and left-lateral shear, and combinations of the above (Text S1). None of these alternative configurations provided an improved fit to both GNSS and InSAR data (Table S2.1) while remaining physically plausible. Our sub-vertical quadrangular dike model emerged as the preferred solution, as it is the only configuration that reconciles all datasets with small residuals and physically plausible parameters." • We have added a detailed discussion of our experiments to test many additional models raised by the reviewer - please see Text S1 and associated supplementary figures. • We have added an extra panel to Fig. S10 showing vertical fits. • We have included a paragraph in the Discussion explaining the shear residual: "Another remaining issue is that predicted horizontal GNSS vectors from our preferred quadrangular dike model at stations on islands to the south (AZTP and PIED on Pico; HORT on Faial) show a consistent eastward trend (Fig. S10), which could indicate a left-lateral shear source. This trend is particularly fascinating given the rotated left-lateral slip of the swarm seismicity. Therefore, we inverted the geodetic data for a sheared dike model (Text S1.1), but found that the resulting model unsatisfactorily increases the GNSS and InSAR residuals (Table S2), and results in a physically implausible near-horizontal shear dike located too far north of the seismicity. The observed left-lateral shear residual may reflect transient stress perturbations during magma emplacement, possibly linked to shear dilatancy effects or local stress rotations, that may be physically linked to the rotated focal mechanisms. Alternatively, the shear residual may be due to the differential timescale of opening between the mantle and crustal intrusion segments, as described

		above. However, the sparse GNSS dataset cannot provide confident constraints on such sources.  We have explicitly mentioned that we inverted “three-component GNSS data” in the third paragraph of Section 2.1.
R1-7	Reviewer comment	As far as I understand, and as the authors state in the Methods section (Section 4 – “Mechanical modeling of surface displacement observations”), “...the model assumes that the volcanic island edifice and upper crust are homogeneous and isotropic.” While this is a common approach when modeling the crust, their Vp velocity model shows a significant contrast between the physical properties of the crust and the upper mantle (a difference of more than 4 km/s from the surface to 10 km depth). The strong variations observed in the 1D velocity model are consistent with earlier studies on the island’s petrology (Zanon et al., 2020 and 2023), which identify distinct transition zones, with the Moho boundary near 10-12 km depth and a second transition around 25 km depth. Furthermore, Supplementary Material Figures S11 and S12 illustrate typical shapes of two overlapping probability distributions in terms of “shear” and “length”.
	Author response	We thank the reviewer for this comment. Although it is sometimes good to introduce more complexity in models, we do not have the data coverage to resolve such small differences. The large vertical extent (> 20 km) of the modelled intrusion exceeds the characteristic scale of these layers. The 1-D velocity model is most uncertain in the upper few km because we have no seismicity shallower than 6 km depth (Fig. S6). Introducing layering will not change the top-depth of the dike because that is well constrained with the observed InSAR lobes. Layering might scale the opening slightly, but we are not overly interpreting that value.
	Changes made	None.

R1-8	Reviewer comment	This evidence, along with the results from earlier studies (for example, the ones that the authors reference in the main text), led me to suggest that the results might indicate a more complex deformation pattern than a simple quadrangular plane (i.e., two overlapping sources of deformation, such as a shallow dike and one or multiple deep reservoirs inflating/deflating, or any other combination). This is a common behavior observed in volcanic systems and oceanic volcanic islands, which has been documented in multiple unrest processes across different geodynamic scenarios, as the authors reference in the manuscript (Hudson et al., 2017), or as suggested by petrological studies in the archipelago (Zanon et al., 2020, 2023). Numerous examples can be found in previous studies, but one clear and recent example cited by the authors is the La Palma eruption, which also exhibited a migration of deep reservoirs (~20-30 km depth) years-to-months prior to the volcanotectonic unrest (Torres et al., 2019). In this case, the process was also dominated by regional-to-local stress (Del Fresno et al., 2022). Why did the authors exclude or avoid considering other possible and well-established models for this type of unrest?
	Author response	We thank the reviewer for this thoughtful and well-informed suggestion. We fully agree that complex deformation patterns involving multiple sources—such as shallow dikes combined with deep reservoirs—are commonly observed in volcanic systems, including in the Azores (e.g., Zanon et al., 2020, 2023; Hudson et al., 2017; del Fresno et al., 2022). The pattern of apparent precursory seismicity at ~30 km depth beneath the São Jorge channel also draws parallels to these other cases. From a formal geodetic view, however, in our study, we find that such a vertically extensive nature of the modelled dike is required by the far-field GNSS stations on adjacent islands; however, the exact depth is one of the most uncertain aspects of the model (formal uncertainty of ± 8 km; Fig. S11), with the best-fit model at the deeper end of the ensemble solutions, but most models reaching beyond 20 km depth bsl (Fig. 4b). Moreover, the localised and spatial symmetric and amplitude of the InSAR signal, combined with the far-field GNSS displacements, are best explained by a single, vertically extensive dike. We believe that deep precursory seismicity indicating migration of deep reservoirs does not contradict a vertically extensive dike opening because such small, deep sources would likely not have a resolvable geodetic signature. Nevertheless, we explicitly tested a wide range of alternative source models, including:  - Deep spherical or ellipsoidal pressure sources (inflating or deflating) - Shallow and deep dike combinations - Fault slip models - Composite models (e.g., dike + reservoir, or dike + fault) Critically, our analysis suggests that none of these multi-source configurations can reproduce the observed GNSS and InSAR deformation patterns. We find that deep inflation/deflation sources produce broad symmetric patterns inconsistent with such localised deformation, and composite models introduce trade-offs that degrade the overall fit, particularly incapable of explaining the eastward motion of São Jorge stations (QEMD and VLAZ), and outward-extensional motion of Graciosa and Pico stations. We could not explain the eastward residuals at Pico and Faial with any logical shear source. We appreciate the reviewer's encouragement to consider these scenarios and believe the revised manuscript now better communicates the robustness of our interpretation.
	Changes made	 • We have added a detailed discussion of our experiments to test many additional models raised by the reviewer - please see Text S1 and associated supplementary figures. • We have added the following paragraph to Section 2.2: “To test the robustness of our a priori quadrangle geometry parameterisation, we conducted an extensive exploration of the model space. These tests included testing a wide range of source geometries, such as single and multiple vertical, inclined, and curved dikes, spherical and ellipsoidal pressure sources, faults with both right- and left-lateral shear, and combinations of the above (Text S1). None of these alternative configurations provided an improved fit to both GNSS and InSAR data (Table S2.1)”

while remaining physically plausible. Our sub-vertical quadrangular dike model emerged as the preferred solution, as it is the only configuration that reconciles all datasets with small residuals and physically plausible parameters.”

- We have made some edits to the second paragraph of Section 3 (Discussion):
“Such deep precursory seismicity^{17,53–55} is often linked to magma accumulation or destabilisation in deep *migrating* reservoirs^{56–58}. Rapid *opening of a large vertically extensive dike, whose bottom depth matches well with this precursory seismicity* (Fig. 4), suggests swift magma ascent with nearly simultaneous dike opening in the crust and upper mantle ...”.

R1-9	Reviewer comment	The authors suggest that a near-vertical dike opens simultaneously from 26 km depth below sea level up to 1.6 km below the surface, with practically no seismic activity during the ascent. As I suggested earlier, a limitation on the methodology might be hindering seismic activity. It is quite difficult to imagine a magma body with enough energy to open a dike quasi-simultaneously from 26 km depth to the surface, facilitated by a “mantle-rooted” structure (PdC-FZ), reaching the basement of the volcanic edifice, stagnating almost aseismically, and yet the same structure being compressed enough to release energy due to lateral volatile transfer. Additionally, if PdC-FZ is a mantle-rooted structure, why is the basement of the dike shifted to the north and not aligned with this structure?
	Author response	We thank the reviewer for this thoughtful and important critique. We agree that the scenario of a near-vertical dike opening quasi-simultaneously from ~26 km depth to ~1.6 km below the surface, largely aseismically, appears to be unusual and warrants careful justification. First, regarding the seismicity methodology and potential observational bias, the seismic network was dense and sensitive enough to detect microseismicity down to ~30 km depth, as evidenced by the deep precursory clusters we identified. Moreover, we searched for precursory tremor many days before the start of the seismic crisis, but did not find any, even though we were able to record ~30 km deep precursory events, and we successfully identified tremor bursts once the seismic swarm started. Therefore, we are confident that the lack of seismogenesis during ascent is not due to methodological limitations, but rather reflects a physical, mechanical phenomenon. Regarding the lack of seismicity during ascent, we emphasize that this is a well-documented phenomenon in other volcanic systems. For example:  • The 2014–15 Bárðarbunga–Holuhraun dike, Iceland (Ágústsdóttir et al, 2019, doi: 10.1029/2018JB016010). • The 2021 La Palma eruption, where the magma ascended aseismically at 13-33 km depth (del Fresno et al., 2023, doi: 10.1038/s41467-023-35953-y). • The 2005 Afar rifting episode in the East African Rift (Belachew et al., 2011, doi: 10.1029/2010JB007908). In fact, the lack of upward migrating hypocentres indicative of vertical melt transport is not that unusual a phenomenon, and may be the rule rather than the exception, as noted by Roman & Cashman (2018, doi: 10.3389/feart.2018.00124). These examples demonstrate that induced/triggered seismicity depends on the stress history of faults and the rheology of the magma and the country rock (e.g., brittle-ductile transition depth). The seismic response also likely depends heavily on the fracture toughness (Rivalta et al., 2015, 10.1016/j.tecto.2014.10.003) at the dike tip. Also, because the 2022 São Jorge dike intruded into a mature fault zone, no new fractures had to be created, leading to aseismic Mode-I opening at the dike tip (e.g., Ágústsdóttir et al, 2019, doi: 10.1029/2018JB016010; Glastonbury-Southern, et al., 2025, doi: 10.1029/2024JB030162). Second, the feasibility of a deep dike opening quasi-simultaneously (within 24 hours) is supported by the geodetic data. The GNSS and InSAR displacement data are consistent, within the limitations of the GNSS and InSAR sampling rates (GNSS daily solutions and the 3-day temporal separation between primary and secondary images of InSAR ascending and descending pairs). Therefore, we can confidently determine that the dike was emplaced in less than 3 days, and possibly, but less certainly, within 24 hours. We find no evidence of precursory surface displacement (see our answer to R1-5 above, and new Text S2). The GNSS and InSAR signals are best explained by a vertically extensive, near-vertical dike (the dip angle towards the north is small. The dip is towards the north, and is not considered a robust feature beneath the 3-4 km depth, where the dip is strongly constrained by the InSAR dense and nearfield observations. Alternative models involving deep inflation sources or multiple stacked intrusions failed to reproduce the observed deformation pattern (see our answer to R1-6 above, and new Text S1). Moreover, previous well-observed dike intrusions show that magma can propagate at speeds of 0.1-1.0 m/s (e.g., Belachew et al., 2021, doi: 10.1029/2010JB007908; Segall et al., 2013, doi: 10.1002/2013JB010251; Wright et al., 2012, doi:

	10.1038/ngeo1428; Ágústsdóttir et al, 2016, doi: 10.1002/2015GL067423; Peltier et al., 2005, doi: 10.1029/2005GL023720; González et al., 2015, doi: 10.1002/2015GL066003). So magma ascending vertically over some ~20 km beneath São Jorge over the timescale of ~1 day is by no means unprecedented, and has been observed not only during lateral dyke propagation along rift-zones, but vertically in similar ocean island volcanoes, e.g., Fogo Island, Cape Verde (González et al., 2015). Such a rapid ascent in São Jorge is consistent with a high magma overpressure and the presence of a pre-existing, mechanically weak structure - i.e., a crustal fault. Such mechanisms for rapid ascent could include:  • A fluid-assisted propagation mechanism. • A hybrid diking-diapiric transport mechanism near the brittle ductile transition (Rivalta et al., 2015; doi: 10.1016/j.tecto.2014.10.003; Rubin, 1993, doi: 10.1016/0012-821X(93)90109-M; Sumita & Ota, 2011, doi: 10.1016/j.epsl.2011.01.032), which would also generate the aseismicity at 15-25 km depth. Third, regarding the alignment with the PdC-FZ, we agree that the dike is slightly offset to the north of the mapped surface trace of the fault. We noted this in the original version of our manuscript in the third paragraph of our Discussion: “The dike intruded parallel to the PdC-FZ and VT seismicity zone, east of the mapped fault zone where it is likely buried by lavas and scoria cones (Fig. 4).” This is not inconsistent with the PdC-FZ acting as a mantle-rooted structure. The surface trace of the fault may not coincide exactly with its deeper geometry (nearly vertical), especially in volcanic terrains where fault segments can be buried or deflected by intrusive bodies. The focal mechanisms, seismicity alignment, and dike strike all support a structural connection between the dike and the PdC-FZ.
Changes made	We have added the following new paragraph to the Discussion: “Although required by the geodetic data and ACF results, a near-vertical dike opening in a matter of days from 26 km depth up to 1.6 km appears to be an unusual aspect of this event. However, magmatic intrusions propagating rapidly at ~0.1-1.0 m/s⁵⁹⁻⁶², with partial or total aseismicity especially at mantle depths, are not without precedence^{57,59,63}. We infer that because the dike intruded into a mature fault zone, no new fractures had to be created, leading to aseismic Mode-I opening, possibly accompanied by devolatilisation and shear dilatancy at the dike tip^{62,63}. Nevertheless, such a stealthy and rapid magma ascent presents a challenge for tracking dike intrusions and forecasting their eruptive potential.”

R1-10	Reviewer comment	In this model, the authors also state that “...magma stalled ~1,600 m below the surface, near the estimated base of the island edifice,” and later attribute the seismic activity to “thermal pressurization and magma devolatilization,” which increased its viscosity. However, if magma crossed the upper mantle and crust in less than a day, with enough devolatilization to generate such high seismic activity (including dozens of felt earthquakes), and stalled at these shallow depths, I would expect gas emissions. Even nowadays, some fingerprints of this unrest may still be detectable. Is there any geochemical evidence or reports supporting your interpretation (e.g., groundwater temperature increases, isotopic variations, or diffuse CO₂ emissions)?
	Author response	We appreciate the reviewer’s thoughtful question regarding the expected gas emissions associated with shallow magma stalling and devolatilization. This is a similar comment to R2-9 and R2-12 below. The original version of the manuscript briefly references a 2023 conference abstract by Asensio-Ramos et al. (https://meetingorganizer.copernicus.org/EGU23/EGU23-6185.html) who reported anomalous soil He and H₂ concentrations during the 2022 São Jorge unrest. However, we do not know if this study had an accurate pre-unrest baseline, and we are not aware of any other open or peer-reviewed datasets reporting additional geochemical anomalies—such as groundwater temperature changes, isotopic variations, or diffuse CO₂ emissions—associated with the 2022 unrest. No further results have been published that could support or challenge our interpretation. We acknowledge that the absence of widespread gas emissions may seem counterintuitive given the inferred shallow stalling depth and intense seismicity. However, we note that the magma likely stalled beneath the base of the expected volcanic edifice, where permeability conditions might be different, and may have inhibited efficient gas escape to the surface. Additionally, the rapid ascent and short-lived nature of the intrusion may have made potential gas anomalies or surface expressions very short-lived in time. These factors, combined with the lack of continuous gas monitoring infrastructure above the dyke intrusion at the time, may explain the apparent absence of detectable emissions.
	Changes made	We added the following text to the fourth paragraph of the Discussion: “We are not aware of any monitoring infrastructure, data, or reports (peer-reviewed or otherwise) of geochemical or gas abnormalities observed during the dike intrusion and start of the seismic crisis. We speculate that the short-lived nature of the intrusion and the different permeability conditions of the volcanic edifice may have inhibited efficient gas escape.”

R1-11	Reviewer comment	At approximately 12 km depth, in the vicinity of an aseismic area, the authors state that there is no evidence of magma presence or injection (a claim with which I am not entirely convinced,) and the seismic activity is primarily driven by thermal pressure and lateral fluid migration. What mechanism could prevent triggering during the dike-opening process yet lead to seismic events afterward due to fluid transfer? How do the authors explain the presence of this ~2 km wide aseismic area? Please provide some reasoning for this observation.
	Author response	In Section 4 of our original Methods section, we describe a suite of forward model tests in which we tested 20,000 randomly-generated models of dike intrusions with different volumes centred on the aseismic zone of the main seismicity cluster. Our results show that volumes corresponding to widths ranging from approximately hundreds of metres to the width of the aseismic zone (2 km) should have been detectable via InSAR and GNSS observations. However, such signatures are not present in the data. Therefore, we can rule out the hypothesis of a single dike intruding into this aseismic zone. However, we cannot rule out smaller-scale fingers of melt intruding into this zone.
	Changes made	In this revised version of the manuscript, we have decided not to completely rule out small fingers of melt helping to trigger the seismicity, rather than focussing exclusively on the devolatilisation interpretation:  • In the Abstract, we now say: "...with melt and fluids leaking laterally along the fault zone, triggering an intense, months-long seismic swarm". • We have added the following sentence to Paragraph 6 of the discussion: "However, we cannot rule out that small fingers of melt with <102 m3 volume propagated laterally from the main intrusion into this area, and possibly intermixed with volatiles." • We have changed the first sentence of Paragraph 7 of the discussion: Lacking evidence for substantial magma directly triggering the main zone of seismicity, we propose that fluid pore pressure increases, driven by thermal pressurisation⁸¹, pockets of melt, and magma devolatilisation⁸²⁻⁸⁵, drove the seismicity. • We have made some edits to the final paragraph of the Introduction: "Our results suggest that pre-existing faults can have opposing effects on magma propagation. The PdC-FZ facilitated rapid vertical melt ascent, but may also have caused magma to stall by allowing lateral devolatilisation and melt escape, increasing viscosity and reducing pressure within the main intrusion. The Azores thus offers new insights into the interplay between magmatism and seismic cycles." • We have made some relevant changes to the final schematic interpretation in Figure 7.

R1-13	Reviewer comment	INTRODUCTION: (Second Paragraph – "... in ocean island setting where seismograph coverage is limited ...") I suggest changing seismograph coverage for seismic network.
	Author response	Agreed.
	Changes made	Changed this sentence to: "... where seismometer network graph-coverage is limited ..."

R1-14	Reviewer comment	Section 2.1 - (First Paragraph) This section only focuses on the horizontal GNSS components, add some statements on your results of the vertical ones, and the uncertainties.
	Author response	Agreed.
	Changes made	 We described the subsidence signal at station QEMD - the only vertical GNSS signal that we can be confident of, based on noise levels and solution uncertainties. We also added a mention in the third paragraph of Section 2.1 that we inverted the three-component GNSS data.

R1-15	Reviewer comment	Section 2.1 - (First Paragraph) "...no significant deformation occurred before or after the seismicity") What do you mean by "after the seismicity", as I can relate, the seismic activity remains months after the first burst of activity. Maybe referring to certain dates this sentence is clearer.
	Author response	Agreed - this sentence was unclear because it was missing some key words.
	Changes made	We have changed this sentence to: "... no substantial significant deformation occurred before or after the onset of seismicity on 2022-03-19 "

R1-16	Reviewer comment	Section 2.1 - (Second Paragraph) "All deformations detected by InSAR occurred between 15 and 21 March 2022 Figs. S3-S5, ...") Could you provide the unwrapped InSAR images in this supplementary figure, it would help to the non-InSAR readers specialist to better understand your great results!
	Author response	Agreed.
	Changes made	We have added a new figure, Fig. S6, which shows the unwrapped ascending interferogram, and have cross-referenced this figure in the above-quoted text.

R1-17	Reviewer comment	Section 2.2.1 - (Second Paragraph) (First Paragraph – “We find deep clusters of ML...”) Which ML equation are you using? Please provide a reference.
	Author response	This is a good point. We did not focus too much on the M_L scale in the original manuscript because plotted magnitudes and the computed b-value were based on our computed moment magnitudes (M_w). We computed a new magnitude scale including station corrections by inverting observed amplitudes of the Sao Jorge seismicity using least-squares, similar to the method described by Hillsley-Kemp, et al. (2017). This computation yielded the following M_L equation: $M_L = \log(A) + 0.656 \log(r/17) + 0.00948 \log(r - 17) + 2 + C$ where A is the Wood-Anderson simulated horizontal-component amplitude in mm, r is the hypocentral distance in km, and C represents station corrections.
	Changes made	We have added a new subsection, “Magnitudes” to the Seismic Catalogue section of Methods, added the above details, and moved the M_w computation details from the b-value section, so this section now reads as follows: Magnitudes We computed a new magnitude scale including station corrections by inverting observed amplitudes of the Sao Jorge seismicity using least-squares¹⁰⁹. This computation yielded the following ML equation: $M_L = \log(A) + 0.408 \log(r/17) + 0.0138 \log(r-17) + 2 + C$ where A is the Wood-Anderson simulated horizontal-component amplitude in mm, r is the hypocentral distance in km, and C represents station corrections. We also computed M_w using SH-wave spectra^{116,117}. We found that compared to P-wave spectra, M_w estimates from SH-wave spectra had smaller uncertainties, had a greater number of good spectral fits, and had a stronger correlation with the corresponding initial ML.

R1-18	Reviewer comment	Section 2.2.1 - (First Paragraph – “The seven weeks immediately before the onset of the seismic swarm was aseismic”) Add “based on our results”.
	Author response	Agreed.
	Changes made	We have added this phrase to the start of the sentence: Based on our results , the seven weeks immediately before the onset of the seismic swarm were aseismic

R1-19	Reviewer comment	Section 2.2.2 - (Second Paragraph) This paragraph is difficult to interpret without a weekly/monthly histogram of activity. The authors properly describe the location evolution, but the reader does not know “when” and “how many”.
	Author response	Agreed.
	Changes made	We have added a daily event rate bar chart to Figure 3, along with a cumulative number of events curve. We have also added some more details describing the space-time evolution of seismicity in this paragraph.

R1-20	Reviewer comment	Section 2.2.2 - (Second Paragraph) (Second Paragraph – “All events after 19.2022 lie between 6-14 depth, most concentrated at 9-14 (Fig.4)”). I suggest adding “near to the Moho discontinuity in the Island (Ferreira et al., (2020))”
	Author response	Agreed. We also estimate the Moho depth from our inverted 1-D velocity model.
	Changes made	We have changed this sentence so it now reads: “All events after 19 March 2022 lie between 6-14 km depth, most concentrated at 9-12 km, near the estimated crust-mantle transition beneath São Jorge based on our inverted 1-D velocity model (Fig. 4) and independent estimates of Moho depth⁴⁷. ”

R1-21	Reviewer comment	Section 2.2.2 - (Second Paragraph) (Second Paragraph – “...extending upwards to ~6km depth from the eastern edges of UL and LL (labelled Shallow, “SH) ...”). I suggest changing SH for any other label, as SL, to avoid confusion with SHmax in Fig. 4.
	Author response	Agreed - good point.
	Changes made	We have changed these labels to “ SB ”, meaning “shallow branch”, and have modified the appropriate notations in the main text.

R1-22	Reviewer comment	Section 2.3 - (First Paragraph – “...São Jorge, PM.PMAN and PM.ROSA (Fig 4a.) ...”) change to PM.PMAN (Fig. 4a) and PM.ROSA (Fig. S13).
	Author response	Agreed
	Changes made	Changed

R1-23	Reviewer comment	Section 2.3 - (First Paragraph – ... show coherent arrivals at lag times of ~2-24 s) Fig. 4b only displays from 6-24 seconds and Fig. S13, from 16 to 24. Also, figure S13 shows a different timespan (x axis) from Fig. 4b. Please unify and correct where necessary.
	Author response	Agreed - thanks for noticing.
	Changes made	We have fixed the text and these figures.

R1-24	Reviewer comment	Section 2 – Frequency magnitude relationships: (First paragraph- “We found that compared to P-wave spectra, M_w estimates from SH-wave spectra had smaller uncertainties”) If able, add a plot to the supplementary material to support this statement.
	Author response	Good point - we weren't clear on this. The M_w estimates from S-waves are more robust because they yield a greater number of observations per station than P-wave estimates, especially for the lower-magnitude events.
	Changes made	 We have edited the text in this section so it now reads: “We found that compared to P-wave spectra, M_w estimates from SH-wave spectra yielded far more observations per station, generating more robust network-averaged M_w values. had smaller uncertainties, had a greater number of good spectral fits, and had a stronger correlation with the corresponding initial ML.” We have included a new figure, S14, which explains the greater number of observations per station for SH-wave spectra.

R1-25	Reviewer comment	Fig. 1-a panel:  Change “Elevation / bathymetry” for “Bathymetry / elevation” to be consistent with color in the legend. Remove the 500 and 1500 ticks in elevation legend to be consistent with the scale in bathymetry. Add a distance scale and arrow pointing to the north. Fig. 1-b panel:  Due to the color palette, the seismic network is practically indistinguishable from the background, therefore I suggest using a grey palette to the elevation Add a distance scale and arrow pointing to the north. Fig. 1-c panel:  If able, providing the unwrapped InSAR image in the supplementary material could help the readers to interpret your results
	Author response	Agreed.
	Changes made	 We have swapped the colourbar label components. We have made the bathymetry and topography colourbar tick-label intervals consistent. We have added north arrows and scale-bars to all panels. We have added a new supplementary figure, Fig. S6, showing the unwrapped interferogram. This figure is cross-referenced from the caption of Fig. 1.

R1-26	Reviewer comment	Fig. 2: In my opinion, this figure could be either removed or moved to the supplementary material.
	Author response	We believe that this figure is an important way for the reader to appreciate the geographic and geologic context of São Jorge island, and to emphasise how the mapped PdC-FZ, which is a key component of the story we present in our paper. Also, we found that the other two reviewers did not comment negatively on this figure.
	Changes made	We have kept the figure, but have modified it by adding the regional SHmax direction for extra context.

R1-27	Reviewer comment	Fig. 3a: Missing uncertainties in the plot. Please plot them. Why is so much missing data from December to February? Fig. 3b:  • The depth cannot be appreciated by this color scale, I suggest adding a subpanel showing the Tp-Ts in ROSA and/or PMAN or any other station in the island, along with the hypocenter-color scale in the plot. • A histogram could help the readers to understand the evolution of the seismic activity through the lapse analyzed • The vertical dashed line seems to be duplicated Fig. 3c:  • A legend to explain the colors of this subplot is needed. • Please highlight the main contributions of this panel with some squares or arrows. • Due to the color palette, the label and the vertical magenta line cannot be appreciated • The y-axis values are from 6-24 s, but in the main text says 2-24 s, please modify accordingly the plot and/or the text (2.3 Subsurface Velocity Changes-First Paragraph: "...from May to September 2021 show coherent arrivals at lag times of ~2-24 s") • Highlight the main changes of this plot, to non-seismologist readers it could be difficult to interpret.
	Author response	Agreed
	Changes made	 • We have added daily error bars to Figure 3a. • We have added new figures, 3c and 4a, that show the depth evolution as a function of time, with a zoom-in on the first few days of seismicity in Figure 4. • We have added a daily event rate bar chart to Figures 3 and 4, along with cumulative number of events curves. • We have cleaned up the vertical dashed lines. • We have added a note to the caption to explain that red and blue colours represent positive and negative amplitudes, respectively, in the ACF waveforms. • We have added arrows at the edge of the axis highlighting the onset of the unrest. • We have made the text consistent with what is shown in Figure 3c. • We have highlighted the main trends in the caption: "At the onset of the seismic-volcanic unrest (indicated by the small black arrow-heads above and below the time axis) the ACFs at lag times of <8 s and >18s become less coherent and shifted in time".

R1-28	Reviewer comment	Figure 4: As a common comment on this plot, please, unify the seismic color palettes. It's quite confusing to see how the seismic activity has 3 different color palettes, Fig. 4a referencing the depth, 4b and 4c the date but with different scales (up to 180 days and up to 200 days from 2023-03-19). Apart from confusing, it does not help the reader with its interpretation. This is a great plot, maybe with some clarifications it could be easier to interpret to the readers Panel a:  ● Add a distance scale and arrow pointing to the north. ● As I pointed out in Fig. 3-b panel, the color palette used to the topography make the plot quite confusing. As the authors are showing the altimetry as contour lines every 250 m, maybe the green background could be removed. ● As shown in Fig. 1-b panel, maybe the QEMD deformation can be added to this plot. ● In the main text the authors describe at least 150 deep earthquakes, why can only be seen a dozen of them? Panel b  ● In my opinion, the "VEN arrow" does not provide any information, at first sight I thought it was some deformation data. ● Please align the Vp major ticks with the distance grid, it is quite confusing with the current display. Panels c and d:  ● I suggest changing the near-surface earthquakes "SH" label to "SL", may be confusing, as long as, you are plotting in 4-a panel the "Regional SHmax".
	Author response	 ● We feel that adding the displacement vector from GNSS station QEMD would make the figure too busy and it would mask too much of the seismicity. ● The 150 deep earthquakes come from template matching, which doesn't yield hypocentral locations. We are able to relocate 36 of these events to high-precision - these were the purple dots in panels (a) and (b). We do however understand the confusion caused.
	Changes made	 ● We have unified the pre- and post-seismic crisis time colourbars, and have used these for all panels. ● We have added a scale-bar and north-arrow to panel (a). ● We have added the location of GNSS stations and labelled them in (a) and (d). ● We have switched the green topography colour scale with a greyscale one. ● We have aligned the P-wave velocity profile with the existing x-axis grid. ● We have changed the label "SH" to "SB" ("shallow branch"). ● We have substantially changed Figure 3 to better show the precursory events, and we have modified the paragraph in Section 2.2 to describe them more clearly.

R1-29	Reviewer comment	Figure 5: Panel a:  From Fig. 4-a, it seems to me, like some seismicity is missing. If you are only plotting those seismic activity clustered, maybe you can add the “unclustered” seismicity with grey dots in the background? Panels b, c and d:  I suggest changing the order, Cluster 4 on the left, Cluster 19 on the right, to be consistent with their real location. If the legend expands from 0 to more than 150 days in Fig. 5-b, why the x-axis in Fig, 5-c only expand to 140? In Fig. 5-c, does the first location (star) compare with affects to this plot? Because I see a slight migration in Cluster 19; but seeing cluster 4, it looks like if I change the position of this first point maybe some migration could be seen. In Fig.5-d, maybe change the waveform colors from clusters 4, and 19 to brown and green as they are shown in Fig.5-a and Fig.5-c
	Author response	Agreed.
	Changes made	 We have added the unclustered seismicity in the background. We have switched the horizontal position of the cluster panels. We have made the colour scale consistent with the time axis (good spot!). We have re-analysed the clusters and instead of showing the original Cluster 4, we now show (new) Cluster 18 which we think also shows a front that is consistent with a diffusivity used for the other cluster. We have made the waveform colours consistent with the cluster panels above.

R1-30	Reviewer comment	Figure 6: Panel a:  I really like this plot; it is super intuitive! Panel b:  Move it to the supplementary material? To me it does not fit with the model in the same figure.
	Author response	Thanks for the comment about Panel a. We agree that the two panels can be separated.
	Changes made	We have moved panel b to a standalone Figure - Fig. 6.

R1-31	Reviewer comment	Supplementary Figures: S1 & S2: Please align the y-major ticks (and the grid) with the daily displacements in each panel. It will help to highlight the displacements. S3 & S10: Please provide a color-scale to the values of the interferograms. Comparing this plot with the Fig. 1c, it seems like the color-scale has different values, please unify them. S4: Same as Fig S3. Additionally, if unwrapped InSAR images could be provided will be helpful, with a scale in [cm]. S8: Many stations codes are overlapped, please fix it. S11 & S12: It seems as the quality of the image is too low, please provide it with higher resolution. S13: As I stated before, this plot is inconsistent with the plot in the main text Fig. 3c, please unify them
	Author response	Agreed.
	Changes made	 ● We have aligned the traces with the grid and station label in Fig. S1. ● We have included the unwrapped ascending interferogram with a colourbar in Fig. S6. ● We have replaced Figs. S11 and S12 with higher-resolution images. ● We have made Fig S13 consistent with Fig. 3c.

Reviewer 2 (Valerie Cayol)

Overall comments

The paper is a detailed analysis of the seismicity and deformation of a failed eruption that took place in 2022 in São Jorge island, Azores, an island with fairly rare eruptions as the previous one took place in 1808. The event is well documented by onshore and offshore seismic recordings from permanent and temporary networks, including OBS which improved events locations. At the time of the intrusion, nine permanent GNSS stations were operating on São Jorge and the neighbouring islands, and these data were completed by InSAR acquisitions, from ascending and descending orbits of the Sentinel-1 satellite. Seismic and displacement data are analysed with up to date methods, showing that a dike stalling at 1.6 km depths was responsible for the unrest. From magnitudes of EQ in swarms west of the dike, the b-value is computed. Hypocentres are relocated, the velocity model is improved, and temporal variations in velocity structure are imaged using autocorrelation. Analysis show that both the seismic swarm and the dike are located along a well-know fault (Pico do Carvao, PdC-FZ). These most remarkable feature shown is that Fault Plane Solutions (FPS) on seismic events that took place during and after the dike emplacement are consistent with left lateral fault displacement, when paleo-seismology, long term GPS solutions, and the tectonics indicate that long term slip along the fault is right lateral. The discussion concludes that the left lateral focal mechanisms are related to the dike opening, despite the right lateral long term slip. The low-b value of the post-intrusion seismic swarm, and well as the large seismic moment relative to the intruded volume are interpreted as consequences of the percolation of magmatic fluids released by magma devolatilation. This devolatilation is assumed to have increased the viscosity and made the dike stall at depth. The paper addresses a well documented and rare event, with relevant methods. It is clearly and logically organized with relevant references. However, I have a few major concerns regarding the main conclusions, which are not discussed thoroughly enough to be fully convincing. Mainly, the reason for the inconsistency of the focal mechanism with the long term slip is not sufficiently explained and, secondly evidences, for the fluid percolation being responsible for the swarm to the west, the increased viscosity and the failed eruption are too slim. Below are suggestions of elements to discuss in more details in order to make the conclusions more robust.

General author response

We are grateful for the insights provided by Reviewer 2, particularly in their independent analysis of the geodetic models and their view on our overall physical interpretation. We hope our modifications, particularly about the focal mechanisms rotation and the evidence for fluid percolation, have led to a clearer set of main conclusions that are more convincing.

R2-1	Reviewer comment	My major concern is the lack of discussion around the left lateral mechanism of seismic events in the swarms west of the main dike, when the tectonics indicates that the sense of slip is right lateral. How can the authors explain that the dike intrusion induced a Coulomb Stress Changes (CSC) large enough to overcome the tectonic loading? Does it mean that this tectonic loading was null at the level of the swarm because it had been released by previous diking events west of the present dike intrusion? Is this possible regarding slip accumulated since the previous dike intrusion west of the dike? It would be convincing to provide some orders of magnitude of this long term slip, and compare them to those potentially induced by the 2022 event.
	Author response	First, we don't believe that the seismicity is solely occurring due to a Coulomb stress change - there must be a primary pore pressure or weakening mechanism due to melts/fluids. Also, given that the earthquakes are due to left-lateral slip, we cannot be certain that they are actually releasing regional tectonic right-lateral accumulated strain. For a rotated focal mechanism, magma pressure must counteract regional stress, and the stress patterns gradually approach those obtained in the absence of any regional stress field. Thus, ubiquitous rotated focal mechanisms are indicative of highly pressurised magma (Vargas-Bracamontes & Neuberg, 2012, doi: 10.1016/j.jvolgeores.2012.06.025), or a weak tectonic stress. Possibly, a relatively weak tectonic stress on São Jorge is not completely unsurprising. The island sits close to, but possibly south of the diffuse Terceira rift transtension plate boundary zone. Also, it is not just long-term rifting that is accommodating tectonic stress, it is possibly also intrusive events, such as the 2022 dike, which accommodates this transtension. As an alternative or complementary explanation to the tectonic stress effect, Roman & Cashman (2006, doi: 10.1130/G22269.1), and later Roman et al. (2021, doi: 10.1038/s41586-021-03400-x) report that the focal mechanism rotation phenomenon depends on magma viscosity. We believe the prevalence of rotated focal mechanisms during the São Jorge unrest is because the dike intruded the fault which happens to also host the triggered seismicity. The rapid ascent of an overpressured dike will also have made a focal mechanism rotation more likely (Roman & Heron, 2007, doi: 10.1029/2007GL030222). These papers above cite numerous examples in the published literature of 90° focal mechanism rotations, with respect to regional stress direction, during volcanic unrest and magma intrusion.
	Changes made	We have rewritten the relevant section of the Discussion: “Lateral melt branching, exsolution and subsequent devolatilisation, likely increased magma viscosity^{4,59,84}, helping the dike to stall ~1.6 km below the surface. Such a dynamic viscosity increase could have led to the observed left-lateral seismogenic shear along the PdC-FZ, which is rotated by 90° to the background right-lateral motion^{21,22,26,30–33,35,44} (Figs. 1a, 2, 4a). Alternatively, shear dilatancy effects from an overpressured magma in a relatively weak background stress field may have also caused the focal mechanism rotation^{13,15,78,85”}.

R2-2	Reviewer comment	It is indicated that the PdC-FZ was probably more pre-stressed than the P-FZ. Which magmatic or tectonic event could have pre-stressed the PdC-FZ?
	Author response	At the end of Paragraph 5 in the Discussion, we say in the original version of the manuscript: “ This suggests the PdC-FZ was more pre-stressed and closer to failure. The PdC-FZ is thus a major mantle-rooted structure in the central Azores that accommodates regional transtension^{23,37,43,51}, especially given its recent eruptive history and that it has generated Mw~7 earthquakes⁴³, likely involving full crustal rupture. ” Here, as well as in the Introduction, we are saying that the PdC-FZ has become the long-term locus of seismogenic and volcanic activity on São Jorge, and we imply, albeit indirectly, that these previous events, along with long-term tectonic loading, may have affected the stress / loading state of the fault in the lead up to the 2022 unrest. However, we realise that we were not entirely clear with this link. But we don’t believe that a transient tectonic event pre-stressed the PdC-FZ.
	Changes made	We have slightly edited these sections: “ This suggests the PdC-FZ was more pre-stressed and closer to failure. The PdC-FZ is thus a major mantle-rooted structure in the central Azores that accommodates regional transtension^{23,37,43,51}, and appears to have been the locus of recent magmatic and seismogenic activity on the island, especially given its recent eruptive history and that it has generated Mw~7 earthquakes⁴³, likely involving full crustal rupture. Such events, along with long-term tectonics, may have pre-loaded the fault in the lead up to the 2022 unrest ”.

R2-3	Reviewer comment	What would the CSC be on the conjugate fault indicated by the focal mechanism?
	Author response	We computed the coulomb stress change for right-lateral slip along the conjugate fault. The two figures below show the original result (left-lateral slip on the WNE-ESE plane) on the left, and the conjugate plate slip result on right (note the different y-axis scales).     (c) This result shows that assuming the conjugate fault plane doesn’t hugely change the results, in that most events lie in a slightly positively stressed region. However, a fraction of events move into a negatively stressed region, which makes failure along the conjugate plane somewhat less likely. Also, none of the seismicity lineations or mapped pre-existing structures suggest activation or existence of the conjugate failure plane. By asking this question, we infer that the reviewer is referring to the the model of Hill (1977, doi: 10.1029/JB082i008p01347), in which a cluster of magma-filled dikes oriented parallel to the direction of regional maximum principal stress is linked by a system of conjugate shear faults joining en echelon offset dike tips at oblique angles, and so dike expansion is transferred through the mesh by right- and left-lateral faulting on such conjugate faults. We don’t think that this model applies to the 2022 São Jorge unrest because most seismicity occurs after the apparent emplacement of the dike (at least the part during the geodetically observed deformation).
	Changes made	We have changed the relevant sentence of the Discussion: “ The PdC-FZ and seismicity lie in a zone positively stressed by the modelled dike intrusion, both for left- and right-lateral strike-slip failure on receiver faults (>50 kPa; Figs. S19-S20). ”

R2-4	Reviewer comment	How do the authors explain that no event took place east of the dike, in a region of increased Coulomb stress as evidenced by Fig. S14?
	Author response	We agree that the symmetry of stress increase predicted by coulomb stress changes does not manifest in the triggered seismicity. In particular, there is no evidence for the classical “dogbone” seismicity distribution. We already made this point in the Discussion section of the original manuscript: “Unlike typical dike intrusions, where seismicity surrounds the dike in a ‘dogbone’ pattern^{15,65-72}, the 2022 São Jorge event showed seismicity confined to one side, with no activity near or elsewhere around the dike, except for the shallow branching “SBH” zones linked to dike ascent. The PdC-FZ and seismicity lie in a positively stressed zone (>50 kPa; Fig. S14), but similarly oriented nearby structures, like the P-FZ, remained aseismic. This suggests the PdC-FZ was more pre-stressed and closer to failure.” Several factors may contribute to this effect:  • No fault on the eastern side of the island has the same orientation as the PdC-FZ. • The surface geology of São Jorge is divided into three main volcanic complexes: Rosais, Manadas (where the 2022 unrest occurred), and Topo. This segmentation in geology might correspond to different properties or fluid/melt propagation barriers that might prevent seismicity from occurring across segment boundaries. Moreover, these segments appear to behave as blocks with distinct long-term motion (Mendes et al., 2013, doi: 10.1016/j.asr.2012.10.019) • Fluids were pre-existing along the PdC-FZ, especially given the fault extends offshore, before the dike intrusion and seismic swarm, enhancing the triggering of seismicity. • Our ensemble of well-fitting models all require a quadrangular model that tilts/leans to the west (Fig. 4, Figs. S7-S9). We wonder whether this tendency is somehow mechanically/dynamically easier for fluids/melt to bleed off laterally from the dike.
	Changes made	We have re-written the relevant section of the Discussion, so it now reads: “Unlike typical dike intrusions, where seismicity surrounds the dike in a ‘dogbone’ pattern^{16,61-68}, the 2022 São Jorge event showed seismicity confined to one side, with no activity near or elsewhere around the dike, except for the shallow branching “SB” zones linked to dike ascent. However, few similarly oriented faults have been mapped on the eastern side of the island (Fig. 2c), and the long-lived segmentation of three volcanic complexes across the island²⁴ might have provided a structural barrier to fluids east of the dike. The PdC-FZ and seismicity lie in a positively stressed zone (>50 kPa; Fig. S14), but similarly oriented nearby structures, like the P-FZ, remained aseismic. Such localised seismicity is thus likely due to the PdC-FZ providing a direct hydraulic trigger between the westward-tilting intruded dike and earthquakes, allowing fluids to directly trigger seismicity. The PdC-FZ is a major mantle-rooted structure in the central Azores that accommodates regional transtension^{24,38,44,52}, and appears to have been the locus of recent magmatic and seismogenic activity on the island, given its recent eruptive history and that it has generated Mw~7 earthquakes⁴⁴, likely involving full crustal rupture. Such events, along with long-term tectonics, may have pre-loaded the fault in the lead up to the 2022 unrest.”

R2-5	Reviewer comment	At the bottom of page 13, it is indicated that “the dike increased stress on PdC-FZ for both left and right lateral modes”, but only CSC on left lateral faults are shown in Fig. S14. To support this assertion, CSC for right lateral slip should also be shown.
	Author response	Agreed.
	Changes made	We have added a new figure, Fig. S20, showing Coulomb stress change for a right-lateral receiver fault, and we have added a cross-reference in the text.

R2-6	Reviewer comment	I also have concerns about the statement that "the seismic swarm released only Mw 4.7, leaving the likelihood of a future large earthquake uncertain". This is very vague. Is the likelihood of a future large earthquake increased or decreased? I believe that if the sentence concerns tectonics induced earthquakes consistent with the right lateral sense of slip, this likelihood should be decreased, but this should be more clearly indicated.
	Author response	We believe that because the seismicity is occurring as left-lateral slip, it doesn't make a future large earthquake (which would presumably occur as right-lateral slip due to accumulated regional tectonic stress), no less likely. We acknowledge that this wasn't clear in the original manuscript.
	Changes made	We have modified this sentence: " the seismic swarm released only M_w 4.7 of left-lateral slip, leaving the likelihood of a future large earthquake, which would presumably release regional tectonically accumulated right-lateral stress, no less likely uncertain .

R2-7	Reviewer comment	Considering that the dike intruded oblique to the minimum principal stress, coeval lateral displacement should have taken place making the dike a sheared intrusion, otherwise the model is inconsistent with the conclusion. Could the displacement data fit be improved by considering such a mechanism? The GNSS residual in Fig. S9, shows that shear displacements following a left lateral sense of slip could provide a better fit of the observation. I agree that such a slip direction is opposite to the tectonic slip direction, and it is surprising. It is also likely that when conducting an inversion considering a sheared intrusion, the other dike parameters (such as the strike) determined by the inversion will be different. This deserves to be further explored.
	Author response	We thank the reviewer for this insightful observation. Indeed, the dike intruded obliquely to the regional minimum compressive stress direction, so we have explored the possibility of coeval shear displacement—i.e., a sheared intrusion. In our current model, we assumed a purely opening-mode (tensile) dike, which provided a good fit to both GNSS and InSAR data. However, as the reviewer correctly notes, the GNSS far-field residuals (particularly in Fig. S9) show a consistent eastward misfit at southern stations, which could be indicative of an additional deep left-lateral shear component (not colocated with dike intrusion, neither in x-y position along the strike of the island or depth certainly not shallow). This is especially intriguing given that the inferred shear sense is more consistent with the observed focal mechanisms during the seismic swarm, and is opposite to the long-term tectonic motion along the PdC-FZ, which is right-lateral. To address this point, we have carried an inversion for a sheared dike model (see Text S1.1-1.4). We find that the inverted model unsatisfactorily increases the GNSS and InSAR residuals (Table S2.1), and results in an implausible near-horizontally dipping shear dike located too far north of the seismicity. We speculate that the observed left-lateral shear residual may reflect transient stress perturbations during magma emplacement, possibly linked to shear dilatancy effects or local stress rotations, that may be physically linked to the rotated focal mechanisms. However, the sparse GNSS dataset cannot confidently constrain such a source. Such mechanisms should be the focus of a future study.

	Changes made	We have added the following paragraph to the penultimate part of the Discussion: “Another remaining issue is that predicted horizontal GNSS vectors from our preferred quadrangular dike model at stations on islands to the south (AZTP and PIED on Pico; HORT on Faial) show a consistent eastward trend (Fig. S10), which could indicate a left-lateral shear source. This trend is particularly fascinating given the rotated left-lateral slip of the swarm seismicity. Therefore, we inverted the geodetic data for a sheared dike model (Text S1.1), but found that the resulting model unsatisfactorily increases the GNSS and InSAR residuals (Table S2), and results in a physically implausible near-horizontal shear dike located too far north of the seismicity. The observed left-lateral shear residual may reflect transient stress perturbations during magma emplacement, possibly linked to shear dilatancy effects or local stress rotations, that may be physically linked to the rotated focal mechanisms. Alternatively, the shear residual may be due to the differential timescale of opening between the mantle and crustal intrusion segments, as described above. However, the sparse GNSS dataset cannot provide confident constraints on such sources. ”
--	---------------------	--

R2-8	Reviewer comment	My second major concern is about the seismic swarm and dike arrest triggered by fluids leaking at depth west of the dike. The occurrence of a seismic swarm west of the dike characterized by a low [author comment: should be "high"] b-value, the high Mw magnitude relative to the dike volume, together with the intrusion not leading to an eruption support this hypothesis. This bundle of indications are in my opinion too weak to be part of the title and one of the main conclusions. I find it doubtful that magmatic fluids would migrate downward from the top of the dike to the LL and UL locations, instead of being released at the top of the dike which had previously formed. Therefore I would recommend being more prudent and I would keep this hypothesis as a likely scenario and not as a major conclusion.
	Author response	This is a similar comment to R3-1. Regarding the title, the phrase "leaky fault" was originally meant primarily to reflect the interpretation of devolatilised fluids further travelling laterally along the fault. However, we understand that the term "leaky" is probably not a physically accurate term for the vertical ascent of the dike. In the original manuscript, we focussed solely on this interpretation of lateral migration of fluids/melt. However, based on our analysis of the dike model uncertainties and its lateral asymmetry, with a requirement to "lean" towards the west, we now added an alternative explanation. It is possible that melt/fluids rose vertically up from the deeper part of the dike along its western edge, which entered the zone in which the seismicity was triggered. The aseismic zone in between the main clusters of seismicity can then be explained by a change in physical properties along the Moho. We do not, however, have the data or other physical constraints to distinguish between these two models. Therefore, we prefer to leave open both scenarios. In this sense, we understand the Reviewer's concern about solely relying on the lateral migration scenario. In our view, both scenarios do not change the main interpretation about melt/fluid migration along a pre-existing fault, so the title does not need to change.
	Changes made	 • We have updated our interpretation in Figure 7. • We have re-written much of the discussion to encompass these other more likely interpretations. The key paragraph in the Discussion now reads: "The highly active swarm had an unusually high b-value of ~2.4, compared to the global volcanic average of ~1.774–76, likely due to the fluid-rich environment along the PdC-FZ in the lower crust. The total seismic moment (~10¹⁶ Nm, or Mw 4.7) was large relative to the intrusion volume (~10⁸ m³) compared to other dike-related VT sequences (Fig. 8), slightly tending toward seismicity caused by fluid injection, which has greater seismic efficiency than most volcanic sequences (Fig. 8); thus indicating fluid-driven seismicity along the PdC-FZ. The rapid westward and downward migration of seismicity (Fig. 4), leaving a ~2 km thick aseismic zone between three seismic lineations ('UL', 'LL', and 'DL'; Fig. 5) may indicate that, following the main dike ascending to the base of the volcanic edifice, a laterally-propagating branch of magmamay have initiated at the western edge of the vertically-stalled intrusion, at 9-10 km depth. Forward modelling (see Methods; Fig. S15) shows such melt branches would need to be small (<10² m³ total volume), so that they do not induce any observable surface deformation. Such thin branches of melt might be consistent with the ultra-thin lineaments of seismicity that we observe (Fig. 5d & 6a). Destabilisation of melt at the western edge of the main dike may be dynamically favourable given the dike's tendency to lean to the west in our ensemble of geodetic solutions (Fig. 5). We still cannot rule out these branches of melt being intermixed with volatiles (Fig. 7), nor can we rule out vertical fluid ascent from the westward-tilting deeper part of the dike (Fig. 7), or a combination of both lateral and vertical transport. Some seismicity clusters, such as Clusters 2, 7, and 18 (Fig. 6; Movie S1), show complex fracture-like geometries and an upward migrating front over weeks to months with slow diffusivity (~0.001-0.002 m²/s), suggesting upward fluid migration along propagating fractures⁸¹ in the PdC-FZ".

R2-9	Reviewer comment	As these seismic events are located beneath the emerged part of the island, I wonder whether gas release could be confirmed by field or remote sensing observations, such as those provided by measurements from satellites, similarly as evidences at Ambrym by Shreve et al., 2019? If not what could be the reason?
	Author response	This is a similar comment to R1-10 above and R2-12 below. The original version of the manuscript currently briefly referenced a 2023 conference abstract by Asensio-Ramos et al. (https://meetingorganizer.copernicus.org/EGU23/EGU23-6185.html) who reported anomalous soil He and H₂ concentrations during the 2022 São Jorge unrest. However, we do not know if this study had an accurate pre-unrest baseline, and we are not aware of any other open or peer-reviewed datasets reporting additional geochemical anomalies—such as groundwater temperature changes, isotopic variations, or diffuse CO₂ emissions—associated with the 2022 unrest. No further results have been published that could support or challenge our interpretation. We acknowledge that the absence of widespread gas emissions may seem counterintuitive given the inferred shallow stalling depth and intense seismicity. However, we note that the magma likely stalled beneath the base of the expected volcanic edifice, where permeability conditions might be different, and may have inhibited efficient gas escape to the surface. Additionally, the rapid ascent and short-lived nature of the intrusion may have made potential gas anomalies or surface expressions very short-lived in time. These factors, combined with the lack of continuous gas monitoring infrastructure above the dyke intrusion at the time, may explain the apparent absence of detectable emissions.
	Changes made	We added the following text to the fourth paragraph of the Discussion: “We are not aware of any monitoring infrastructure, data, or reports (peer-reviewed or otherwise) of geochemical or gas abnormalities observed during the dike intrusion and start of the seismic crisis. We speculate that the short-lived nature of the intrusion and the different permeability conditions of the volcanic edifice may have inhibited efficient gas escape.”

R2-10	Reviewer comment	What could be the explanation for fluids not leaking from the top of the dike, but instead moving to reach LL and UL locations? fluids tend to move upward and not downward so the sequence of seismic events from SH to UL and LL, deeper than the top of the dike is surprising.
	Author response	This is a similar comment to R2-14 below. Based on these reviews and our re-analysis of the seismicity and the uncertainties and solution space in the dike model, our revised preferred interpretation is that melts and/or fluids instead leaked laterally from the main dike, at ~9-10 km depth, once it had stalled at the base of the volcanic edifice. Such volumes would need to be low (<100-to-1000 m ³), as found in our forward model tests described in the Methods and shown in Fig. S21. However, our available data does not allow us to distinguish small volumes of melts versus magmatic (i.e. devolatilised) fluids. We do, however, note that the LL and UL seismicity zones both appear to dip slightly towards the west, which might indicate a slight component of downward flow. This might be controlled by the normal-stress effect of the island topography or the pre-existing structure of the PdC-FZ.
	Changes made	We have included this hypothesis in our revised Fig. 7. We have re-written this section of the Discussion: “The rapid westward and downward migration of seismicity (Fig. 4), leaving a ~2 km thick aseismic zone between three seismic lineations (‘UL’, ‘LL’, and ‘DL’; Fig. 5) may indicate that, following the main dike ascending to the base of the volcanic edifice, a laterally-propagating branch of magma may have initiated at the western edge of the vertically-stalled intrusion, at 9-10 km depth. Forward modelling (see Methods; Fig. S15) shows such melt branches would need to be small (<10² m³ total volume), so that they do not induce any observable surface deformation. Such thin branches of melt might be consistent with the ultra-thin lineaments of seismicity that we observe (Fig. 5d & 6a). Destabilisation of melt at the western edge of the main dike may be dynamically favourable given the dike’s tendency to lean to the west in our ensemble of geodetic solutions (Fig. 5). We still cannot rule out these branches of melt being intermixed with volatiles (Fig. 7).” We have also made some minor changes to the Abstract to cover this interpretation.
R2-12	Reviewer comment	Is there any remote sensing or field observations of an increase in the volume of gases released at the top of the intruded dike ? Gases tend to collect above intrusions (See for Instance Menand and Tait, 2002) and usually take the shortest pathway to leak in the atmosphere, so that the region at the top of the dike, also located along the PdC-FZ fault, is the most likely places for fluids to be released. If not, how can this be explained?
	Author response	This is a similar comment to R1-10 and R2-9 above. To avoid duplication, please see our comments made in R1-10 and R2-9 above.
	Changes made	To avoid duplication, please see our changes detailed in R1-10 and R2-9 above.

R2-13	Reviewer comment	Is the speed at which fluids migrate in the fault compatible with what is known of fluid migration along faults ?
	Author response	As shown in the new Figure 4, which shows a zoom-in on the temporal variation of seismicity during the onset of the swarm, we see the initial branches of seismicity migrating upwards and eastwards and downwards and westwards at speeds of ~400-800 m/hr. Such speeds are consistent with other dike ascent rates (see response to R1-9 above, which cites key examples). However, seismicity migration rates within the main zone of seismicity are less clear, and in Fig. 6, we analysed individual clusters to infer possible diffusivity values, but patterns were unclear. Seismicity within individual clusters of UL (Fig. 6; Movie S1) appear to slowly migrate upwards, consistent with a diffusivity value of ~0.001-0.002 m²/s. Although they are very interesting, we feel that it is beyond the scope of this manuscript to interpret these different seismicity migration rates, and these will likely be focussed on in a separate manuscript which analyses the different lineations and filaments of seismicity in more detail.
	Changes made	Nevertheless, we have improved our description of these seismicity migration rates and diffusivities in Section 2.2, and we have improved several aspects of Figure 6 to make the results clearer.

R2-14	Reviewer comment	An alternative hypothesis, for the LL, and UL swarms is that that magma first intruded the area of the SH swarm and then tried to intrude west of this dike in the area of swarms LL and UL, but was further arrested by the high normal stress inherited from a previous historical intrusion. The involved volume could be low enough not to induce any deformation (with volumes consistent with S15).
	Author response	This is a similar comment to R2-10 above. Yes, having now refined our manuscript and considered in more detail the uncertainties and limitations of the geodetic model, and the first few hours of seismicity, we believe this is a plausible model. Indeed, such volumes would need to be low (<100-to-1000 m ³), as found in our forward model tests described in the Methods and shown in Fig. S21. However, in this interpretation, our available data does not allow us to distinguish small volumes of melts versus magmatic (i.e. devolatilised) fluids.
	Changes made	We have included this hypothesis in our revised Fig. 7. We have re-written this section of the Discussion: “The rapid westward and downward migration of seismicity (Fig. 4), leaving a ~2 km thick aseismic zone between three seismic lineations (‘UL’, ‘LL’, and ‘DL’; Fig. 5) may indicate that, following the main dike ascending to the base of the volcanic edifice, a laterally-propagating branch of magma may have initiated at the western edge of the vertically-stalled intrusion, at 9-10 km depth. Forward modelling (see Methods; Fig. S15) shows such melt branches would need to be small (<10² m³ total volume), so that they do not induce any observable surface deformation. Such thin branches of melt might be consistent with the ultra-thin lineaments of seismicity that we observe (Fig. 5d & 6a). Destabilisation of melt at the western edge of the main dike may be dynamically favourable given the dike’s tendency to lean to the west in our ensemble of geodetic solutions (Fig. 5). We still cannot rule out these branches of melt being intermixed with volatiles (Fig. 7).” We have also made some minor changes to the Abstract to cover this interpretation.

R2-14	Reviewer comment	Magma propagation is controlled by fracture toughness, magma overpressure, magma buoyancy and magma viscosity (Rivalta et al., 2015). Alternative mechanisms could explain that magma stalled at depth. For instance, the injected magma volume and hence the dike overpressure, or alternatively the buoyancy could be insufficient for further upward propagation.
	Author response	We thank the reviewer for highlighting the importance of alternative mechanisms that could explain why magma stalled at depth. We fully agree that, in addition to thermal pressurization and devolatilization, other physical parameters such as fracture toughness, magma overpressure, buoyancy, and viscosity (Rivalta et al., 2015) play critical roles in controlling magma propagation.
	Changes made	We have substantially changed the fourth paragraph of the Discussion to acknowledge that insufficient overpressure or buoyancy could also have contributed to the arrest of the dike. These mechanisms are consistent with the observed deformation and seismicity patterns and provide a complementary explanation to the dynamic viscosity increase proposed in our original interpretation. “Magma propagation is governed by a combination of fracture toughness, overpressure, buoyancy, and viscosity⁶³. One explanation for stalling at this depth is that the injected magma volume, and hence the resulting overpressure, may have been insufficient to overcome the increasing lithostatic and edifice-related stresses at shallow depths. Compressive stress from the edifice load Similarly, a reduction in buoyancy due to cooling or crystallization during ascent could have diminished the driving force for further propagation^{64,65}. Alternatively, a dynamic increase in its viscosity, such as due to devolatilisation, may have inhibited further ascent promoted stalling. These factors, acting individually or in combination, may have led to the observed stalling of the dike at ~1.6 km depth, at around the depth of the base of the volcanic island.” We also added a reference to Rivalta et al. (2015) in the Introduction and mentioned fracture toughness as another factor on how melts propagate.

R2-15	Reviewer comment	A timeline in the supplementary material, with the number of stations as a function of time would help tracking the seismicity coverage.
	Author response	Excellent idea.
	Changes made	We have added this station timeline to the Supplementary Material (Fig. S12) and it is cross-referenced in the Methods section.

R2-16	Reviewer comment	Similarly, it would be useful to have the cumulated seismicity as a function of time in Fig. 3b, not only before the intrusion but also after;
	Author response	Agreed.
	Changes made	We have added a new panel to Figure 4 that shows both the daily rate of seismicity, and the cumulative number of earthquakes. We have also added a new Figure 5 that shows similar parameters for a zoomed-in time window focussing on the onset of the seismic swarm.

R2-17	Reviewer comment	The boundary element method and inversion used for analysis of displacement data allows for curvatures. A model with an along dip curvature could connect the deep earthquake swarm to the fracture at shallower depth. Would such a model provide a better or as good fit of the displacement data, according to the Akaike Information Criteria?
	Author response	As already explained in comments R1-7, R-8, and R2-7 above, we explored a number of alternative models (Text S1), including a (1) half-space inversion with two concatenated dykes, and (2) an extended BEM inversion allowing for a vertically curved dike. In both cases, the misfit to the data was substantially worse than our simpler and preferred quadrangular model (Table S2.1). For the double-dike model, the dip of the deeper dyke dips more steeply to the north, but the uncertainties are high - 60-89 degrees).
	Changes made	Please see the changes laid out in our responses to comments R1-7, R-8, and R2-7 above, where we show in detail how we changed the manuscript to encompass these alternative model inversions.
R2-18	Reviewer comment	The inversion parameters should be described in a figure. Parameter "Depth" should perhaps be called "Height"?
	Author response	We acknowledge any potential confusion here.
	Changes made	We have added a new row to Table S1 that explicitly explains the definition of each geometric parameter. We have added the following note to Section 4 of Methods: "See Table S1 for a full description of these geometric parameters" .
R2-19	Reviewer comment	What is the convergence of the inversion solution? This indication is important to show readers how the solution has converged (spread of last computed models, whether iterations stopped because the standard deviated on computed solutions was low enough or because the maximum number of iterations decide at the beginning was reached).
	Author response	Convergence of the inversion was assessed by monitoring the evolution of model misfit and the spread of accepted solutions. The inversion was terminated after 250 iterations, when the model "cost" fell below 750 and was stable for approximately the last 100 iterations, indicating that the solution had stabilized (see new Figs. S9 & S22). The posterior distribution of model parameters (original Figs. S7-S8) show well-constrained values for dike strike, dip, and opening, with broader uncertainty in depth extent. This convergence behavior supports the robustness of the inferred intrusion geometry.
	Changes made	We have added new Figs. S8 & S22 that show the convergence of the cost function through the inversion iterations, for the overall inversion and for the individual geometric parameters, respectively. We have added the following paragraph to Section 4 of Methods: "Convergence of the inversion was assessed by monitoring the evolution of model misfit and the spread of accepted solutions. The inversion was terminated after 250 iterations, when the model cost fell below 750 and was stable for the last 100 iterations, indicating a stable inversion (Figs. S9 & S22). The posterior distribution of model parameters (Figs. S7-S8) show well-constrained values for dike strike, dip, and opening, with broader uncertainty in depth extent. This convergence behavior supports the robustness of the inferred quadrangular intrusion geometry."

R2-20	Reviewer comment	It would also indicate where the best fit solution is located with respect to the search limits.
	Author response	The best-fit solution and the initial search limits were shown in the original Table S1. We acknowledge that a graphical illustration would have been better.
	Changes made	We have added a new figure, Fig. S9, which shows the convergence and stability of each model parameter with respect to the initial search limits.

R2-21	Reviewer comment	The plots showing the 1D and 2D PPD should be shown with better resolution.
	Author response	Agreed.
	Changes made	We have replaced Figs. S11 and S12 with higher-resolution images.

R2-22	Reviewer comment	Considering that the inversion were conducted giving equal weight to the three data set, you could perhaps quote Dumont et al., 2021 who showed the relevance of such an approach.
	Author response	We thank the reviewer for this helpful suggestion. Indeed, we conducted the inversion by assigning equal weight to the three datasets (GNSS and both InSAR orbital passes) to avoid biasing the solution toward the denser InSAR data.
	Changes made	We added the following two sentences to the end of the first paragraph in Section 4 of Methods: “The weights of the three types of observations were normalised in the inversion. This weighting approach prevents biasing the inversion to the denser InSAR data, ensuring that each dataset contributes equally to the model solution (Dumont et al., 2021).”

R2-23	Reviewer comment	The film showing the migration of events is too long (more than five minutes). Perhaps snapshots at irregular time, along with the cumulated seismicity, would be more informative and could be joined in a plot. The colorcode for the events is missing and sometimes the colors reset to red, for an unknown reason.
	Author response	Agreed.
	Changes made	We have fully updated the video.

R2-24	Reviewer comment	The order of the supplementary figures should follow the order of the main text. For instance, on page 4, before the seismicity distribution section, reference to the supplementary material jumps from S5 to S9. There are other such places where supplementary figures should be reordered.
	Author response	Agreed.

	Changes made	We have updated the order of all-cross-references.
--	---------------------	--

R2-25	Reviewer comment	Page 13, line 7: "hydrothermal fluids or melt or volatile..". In this sentence a word (an "if") might be missing "hydrothermal fluids or if melt or volatile..".
	Author response	Agreed.
	Changes made	We added the word "whether" here.

R2-26	Reviewer comment	Page 13, line 9: before the end of the page "observed transtensional environments", a word is missing "observed in transtensional environments"
	Author response	Agreed.
	Changes made	Changed.

R2-27	Reviewer comment	On Fig. 4: In subplot a. the deep swarm is not visible enough (dots are too small and colorcode is different from Fig 4b).
	Author response	Agreed.
	Changes made	We have increased the opacity and edge thickness in Figure 4 (all panels) to emphasise these early events.

R2-27	Reviewer comment	No date color code is indicated in fig 4a. In a, b, and c colorcodes for dates since 03-19-2022 are different. I suggest unifying colorscales of the different subplots for clarity.
	Author response	Agreed.
	Changes made	We have unified the pre- and post-crisis time colourbars and shown the same colourscale for all panels.

R2-27	Reviewer comment	Fig S6, referred to before paragraph 2.3, does not seem to be the right one.
	Author response	Agreed.

	Changes made	We have updated all cross-references.
--	---------------------	---------------------------------------

Reviewer 3

Overall comments

The manuscript presents an analysis of the 2022 magma intrusion event on the island of São Jorge, within the Azores archipelago. This event offers a valuable opportunity to advance our understanding of magmatic processes in tectonically active oceanic settings. By combining InSAR and GNSS ground deformation inversions with seismic analyses of earthquake distributions and source characteristics, the work provides an integrated perspective on both the magmatic and tectonic aspects of the event. I particularly like the analysis of the induced seismicity, and I agree that the patterns highlighted are not typical of seismicity directly induced by a dike. This is a very unusual and interesting observation.

A key point of the manuscript lies in the interpretation of the intrusion as a “leaky fault” phenomenon, with the proposed role of pre-existing fault zones in both enabling and contributing to the arrest of magma ascent. The manuscript is generally well-written and clear in its exposition. The language is precise, and the structure supports the logical flow of the argument.

Overall, this is a valuable and promising manuscript that has the potential to make a significant contribution to our understanding of magma-tectonic interactions in rifted oceanic environments. However, there are two key issues that, if addressed, could significantly strengthen the clarity and impact of the conclusions:

1) Conceptual clarity regarding leaky faults and dikes: At several points, the manuscript appears to conflate or contrast the concepts of leaky faults and “regular” dikes without explicitly defining or distinguishing them. It would greatly benefit the manuscript and advance the debate in the general literature on magma intrusions to clearly articulate the theoretical and observational criteria that separate these two mechanisms and to frame the interpretation of the São Jorge event within this context. A more systematic discussion would help clarify whether the observations more strongly support a leaky fault model, a diking process, or some hybrid scenario such as a sheared intrusion.

2) Missing or incomplete supporting information: Some of the key data and analyses needed to evaluate the proposed mechanism are currently absent or insufficiently detailed in the text, figures, or tables. Addressing these gaps, as detailed below, will improve the transparency and reproducibility of the interpretations.

General author response

We truly appreciate the thoughtful comments of Reviewer 3, which have incentivised us to revisit several aspects of our study. We agree that the seismicity patterns are atypical and deserve to be fully analysed and interpreted in our study. In our response to R3-1 below, we have revisited the concept of a “leaky fault” and have toned-down its use in the manuscript, where it was previously only mentioned in the title. We have fixed the gaps that the reviewer has noticed, and have clarified these points in the individual responses below.

R3-1	Reviewer comment	Conceptual clarity regarding leaky faults and dikes. Some differences between "regular" dikes and faults can be summarized in the following way: "Regular" dikes are hydraulic fractures (Pollard, Gudmundsson, Anderson, Rubin, and many others):  1) Predominantly opening component on the dislocation modelling the deformation (i.e. displacement is roughly perpendicular to the dislocation wall); 2) Orientation roughly coincides with most compressive + intermediate stress axis, perpendicular to least compressive stress axis (usually close, +/-10 degrees) -> strike is parallel to strike of local normal faulting mechanisms 3) Dikes are usually formed without the need of any pre-existing structure, i.e. the rock is actively fractured at the intrusion tip due to the extremely intense stress concentration forming there due to the dilation of the intrusion walls. Leaky faults (my take on this): I understand it is a tectonic fault zone that allows magma or fluids to ascend through it, effectively leaking material from depth toward the surface or upper crust. I understand a leaky fault is not a dike, though it may host magma like a dike would. Rather, magma exploits a pre-existing fault or fracture, rather than creating its own new pathway (like a dike does). The intrusion occurs on the fault plane, rather than perpendicular to the least principal stress, which is typical for dikes. I would expect:  1) Predominantly shearing component on the dislocation modelling the deformation (i.e. displacement is predominantly parallel to the dislocation wall). This is because usually faults have accumulated shear stress on them and as soon as they are lubricated by any fluid intruding them, they would also shear as well as open to accommodate the magma; 2) Orientation strikes between most compressive and least compressive stress axis (usually 30 to degrees) (Anderson's theory of faulting) 3) Slip usually occurs mostly on pre-existing faults or on pre-existing fault segments potentially linked by brief segments of pure opening. Since the main point of the article is clarify the specific mechanism behind the intrusion, I suggest to be really clear about what is intended every time a dike is mentioned. For example, in the introduction, references 5 to 8 (as well as some others further below) on dike intrusions are listed as studies on magma leaking through pre-existing faults. These are incorrectly listed citations.
	Author response	We thank the reviewer for this detailed comment, which we have read and tried to understand. This is a similar comment to R2-8. In our view, "regular dikes" and "leaky faults" are not mutually exclusive, certainly from a kinematic perspective. In addition, we disagree that "regular" dikes do not form exclusively without breaking new structure - this likely depends on magma overpressure relative to the rheology of the surrounding rock (e.g., Rivalta et al., 2015). In the original manuscript, we were referring to the zone of seismicity as being the "leaky" component. Nevertheless, we understand the potential conceptual confusion with the ascent of the main dike, especially from mantle depths.
	Changes made	To prevent further confusion, we have slightly changed the title of the manuscript to: "Mantle-rooted faultLeaky faults modulated magma ascent and seismicity during the 2022 São Jorge (Azores) volcanic unrest"

R3-2	Reviewer comment	As for the São Jorge event, is the evidence supporting a "regular" dike or magma intruding into a pre-existing fault (these two are very different concepts, as explained in point a, and cannot be conflated)? It is not very clear from the manuscript. The inversion results in the text suggest pure opening, as no shearing component of the displacement is mentioned anywhere (or at least I could not find it). The method used is declared "3D mixed BEM". Does this mean that a shearing component on the dislocations is allowed and used? If the argument is that of a leaky fault, this possibility should at the very least be explored. As it is now, the inversion seems to assume a purely opening dike, contradicting one of the main discussion arguments of the manuscript. Moreover, the intrusion is parallel to the strike of the closest normal faulting events, it is therefore roughly perpendicular to the axis of least compression. The authors do not list any evidence of a vertical pre-existing fault rooted at 25 km depth (the fractures mapped on the island are all broken and can be hardly linked to a consistent major strike slip fault that could leak a fluid as viscous as magma). Based on the evidence presented, the "regular" dike interpretation would seem far more logical. I may have misunderstood some of the assumptions or evidence: please clarify.
	Author response	This comment is very similar to R1-6, R2-7, R3-5. We inverted for a model that included a shearing component, rather than just pure opening (see new Text S1), but we found that such an alternative configuration failed to provide an improved fit to both GNSS and InSAR data (new Table S2.1), while remaining physically plausible. Our sub-vertical quadrangular dike model emerged as the preferred solution, as it is the only configuration that reconciles all datasets with small residuals and physically plausible parameters.
	Changes made	We included the following new paragraph in the Discussion: “Another remaining issue is that predicted horizontal GNSS vectors from our preferred quadrangular dike model at stations on islands to the south (AZTP and PIED on Pico; HORT on Faial) show a consistent eastward trend (Fig. S10), which could indicate a left-lateral shear source. This trend is particularly fascinating given the rotated left-lateral slip of the swarm seismicity. Therefore, we inverted the geodetic data for a sheared dike model (Text S1.1), but found that the resulting model unsatisfactorily increases the GNSS and InSAR residuals (Table S2), and results in a physically implausible near-horizontal shear dike located too far north of the seismicity. The observed left-lateral shear residual may reflect transient stress perturbations during magma emplacement, possibly linked to shear dilatancy effects or local stress rotations, that may be physically linked to the rotated focal mechanisms. Alternatively, the shear residual may be due to the differential timescale of opening between the mantle and crustal intrusion segments, as described above. However, the sparse GNSS dataset cannot provide confident constraints on such sources.”

R3-3	Reviewer comment	Incomplete information:  1. In the ground deformation section the maximum opening is listed but not the average opening or, alternatively, the total volume. The total volume (or the average opening, which would allow me to infer the volume) is an important parameter and should be included. Was the pressure in the BEM model inverted for or fixed? This is not clear. Was a simple rectangular dislocation model used first to fit the data, or only the BEM model? I believe it is always useful to include the simplest model possible, beside the complex model. 2. Please add the vertical GNSS displacements to Fig. S9. Without them, I am not able to check the calculations nor it is possible to the reader to reproduce the results. In the caption of fig. S9, please add the time interval to which the displacements refer to (is it 15 to 21 March 2022 as mentioned in the main text at the bottom of pag. 18?). "We used all three component GNSS vectors" -> "We used all three components of the GNSS vectors"? 3. The resolution of some of the figures in the supplement is far too low to read them, please increase resolution. 4. The inversion procedure is described well but please include in the table caption a definition for the parameters inverted for. For example, what is Bot. length? The length of the dike quadrangle at its bottom? What is shear angle?? How much is the total volume?
	Author response	 • We solve for the opening at each point to explicitly show the average opening and total volume. The total volume of the dike is $7.9 \times 10^7 \text{ m}^3$. • Geometrical parameters are first inverted in a nonlinear (Bayesian) inversion. Once these parameters are fixed, pressure, and hence opening, is linearly inverted, so it is not a fixed, nor a free parameter during the nonlinear inversion. This was already described in the second paragraph of Section 4 of the Methods: "First, the inversion explores nonlinear geometry parameters followed by the solution of the opening which satisfies constant overpressure, by linear inversion. Note that a single magma overpressure parameter controls the dike opening pattern." • We considered simple Okada type models in our exploration of the solution space (see new Text S1). • The bottom length of the dike is indeed the horizontal length of the base of the quadrangle. The shear angle describes the "leaning" of the quadrangle, and can be formally defined as an angle measured from the perpendicular line to the strike direction, to the line that connects the middle points of the top and bottom side of the quadrangle (line of steepest descent) along the plane.
	Changes made	 • We have included the total volume of the dike in the third paragraph of Section 2.1. • We have added a second panel to Fig. S9, which shows the observed and modelled vertical GNSS vectors. • We changed the quoted text in Point 2. • We have replaced Figs. S11 and S12 with higher-resolution versions. • We have included an extra row to Table S1 which defines all of the geometric parameters of the nonlinear inversion.

R3-4	Reviewer comment	The bottom of the intrusion is very deep. Please state what specific data, among those used, require this very unusual bottom depth. Please discuss how robust this is.
	Author response	This comment is similar to comment R1-6 . A deep dike source is needed to fit the far-field displacements, while the shallow top-depth fits the InSAR data and GNSS stations on São Jorge. However, the down-dip width of the dike is one of the most uncertain parameters.
	Changes made	We have added the following text to the third paragraph of Section 2.1: “The inversion yields a bottom depth of the dike at ~26 below sea level (bsl), within the lithospheric mantle. This vertically extensive nature of the modelled dike is required by the far-field GNSS stations on adjacent islands; however, the exact depth is one of the most uncertain aspects of the model (formal uncertainty of ± 8 km; Fig. S11). with the best-fit model at the deeper end of the ensemble solutions, but most models reaching beyond 20 km depth bsl (Fig. 4a)”.
R3-5	Reviewer comment	The southern GNSS horizontal displacements are not really well-fit, fig S9. Clearly the data are not sufficient for a strong discussion on this, but what is your interpretation of this misfit?
	Author response	This is a very similar comment to R1-6 and R2-7 . Because the dike intruded obliquely to the regional minimum compressive stress direction, we explored the possibility of coeval shear displacement—i.e., a sheared intrusion. In our preferred model, we assume a purely opening-mode (tensile) dike, which provides a good fit to both GNSS and InSAR data. However, the GNSS far-field residuals show a consistent eastward misfit at southern stations (Fig. S9), which could be indicative of an additional deep left-lateral shear component (not colocated with dike intrusion, neither in x-y position along the strike of the island or depth certainly not shallow). This is especially intriguing given that the inferred shear sense is more consistent with the observed focal mechanisms during the seismic swarm, and is opposite to the long-term tectonic motion along the PdC-FZ, which is right-lateral. To address this point, we have carried an inversion for a sheared dike model (see Text S1.1-1.4). We find that the inverted model unsatisfactorily increases the GNSS and InSAR residuals (Table S2.1), and results in an implausible near-horizontally dipping shear dike located too far north of the seismicity. We speculate that the observed left-lateral shear residual may reflect transient stress perturbations during magma emplacement, possibly linked to shear dilatancy effects or local stress rotations, that may be physically linked to the rotated focal mechanisms. However, the sparse GNSS dataset cannot confidently constrain such a source. Such mechanisms should be the focus of a future study.
	Changes made	We have added the following paragraph to the penultimate part of the Discussion: “One remaining issue is that predicted horizontal GNSS vectors from our preferred quadrangular diking model at stations on islands to the south (AZTP and PIED on Pico; HORT on Faial) show a consistent eastward trend (Fig. S9), which could indicate a left-lateral shear source. This trend is particularly interesting given the rotated left-lateral slip of the swarm seismicity. Therefore, we inverted the geodetic data for a sheared dike model (Text S1.1), but found that the resulting model unsatisfactorily increases the GNSS and InSAR residuals (Table S2.1), and results in an implausible near-horizontally dipping shear dike located too far north of the seismicity. We speculate, therefore, that the observed left-lateral shear residual may reflect transient stress perturbations during magma emplacement, possibly linked to shear dilatancy effects or local stress rotations, that may be physically linked to the rotated focal mechanisms. However, the sparse GNSS dataset cannot confidently constrain such a source.”

Response to Reviewers (2nd round)

Author response

Many thanks for handling our manuscript for a second round of review. We have responded point by point to all remaining minor concerns from the reviewers, and we hope all parties find these satisfactory.

Reviewer 2 (Valerie Cayol)

Overall comments

I think overall the authors have addressed the comment appropriately. Thank you for the very clear and detailed answer provided to all reviewers. The different points of the answer were numbered. I will address the answer to my comments in the order they were answered.

Author response

We're really glad you have approved our changes - thank you very much for taking the time to read our manuscript once again.

R2-1	Reviewer comment	Ref: original R2-1. The discussion on the left lateral mechanism of micro-seismicity in a context of right lateral long term slip is providing some more details. I agree, that we need to assume that the background stress is weak. The more convincing element is provided by the Coulomb stress changes.
	Author response	We're glad you approve of our new Discussion, which mentions the potentially weak background stress. The change in Coulomb stress field will naturally be somewhat uncertain, particularly in the area of early shallow seismicity, because it depends on the geodetic model.
	Changes made	N/A

R2-2	Reviewer comment	Ref: original R2-2. Thank you for clarifying the origin of the pre-stress on the PdC-FZ. However, I still do not understand why the pre-stress, as indicated in the introduction (l. 110-111) is for right lateral displacement, when failure took place in left lateral mode. For failure to take place for a given failure mechanism, the pre-stress has to be in the same direction. A possibility is that anterior slip (like that associated to a M 7 EQ) released background stress and that the Coulomb stress change induced by the dike, associated to the fluid release, was sufficient to lead to failure. This would mean that the recent seismic activity did not pre-load but unloaded the fault, if this previous failure was west of the dike. A Mohr circle, given in the appendix, would perhaps help understanding how fluids may have brought a weakly loaded fault closer to failure.
	Author response	The regional tectonic stress field favours right-lateral slip, which is also consistent with the geological offsets observed along the PdC-FZ. This is also consistent with the longer-term background seismicity, with mechanisms determined from the Global-CMT catalogue, which shows a propensity for NE-SW oriented T-axes, which are 90 degrees rotated to the 2022 Sao Jorge events:  We disagree that pre-stress has to be in the same direction as subsequent slip. Observations of so-called anti-repeating earthquakes are becoming more commonplace in a variety of settings, including in magmatic environments and during swarm sequences associated with intrusions (e.g., Cesca et al., 2024, Nature. Comms., doi: 10.1038/s43247-024-01290-1). Failure occurs in the opposite direction to the pre-existing stress because of the local stress perturbation introduced by the intruding dike. However, we acknowledge that models of antirepeating earthquakes need further development. In future work, we would like to investigate further the long-term background stress on Sao Jorge and the mechanisms of microseismicity.
	Changes made	We have modified slightly the fourth paragraph of the Introduction to make this point clear: structures and magmatism²¹ (Fig. 1a). The Azores is shaped by the diffuse boundary between the Eurasian and Nubian plates, with the Terceira Rift (Fig. 1a) as the main spreading system, where slow (~4.5 mm/yr), WSW-ENE-oriented relative motion²²⁻²⁵, produces right-lateral transtension²⁶, which is consistent with the mechanisms of past earthquakes in the region²⁷⁻²⁹, exhibiting showing rift-parallel normal faulting and off-rift strike-slip faulting (Fig. 1a). Deformation across the Azores is

R2-4	Reviewer comment	Ref: original R2-4: perfect answer, but Fig S19 has replace S14.
	Author response	Good catch - thanks.
	Changes made	We have updated the cross-references to the Coulomb stress change figures.
R2-6	Reviewer comment	Ref: original R2-6: I do not understand the answer. I think the fact that M4.7 EQ were released make the release of regional tectonically accumulated right-lateral stress less likely.
	Author response	From a Coulomb stress change viewpoint, left-lateral slip along the PdC-FZ further loads right-lateral slip along the same patch of the fault zone. Moreover a cumulative moment equivalent to Mw4.7 is still only a small fraction of the total seismic moment that would be released in a M~7 earthquake, which this fault is capable of, based on paleoseismology results. Therefore, we have opted to keep this phrasing in the manuscript.
	Changes made	None.

R2-7	Reviewer comment	Ref: original R2-7: My comment on the possibility of coeval sheared intrusion to better fit distant E-W displacements is addressed, but it would have been more satisfactory to conduct the shear stress inversion with the same dike or a dike very close as the best-fit solution determined by the boundary element inversion (DefVolc) or by the dislocation solutions (GBIS). Using GBIS Model, a horizontal source, very different from the best fitting solution is determined. This indicates that the parameter space is probably too large and not sample enough. Consequently, the data fit is decreased with respect to both the simple rectangular solution (GBIS) or the more complex quadrangle (DefVolc). Because the discrepancy between the data and the DefVolc best fit model is small, it would have been better to search a best fit model in the vicinity of the model already determined, somehow using it as an a-priori for the search. This approach could have been conducted with both the GBIS or the DefVolc inversions.
-------------	-------------------------	---

Author response

We appreciate the reviewer's comment. Our intention when exploring a broader parameter space was to evaluate whether alternative geometries could explain the observed E–W displacements without imposing strong prior constraints. Our results allowed us to test the robustness of the shear stress inversion independently of the previous boundary element and dislocation models inversions. We acknowledge that the GBIS inversion identified a horizontal source that differs from the DefVolc best-fit solution, reflecting the trade-off between model complexity and parameter sampling. However, the discrepancy in misfit between the DefVolc solution and the shear inversion is small, and the overall interpretation remains consistent: the dominant deformation is controlled by the main dike intrusion, with only minor contributions from shear components.

We run an additional Bayesian inversion for a sheared dyke solution using a modified version of the GBIS software to accommodate this type of source. The inversion was done over a narrower set of parameters (from the posterior bounds for a single dyke). The figure below shows the posterior PDFs. Importantly, the strike-slip component is shown to be unstable over the range of -10 to 10 m of slip, which is unrealistically high. Dip-slip was set to almost zero (SDIKE DipSlip).

The misfit for this inversion was larger (see figures below) than a single dyke. In summary, from this additional inversion, we can conclude that if shearing occurred during the most intense period of the unrest, this was not collocated with the dyke opening.

“Best-fit” sheared-dyke GNSS prediction.

		  “Best-fit” sheared-dyke Ascending interferogram. As we can see, the fit with the observations is not improved, with respect to our preferred single inclined dyke from the main text.
	Changes made	None
R2-8	Reviewer comment	Ref: original R2-8: Leaky faults. A. I think the new title is more appropriate, as the evidences in favor of a fluid leak are too weak. B. I fully agree with the new conclusion, moreover fluid migration from the dike provides an explanation for the micro EQ being triggered at low stress change (see reply to comment R2.2). C. I wonder if there is an error in the interpretation in Fig 7. Indeed Fig 4 shows a North dipping dike, when Fig. 7 shows a south dipping dike. D. The new text version and Fig. 8 comparing b-value in different context brings a factual element to the possible role of fluids in the seismicity trigger. E. Could you unify the cluster names ? There might not be a need for additional lineation names as lineations are indicated by clusters. In the text, clusters are called cluster 2, ..., 18. On figures 4 and 6, letters appear (LL, UL), as well as a mix of letters and numbers (Fig. 6). This is slightly confusing. Moreover the numbers (2, 7 and 18) described in the text are not found in any of these figures.
	Author response	A. Great that you agree with the new title. B. Good point. C. Fig. 7 is meant to show a near-vertically dipping dike. Possibly the 3-D perspective view has caused some confusion. D. Thanks for the positive comment. E. We have decided to keep the cluster numbering system since these are decided based on a hierarchical order (larger number of events first) based on the DBScan clustering algorithm. Clusters 2, 7 and 18 are labelled in Fig. 6.
	Changes made	B. We have modified slightly a sentence in paragraph 5 of the Discussion: “Such asymmetrically localised seismicity is thus likely due to the fluid-triggered seismicity, with due to the PdC-FZ providing a direct hydraulic connection between the westward-tilting intruded dike and earthquakes, allowing fluids to trigger seismicity directly.” C. To prevent any confusion, we have explicitly stated in the caption of Fig. 7 that the dike is “near-vertical”.

R2-11	Reviewer comment	Ref original R2-13: Fig 6. There are inconsistencies between the numbering of the cluster in the figure, in the caption and in the text.
	Author response	Agreed - many thanks for spotting this (also noticed by Reviewer 3).
	Changes made	Changed.

R2-14	Reviewer comment	Ref original R2-16: I do not see the new panel with the daily rate of seismicity in Fig 4.
	Author response	Apologies, we may not have given the correct figure number in our response. Seismicity rates are shown in Figs. 3b and 5a.
	Changes made	None.

R2-15	Reviewer comment	Ref original R2-17: R2.17: Two dikes models have many more parameters (12 parameters + 4 position parameters = 16 parameters) than a single dike with a curvature (10 parameters). Hence, such models may lead to local minima, which might explain why the 2 dikes model did not give any satisfactory results. The parameter space might have been again too large, as two shallow sub-horizontal dikes were determined which fit the two deformation patterns, which is unlikely.
	Author response	We thank the reviewer for highlighting the complexity introduced by the two-dike model. We agree that the large number of free parameters (16 in total) significantly increases the risk of local minima and reduces the reliability of the inversion, particularly when the deformation signal is dominated by a single primary intrusion. Our intention when testing this configuration was, again, exploratory, to assess whether such alternative geometries could capture the distributed deformation patterns; not to prove it, but to show evidence that we fully explored the possible model geometries - as suggested by Reviewer #1 in the first revision round. The results (two separated shallow dikes) suggest that the parameter space was too broad to yield physically plausible solutions. The lack of satisfactory results of the extensive magmatic geodetic source exploration supports the interpretation that a single dike provides a more compact, parsimonious and robust explanation of the geodetic observations.
	Changes made	None

Reviewer 4 (Þorbjörg Ágústsdóttir)

Overall comments

I enjoyed reading the extensive paper of Stephen P. Hicks and Pablo J. Gonzalez et al. "Faults modulate magma propagation and triggered seismicity: the 2022 São Jorge (Azores) volcanic unrest". It is likely to further our understanding of magmatic and tectonic interaction, and it has an impressive multidisciplinary approach. The paper has already undergone insightful reviews by three reviewers, that have clearly helped improve the manuscript. I recommend it to be published in Nature Communications after some minor reviews.

Author response

Thanks so much for your positive view of our paper and for taking the time to read it with a fresh pair of eyes. We have made changes in light of your minor comments listed below.

R4-1	Reviewer comment	Abstract and text: ultra is a bit extreme... "ultra-high-precision hypocentre locations"
	Author response	Agreed
	Changes made	Removed "ultra".
R4-2	Reviewer comment	Fig.1 colour scheme - not clear contrast to the background, especially the brown PM circles, the purple 3K and the grey 4U. For the seismicity (orange circles), I recommend choosing a brighter colour.
	Author response	Good idea.
	Changes made	We've now changed some of the symbol colours in Figs. 1a and 1b.
R4-3	Reviewer comment	Fig. 2: perhaps you could put N-arrow on a) as well?
	Author response	Agreed.
	Changes made	We have added an arrow showing the look direction (WNW) in Fig. 2a.
R4-4	Reviewer comment	Fig2 - caption: Dotted orange lines highlight the vents of past fissure eruptions – add past to sentence.
	Author response	Good catch - agreed.
	Changes made	Added "done".
R4-5	Reviewer comment	Fig 3: Note that the magnitudes for the first D1 cluster seems to be missing from Fig 3c.
	Author response	Good spot. But not all events had a good enough signal-to-noise ratio to determine their magnitudes, which is why a few have no magnitude assigned.

	Changes made	None
R4-6	Reviewer comment	Fig 4a: purple dyke in rose diagram is nearly invisible (unless I zoom in a lot on the figure)– could you choose another/brighter colour? Also, add to figure caption for a) about the quadrangular dyke-opening solution. cross-sections – could you label Moho?
	Author response	These are all excellent comments.
	Changes made	 - Instead of showing the dike, SHmax, and fault directions as shaded lines, we have switched to labelled arrows around the edge of the plot. - We added the following sentence to the caption: “The red-orange shaded area is our geodetic quadrangular dike-opening model.” - We added a horizontal grey striped zone to (b) indicating the approximate area of the Moho, based on the 1-D P-wave velocity model.
R4-7	Reviewer comment	Section 2.2 (P8). On uncertainty: What is the uncertainty of the manually picked events?
	Author response	We are unsure what uncertainty is being referred to here, especially as we didn't manually pick events. Phase picks were made with PhaseNet, which doesn't give pick errors, so we used a nominal uncertainty value of 0.1 s.
	Changes made	None.
R4-8	Reviewer comment	Section 2.2 (P8). Deep precursory seismicity p.6: “bsl south of São Jorge (Figs. 3b-d & 4).” delete extra .
	Author response	Agreed.
	Changes made	Done.
R4-9	Reviewer comment	Section 2.2 (P8).Regarding the D1-3: how many stations were they detected on? Are they LF events or VT?
	Author response	The largest event of the precursory sequence, with ML 2.6, was detected on nine stations. We computed frequency indices (FI) for the entire catalogue, but found no clear evidence of substantially lower frequency content in the deeper precursory events. More generally, we prefer not to use FI values since deeper events are likely to have reduced high frequency content due to attenuation over longer path lengths.
	Changes made	We included the number of stations to this paragraph: “ The largest of these earthquakes has a moment magnitude (M_w) of 2.6, which was detected on nine stations. ”
R4-10	Reviewer comment	Section 2.2 (P8). First 24 hours of the seismic swarm p6: change accelerating seismicity rat,. to accelerating seismicity rate.
	Author response	Good spot - thanks.
	Changes made	Fixed

R4-11	Reviewer comment	Section 2.2 (P8). The median depth uncertainty of all events in the catalogue is 46 m. Add “relative depth”
	Author response	Agreed.
	Changes made	Added.
R4-12	Reviewer comment	Section 2.2 (P8). ~1702 UTC on 19 March 2022 - “ colon: is missing”
	Author response	Agreed.
	Changes made	Done.
R4-13	Reviewer comment	Section 2.2 (P8). “the first locatable eq “ is misleading here – perhaps add “in this shallowing sequence / or add time stamp for clarification“
	Author response	Agreed.
	Changes made	We have added to this sentence: “ The first locatable earthquake in the swarm occurred at ... ”
R4-14	Reviewer comment	Fig 5d) is seem to be more like 5 km downwards – but perhaps I’m misunderstanding – pls clarify the text. And 5c) 10 km west rather than 8km?
	Author response	Agreed.
	Changes made	We have changed to “5 km downwards, and 10 km westwards”.
R4-15	Reviewer comment	The movie is great!
	Author response	Excellent - glad you like it!
	Changes made	N/A
R4-16	Reviewer comment	“These are separated by a ~2 km-wide aseismic region (“AS” in Fig. 5d).” : From Fig6a seems more like 1.5 km wide and 1.5 km long??
	Author response	Agreed.
	Changes made	Changed to “~1.5 km wide”

R4-17	Reviewer comment	P8 - "Well-constrained focal mechanisms (see Methods; Fig. S16) show strike-slip faulting, with one nodal plane parallel to the main lineation of seismicity and the PdCFZ, indicating left-lateral faulting (Fig. 4a)." : In the discussions there is no mention of the focal mechanisms, if they're similar or not to other dyke intrusions. I suggest adding a couple of sentences and references there of.
	Author response	The rotated focal mechanisms were mentioned, albeit too briefly, in Paragraphs 7 and 9 of the Discussion.
	Changes made	We have elaborated upon the text in Paragraph 7 of the Discussion so it now reads: the dike to stall ~1.6 km below the surface. The resulting static stress change led to the observed focal mechanisms that indicate left-lateral seismogenic shear along the PdC-FZ, which are rotated by 90° with respect to the long-term background right-lateral motion ^{21,22,26,30-33,35,44} (Figs. 1a, 2, 4a). Such stress rotations due to dike intrusions have been reported before, with a possible mechanism of shear dilatancy effects from an overpressured magma in a relatively weak background stress field. may have also contributed to the focal mechanism rotation ^{13,15,77,84} .
R4-18	Reviewer comment	Fig.5. Pls add time stamp to the top of a) for clarity
	Author response	Agreed.
	Changes made	We have added tick labels to the upper x-axis of the top subplot.
R4-19	Reviewer comment	Fig. S6 Caption say the blue dashed lines – they look purple to me on the figure.
	Author response	I checked the plotting script, and it turns out they are indigo-coloured lines, so a mix of blue and purple!
	Changes made	Change "blue" to "indigo" in the caption.
R4-20	Reviewer comment	Fig S13: Pls specify the bin width
	Author response	Agreed.
	Changes made	Added " Magnitudes are binned in 0.1 units " to the caption.
R4-21	Reviewer comment	Fig S16: for clarity, remove the N-S E-W automatic fault planes by MTfit's plotting routine.
	Author response	Agreed.
	Changes made	We've updated Fig. S16.

R4-22	Reviewer comment	Fig S18 caption: The grey shading shows the range of inverted models that have an RMS misfit within 5% of the best-fitting model – can you add the software used to the sentence? I'm unsure from the methods if this is 5% range of the 1D velocity model.
	Author response	We used a brute-force Monte-Carlo approach to randomly vary the prior velocity model, yielding a range of models inverted with the VELEST software that have a given RMS misfit relative to the best-fitting velocity model.
	Changes made	We have modified this part of the caption so it now reads: " We randomly generated a suite of 2,000 starting models, which were then inverted using the VELEST software. The grey shading shows the range of inverted models with an RMS misfit within 5% of the best-fitting model. "
R4-23	Reviewer comment	P11 - "we infer that PM.PMAN sensitivity kernel includes the geodetically imaged main dike, whereas PM.ROSA is too far west of the dike": - perhaps add: main dike onset/ascent whereases
	Author response	Agreed.
	Changes made	We have changed this sentence to: " These two stations are ~16 km apart, and given the waveform frequency and inferred depth of subsurface changes, we infer that PM.PMAN sensitivity kernel includes the geodetically-imaged main dike that ascended vertically, whereas PM.ROSA is too far west of the dike (Fig. 4). "
R4-24	Reviewer comment	P13 - "The deep (> 9 km) and shallow (< 5 km) portions of melt ascent were aseismic.": Perhaps you should add *initial* melt ascent or something like that and reference Fig.4 bluish dots – otherwise it can be confusing.
	Author response	Agreed.
	Changes made	We have added a cross-reference to Fig. 4 here.
R4-25	Reviewer comment	P13 - "...these shallow earthquakes and tremor-like bursts likely mark the flanks of the ascending dike." Pls add reference to Fig 5 c) for the tremor bursts.
	Author response	Agreed.
	Changes made	Done.
R4-26	Reviewer comment	P14 - "The dike intruded parallel to the PdC-FZ and VT seismicity zone, east of the surface-mapped fault zone, where the fault is likely buried by lavas and scoria Cones". I feel like there is something missing in this sentence. "VT seismicity zone" is mentioned here for the first time here – I assume it refers to the redish seismicity in the PdC-FZ, but that seismicity surely reflects a dyke intrusion at depth? So perhaps you can rephrase, to something like highlighting a narrow zone of seismicity.
	Author response	Agreed.
	Changes made	We have now changed this sentence to: The dike intruded parallel to the PdC-FZ and main zone of VT seismicity streaks and lineations at ~8-12 km depth-zone , east of the surface-mapped fault zone, where the fault is likely buried by lavas and scoria cones (Fig. 4). The fault therefore provides a direct structural and hydraulic

R4-27	Reviewer comment	P14 - This comment is of large significance: “the 2022 São Jorge seismic-volcanic unrest shows seismicity confined to one flank of the dike with no activity within or to the east of the dike.” Here the dike refers to the geodetically modelled dyke – please state that clearly in the text wherever that is the case – as there often a discrepancy between dikes constrained by geodetic models or seismicity.
	Author response	Agreed.
	Changes made	We have now changed this sentence to: connection between the dike and seismicity. Unlike typical dike intrusions, where seismicity surrounds the dike in a ‘dogbone’ pattern ^{16,60-67} , the 2022 São Jorge seismic-volcanic unrest shows the main zone of seismicity confined to only one flank of the geodetically-modelled dike with no activity within or to the east of the dike. This pattern highlights a discrepancy between potential interpretations that would be made using either seismicity or geodesy alone during volcanic unrest.
R4-28	Reviewer comment	P17 - “ultra-thin lineaments” : change to very? thin lineaments – ultra is quite extreme
	Author response	Agreed.
	Changes made	Changed to “very thin”.
R4-29	Reviewer comment	P17 - “may indicate that, following the main dike ascending to the base of the island edifice, a laterally propagating branch of magmatic fluids ^{71,79,80} may have initiated at the western edge of the vertically-stalled intrusion, at 9-10 km depth ⁶ ” - this explanation is quite confusing – I’m unsure of what it means – do you perhaps mean that there is a magma pocket in this area left over from the initial ascent?
	Author response	This a good point. Ultimately, because of the rapid nature of the dike ascent, we cannot constrain the timing and dynamics of a potential lateral melt branch, and whether it was synchronous with the ascent, or whether the branch occurred as the main dike stalled.
	Changes made	We have re-read this section, and it explains our reasoning, but we have made a few minor changes to improve clarity: rapid westward and downward migration of seismicity (Fig. 4), which left a ~2 km thick aseismic zone (‘AS’) between three seismic lineations (‘UL’, ‘LL’, and ‘DL’; Fig. 4), may indicate that, following the main dike ascending to the base of the island edifice, a laterally propagating branch of magmatic fluids ^{71,79,80} may have then initiated at the western edge of the vertically-stalled intrusion, at 9-10 km depth ^{65,73,74} Forward modelling (see Methods ; Fig. S21), however, shows that such melt branches

R4-30	Reviewer comment	P17 - “suggesting long-lived upward fluid migration along propagating fractures ⁷⁵ in the PdC-FZ” Are you suggesting propagation of fluids along faults or fault-propagation?
	Author response	This is a good point, and we were referring to propagating fluids rather than fractures per se. Nevertheless, there is no reason why intense swarms of fluid-driven seismicity could not represent hydraulic fracturing of crustal rock.
	Changes made	We have changed this sentence to: (Fig. 7), or a combination of both lateral and vertical transport. Some seismicity clusters, such as Clusters 2, 7, and 18 (Fig. 6; Movie S1), show complex fracture-like geometries and an upward migrating front over weeks to months with slow diffusivity (~0.001-0.002 m ² /s), suggesting long-lived upward fluid migration along propagating fractures ⁷⁵ in the PdC-FZ.
R4-31	Reviewer comment	P17 - The PdC-FZ likely facilitated fluid channelling ⁸¹ through the lowermost crust near the crust-mantle boundary.” – can you add the value in brackets last?
	Author response	Agreed - we have added the estimated Moho depth.
	Changes made	Changed to: of melt, and magma devolatilisation ⁷⁷⁻⁸⁰ , triggered the main seismicity. The PdC-FZ likely facilitated fluid channelling ⁸¹ through the lowermost crust near the crust-mantle boundary at ~12 km. Lateral
R4-32	Reviewer comment	P19 - “Our results suggest that pre-existing faults can have opposing effects on magma propagation.” Can you build under this sentence earlier in the discussion – it comes a bit out of the blue, as you mention the fault zone multiple times as facilitating the intrusion.
	Author response	Some context and concepts of how faults may impact magma propagation were already explained in Paragraph 2 of the Introduction. Nevertheless, we agree that this sentence in the Discussion needed more clarity.
	Changes made	We have changed this sentence in the Discussion to: Integrating our geodetically-modelled dike intrusion, the location of our relocated seismicity to one side of the dike, and our knowledge of fault structure on São Jorge, Our results we suggest that pre-existing faults can have opposing effects on magma propagation. The PdC-FZ facilitated rapid
R4-33	Reviewer comment	Fig. 8 : Can you add the Moho depth as a number somewhere in the text? Then the discussions and the dimension of Fig8 will be clearer.
	Author response	Agreed.
	Changes made	We’ve added an annotation to the figure highlighting the Moho and giving its depth.

R4-34	Reviewer comment	Methods: Did you try using QM for the entire period? How reliable is EQ transformer in the first days with few stations?																																	
	Author response	This is an interesting point raised. We carried out a test using the detection, picking, and association workflow of QuakeMigrate for the first two days of the seismic swarm, but it found substantially fewer events than the PhaseNet + PyOcto workflow. Therefore, our original workflow is more successful.  The graph plots the cumulative number of events (Y-axis, 0 to 800) against time (X-axis, from 2022-03-19 18:00 to 2022-03-20 21:00). Two data series are shown: 'PhaseNet picks + PyOcto association (this study)' (blue line) and 'QuakeMigrate' (orange line). The blue line shows a steady increase, reaching approximately 800 events by the end of the period. The orange line shows a much slower increase, reaching only about 250 events by the end of the period.  <caption>Approximate data points from the cumulative event count graph</caption>   Date PhaseNet picks + PyOcto association (this study) QuakeMigrate     2022-03-19 18:00 ~20 ~10   2022-03-19 21:00 ~100 ~30   2022-03-20 00:00 ~150 ~40   2022-03-20 03:00 ~200 ~50   2022-03-20 06:00 ~300 ~60   2022-03-20 09:00 ~350 ~80   2022-03-20 12:00 ~400 ~100   2022-03-20 15:00 ~450 ~150   2022-03-20 18:00 ~550 ~200   2022-03-20 21:00 ~800 ~250   	Date	PhaseNet picks + PyOcto association (this study)	QuakeMigrate	2022-03-19 18:00	~20	~10	2022-03-19 21:00	~100	~30	2022-03-20 00:00	~150	~40	2022-03-20 03:00	~200	~50	2022-03-20 06:00	~300	~60	2022-03-20 09:00	~350	~80	2022-03-20 12:00	~400	~100	2022-03-20 15:00	~450	~150	2022-03-20 18:00	~550	~200	2022-03-20 21:00	~800	~250
Date	PhaseNet picks + PyOcto association (this study)	QuakeMigrate																																	
2022-03-19 18:00	~20	~10																																	
2022-03-19 21:00	~100	~30																																	
2022-03-20 00:00	~150	~40																																	
2022-03-20 03:00	~200	~50																																	
2022-03-20 06:00	~300	~60																																	
2022-03-20 09:00	~350	~80																																	
2022-03-20 12:00	~400	~100																																	
2022-03-20 15:00	~450	~150																																	
2022-03-20 18:00	~550	~200																																	
2022-03-20 21:00	~800	~250																																	
	Changes made	None																																	

Reviewer 5 (Luigi Passarelli)

Overall comments

I have just finished reading the paper, "Faults modulate magma propagation and trigger seismicity: the 2022 São Jorge (Azores) volcanic unrest," by Hicks and coauthors. The manuscript has already undergone its first round of revision, and as requested by the editor, I have carefully reviewed this new version with particular focus on the comments from Referee#1.

The manuscript addresses the seismic crisis and volcanic unrest on São Jorge Island (Azores) using an impressive multidisciplinary approach incorporating seismic and geodetic data. Following careful analysis, the authors propose a stealthy dike intrusion as the main driver of the seismic unrest. This work is outstanding for several reasons: 1) The impressive seismic catalog retrieved via state-of-the-art machine learning techniques, which allowed for a careful analysis of magma-fault interaction. 2) The number of geodetic models employed to constrain the deformation source, even despite sparse data coverage. 3) The proposed very fast dike intrusion that propagated within a fault zone and eventually stalled beneath the volcanic edifice—a rare case observed with modern instrumentation.

I have carefully reviewed the rebuttal letter, focusing especially on the excellent comments provided by Reviewer #1. I believe the authors have addressed all points in a punctual and satisfactory manner, including the required additional analysis. This has resulted in a much stronger manuscript compared to the previous version. The authors should be commended for the additional work performed to address in an excellent manner the referee#1 criticisms, which I agree with in the first place. As a result, I strongly support the consideration of this paper for publication in this journal.

Below I report a few thoughts I had while reading the manuscript. I hope those can be of any help to further strengthen the presented work. I noted some typos and errors, especially in the supplementary figures. For example, the caption for Fig. S19 describes a panel (d) that is not displayed. The authors should carefully proofread the entire manuscript and supplement. I also have a few minor comments concerning the interpretation of the results, particularly the dike-fault interaction and subsequent seismicity. The modeling component (seismic and geodetic) appears to be sufficiently and carefully handled/presented given the data availability. The authors have the choice whether (or not) to include these suggestions in the manuscript, except for Comment number 1), which requires specific attention and must be addressed and mentioned within the manuscript.

Author response

Thank you for your interest in our paper, for taking the time to review it, and for the care you gave in your report. We agree that the early magnitude of completeness is not entirely satisfactory, but we feel this is an inherent issue with off-the-shelf machine-learning pickers at very high rates of seismicity, together with the lack of seismic stations on the island at the onset of the crisis. We have responded in detail below and made some necessary changes to the manuscript.

R5-01	Reviewer comment	Fig. 3c shows a clear difference of completeness magnitude throughout the seismic sequence and especially from the beginning of the seismic crises (~5 days) and later during the complex swarm activity. The b-value analysis (given the method used for the Gutenberg-Richter fit) results in a high $M_c \sim 2.5$. Now the problem is that the interpretation of early dynamics of seismicity can be biased from the fact that smaller events go undetected. Conversely small earthquakes are used to interpret later phases of the magma-fault dynamics and to discuss further intricacies in the development of the seismicity along the Pico do Carvao Fault zone. I do not think that the lack of smaller events would change the overall picture depicted by the authors regarding the interpretation of the dike-earthquake interaction. However, I ask the authors to add one or two sentences in the discussion section reporting the absence of smaller earthquakes during the early phase of dike propagation/emplacement, noting that this may have hindered the detection of finer scale processes discussed for later phases.
	Author response	Indeed, this is an excellent point about the early catalogue completeness, which is hampered further by the small number of stations of São Jorge at the start of the unrest. It is becoming better known that off-the-shelf AI pickers (e.g., PhaseNet, EQTransformer) do not deal so well with very high rates of overlapping low-magnitude seismicity (Beroza, personal communication; Tan et al., 2025, doi: 10.1126/science.adw9038), such as during the first few hours of the Sao Jorge crisis. Ultimately, template matching or retraining the picking model would be needed to circumvent this issue. Nevertheless, we fully agree with the reviewer that the main features of the seismicity still offer a clear view of the dynamics during the dike emplacement.
	Changes made	We have added the following text to the third paragraph of the Discussion: PM.PMAN, the station located closest to the ascending dike (Figs. 3d & 4). Our seismicity catalogue has a higher completeness magnitude during the first few days of the swarm (Fig. 3c), during dike emplacement. This issue likely arises from the initial ultra-high-rate of seismicity (Figs. 3b & 5a), and further compounded by the fact that there were only three operational seismic stations on São Jorge at the onset of the crisis, with the network densified a few days later (Fig. S12). Still, we consider that our seismicity catalogue gives us a clear overall picture of processes during and after emplacement of the dike. The deep (> 9 km) and shallow (< 5 km) portions of melt ascent were aseismic (Fig. 4). Only

R5-02	Reviewer comment	Aseismic seismicity zone and left-lateral residuals in the GNSS data: I would suggest considering the hypothesis that the dike emplacement could have triggered aseismic slip (creep or slow slip) along the PdC-FZ, a possibility not considered/discussed by the authors. Creep or slow-slip transients induced by dike emplacement have been documented, (see for example Cattania et al., 2014 JGR https://doi.org/10.1002/2016JB013722 , Xu et al., 2016 https://doi.org/10.1002/2015JB012505 , and many examples of aseismic graben forming at dike intrusion during rifting episode). I know the authors attempted to consider models including a fault, but could it be that aseismic left-lateral slip is the cause and/or can explain the left-lateral residuals in your model and reversal of focal mechanisms?
	Author response	We appreciate the reviewer's suggestion and the references provided. We already considered the possibility of aseismic slip along the PdC-FZ during dike emplacement and tested models including a fault shear component (see our Supplementary Text S1.3 and S1.4). Our tests allow us to rule out slow-slip associated with the dike opening, as such scenarios do not improve the fit to geodetic observations. Similarly, introducing shear within the dike does not substantially reduce residuals. While we cannot exclude the occurrence of minor, localised slip at the cm–mm scale, this remains unobserved with the available datasets. There is no geodetic evidence supporting a measurable aseismic slip event, and the reversal of focal mechanisms could be due to stress redistribution following the intrusion and/or (more speculatively) undetected fault creep. We believe this issue was already addressed in Paragraph 9 of the Discussion; however, we have made a few minor changes as detailed below.
	Changes made	We have clarified that such a shearing source would be “slow and aseismic”, and we have added a citation here to Cattania et al.
R5-03	Reviewer comment	The heterogeneities of the focal mechanisms (oblique e normal), and the across fault distribution of seismicity (from the video) seem to point out at the activation of a distributed fault/damage zone rather than a simple planar strike slip structure. In this case, the peculiar streak-like distribution of seismicity may come from a complexity of the fault system. Would it be this an additional possibility to add to the discussion to interpret the lineaments in Fig. 6? The complexity can be a complementary explanation of seismicity induced by magmatic/fluid fingers occurred after the dike emplacement in such a complex fault zone.
	Author response	We appreciate this point and completely agree.
	Changes made	We have noted the possibility of a distributed damage zone of the PdC-FZ in Paragraph 6 of the Discussion: (Fig. 7), or a combination of both lateral and vertical transport. Some filament-like seismicity clusters, such as Clusters 2, 7, and 18 (Fig. 6; Movie S1), show complex fracture-like geometries and an upward migrating front over weeks to months with slow diffusivity (~0.001-0.002 m²/s), suggesting long-lived upward fluid migration along propagating fractures⁷⁴ in a likely distributed the damage zone of the PdC-FZ.]

R5-04	Reviewer comment	The dike arrest below the volcanic edifice/island should be primarily due to the increasing compression due to the edifice load on the dike upper tip approaching the surface in the first place. To emplace, 20 km dike intrusion in only few days, magma should be primitive (mafic and low viscosity) with little gas content. The intrusion is so fast that exsolution of volatiles should not be occurring in this short time span, while volatiles would start to exsolve only later after the dike get arrested and at its final length. Due to the compression of the edifice the magmatic fluid can escape along the fault zone rather than upward explaining the peculiar spatial-temporal distribution of seismicity (streaks) and the absence of degassing at the surface. The authors seem to give equal weight to the mechanical arrest due to edifice load and increase in magma viscosity. I just would like to remark that the two processes may occur on slightly different time scales.
	Author response	We thank the reviewer for this detailed comment. Whilst we disagree that rapid ascent of a primitive magma does not preclude a gas-rich magma, as shown by the Tajogaite eruption of Cumbre Vieja on La Palma (Burton et al., 2023, doi: 10.1038/s43247-023-01103-x), we agree about the timescale issue and the weight given to the different magma stalling factors. We are in agreement that the compressive load of the island edifice was probably the main factor that caused the vertically ascending dike to stall.
	Changes made	We have made some modifications to the Discussion. - Paragraph 4: overcome the lithostatic and edifice-related stresses at shallow depths. A dynamic increase in magma viscosity, such as due to devolatilisation⁴, may have also inhibited magma ascent although exsolution of volatiles and increase in viscosity is unlikely to occur on such a short timescale given the rapid ascent of the dike and subsequent adjacent triggered seismicity. These factors, acting individually or in combination, likely led to the observed stalling of the dike at the base of the island edifice. - Paragraph 6: by fluid injection, which has greater seismic efficiency than most volcanic sequences⁶⁹ (Fig. 8). The rapid westward and downward migration of seismicity (Fig. 4), which left a ~2 km thick aseismic zone ('AS') between three seismic lineations ('UL', 'LL', and 'DL'; Fig. 4), may indicate that, following the main dike stalling ascending to the base due to the compressive load of the island edifice, a laterally propagating branch of magmatic fluids may have then initiated at the western edge of the vertically-stalled intrusion, at 9-10 km depth^{65,72,73}, and escaping into the permeable PdC-FZ.

R5-05	Reviewer comment	Page 8, the sentence: “whose front is consistent with a fluid diffusivity of ~0.001-0.002 m ² /s (Fig. 6b-c).” Here the authors report a very low value of diffusivity while in Fig.6c it is reported a much higher value.
	Author response	Many thanks for spotting the inconsistent values. The figure had the correct value (~0.014 m ² /s).
	Changes made	We have modified the text.
R5-06	Reviewer comment	Page 18 sentence: “The resulting static stress change led to the observed left-lateral seismogenic shear along the PdC-FZ, rotated by 90° with respect to the background right-lateral motion”. Is this 90 or 180 degree rotation.
	Author response	We confirm that this is a 90° rotation. Because earthquakes are described as double couples of forces, a rotation of 180° generates the same seismic radiation, giving rise to the well-known ambiguity of the two possible fault planes in focal mechanisms. Polarity flips correspond to 90° rotations of the stress field. In this case we observe a local rotation of the stress field by 90°, which causes backslip on the PdC-FZ, thus generating left-lateral earthquakes on a usually right-lateral fault. However, we agree that the sentence was a little unclear since it didn't explicitly refer to the stress rotation.
	Changes made	We have changed this sentence: helping the dike to stall ~1.6 km below the surface. The resulting static stress change led to the observed focal mechanisms that indicate left-lateral seismogenic shear along the PdC-FZ, indicating a rotated by 90° rotation of stress with respect to the long-term background right-lateral
R5-07	Reviewer comment	At Page 18 the sentence “We infer that because the dike intruded into a mature fault zone, few new fractures had to be created, leading to aseismic Mode-I opening, possibly accompanied by devolatilisation and shear dilatancy at the dike tip” reads odd. Dikes always open in Mode I, if the authors here refer to the dike-induced fractures opening in Mode I, then they need to clarify this concept.
	Author response	We agree with the reviewer's comment.
	Changes made	We have removed the redundant word, “Mode-I”.

Review Round 3

Reviewer 2 (Valerie Cayol)

Overall comments

I think my comments were addressed appropriately.

R2-1	Reviewer comment	Ref: original R2-7 (Round 2). Sheared intrusion or not ? I am not really sure of the direction of lateral slip that was taken into account. From the figure enclosed in the response, it seems to me that the addition of a lateral slip makes the solution worse, and corresponds to a right lateral displacement, when a left lateral movement is needed. From the posterior PPDs shown in the answer, it seems to me that the parameter range correspond to 0-10m and not -10-10m. Could a wrong slip direction explain that the solution is not improved and the PPD is uniform ? I may be wrong and misunderstanding the outputs from GBIS.
	Author response	We double-checked the inversion with the sheared intrusion, and we allowed a slip of -20 m to 20 m of horizontal slip on the wider parameter search, and then -10 m to 10 m in the narrower parameter search, starting with the uncertainty bounds from the previous inversions. So, we can confirm we didn't make an error and that the addition of the shear component makes the misfit larger.
	Changes made	N/A
R2-2	Reviewer comment	Ref: original R2-8 (Round 2). Leaky faults C: thank you for the clarification in the caption, the problem was indeed the perspective.
	Author response	Great!
	Changes made	N/A
R2-3	Reviewer comment	Ref: original R2-17 (Round 2). Perfect thank you ! The Akaike Information Criteria AIC are a good way to find the best compromise between fitting the data and having a large number of parameters. In your case you do not even need AIC, as more complexity degrades the best fit model.
	Author response	Understood!
	Changes made	N/A

R2-4	Reviewer comment	As indicated in your paper, the Moho is determined from seismic velocities at a depth of 11.5-13 km, while the first clusters (D1, D2, D3) are in fact at depths of 33-25 kilometers, in the mantle. This is also the storage depths indicated by microthermometry (25.5 km Zanon et al., 2023). This mantle source might be worth emphasizing as it is not that common.
	Author response	Excellent point - many thanks.
	Changes made	 - We slightly changed the abstract text to emphasise more clearly that the magma ascended from the mantle. - We have slightly modified the final sentence in Paragraph 2 of the Discussion to: “Such deep precursory seismicity^{18,54–56} is commonly interpreted as reflecting magma accumulation, destabilisation, or migration between deep reservoirs^{56–58}, with a depth of 25-35 km consistent with an upper mantle storage region inferred from fluid inclusion barometry of São Jorge lavas⁵³.” - In the concluding paragraph of the paper, we have added that the magma ascended from the upper mantle.

Steve Hicks et al. Nat. Comm.

I enjoyed reading the extensive paper of Stephen P. Hicks and Pablo J. Gonzalez et al. "Faults modulate magma propagation and triggered seismicity: the 2022 São Jorge (Azores) volcanic unrest". It is likely to further our understanding of magmatic and tectonic interaction, and it has an impressive multidisciplinary approach. The paper has already undergone insightful reviews by three reviewers, that have clearly helped improve the manuscript. I recommend it to be published in Nature Communications after some minor reviews.

Comments and questions:

Abstract and text: ultra is a bit extreme... ultra-high-precision hypocentre locations :

Fig.1 colour scheme - not clear contrast to the background, especially the brown PM circles, the purple 3K and the grey 4U. For the seismicity (orange circles) I recommend choosing a brighter colour.

Fig. 2: perhaps you could put N-arrow on a) as well?

Fig2 - caption: Dotted orange lines highlight the vents of **past** fissure eruptions – add past to sentence.

Fig 3: Note that the magnitudes for the first D1 cluster seems to be missing from Fig 3c.

Fig 4a:

purple dyke in rose diagram is nearly invisible (unless I zoom in a lot on the figure)– could you choose another/brighter colour? Also, add to figure caption for a) about the quadrangular dyke-opening solution.

cross-sections – could you label Moho?

Section 2.2 , page 6:

On uncertainty: What is the uncertainty of the manually picked events?

Deep precursory seismicity p.6: "bsl south of São Jorge (Figs. 3b-d & 4)..” delete extra .

Regarding the D1-3: how many stations where they detected on? Are they LF events or VT?

First 24 hours of the seismic swarm p6: change accelerating seismicity rat., to accelerating seismicity rate.

The median depth uncertainty of all events in the catalogue is 46 m. Add “relative depth”

~1702 UTC on 19 March 2022 - “ colon: is missing”

“the first locatable eq “ is misleading here – perhaps add “in this shallowing sequence / or add time stamp for clarification“

fix text “Almost concurrently, seismicity then migrates ~8 km westward and ~3 km downward at ~400-800 m/hour to depths of 8-13 km, where the main zone of seismicity develops and remains for subsequent months”

Fig 5d) is seem to be more like 5 km downwards – but perhaps I’m misunderstanding – pls clarify the text. And 5c) 10 km west rather than 8km?

p. 8

The movie is great! 😊

“These are separated by a ~2 km-wide aseismic region (“AS” in Fig. 5d).” : From Fig6a seems more like 1.5 km wide and 1.5 km long??

“Well-constrained focal mechanisms (see Methods; Fig. S16) show strike-slip faulting, with one nodal plane parallel to the main lineation of seismicity and the PdCFZ, indicating left-lateral faulting (Fig. 4a).” : In the discussions there is no mention of the focal mechanisms, if they’re similar or not to other dyke intrusions. I suggest adding a couple of sentences and references there of.

Fig.5. Pls add time stamp to the top of a) for clarity

Fig. S6 Caption say the blue dashed lines – they look purple to me on the figure.

Fig S13: Pls specify the bin width

Fig S16: for clarity remove the N-S E-W automatic fault planes by MTfit’s plotting routine.

Fig S18 caption: The grey shading shows the range of inverted models that have an RMS misfit within 5% of the best-fitting model – can you add the software used to the sentence? I’m unsure from the methods if this is 5% range of the 1D velocity model.

p11.

“we infer that PM.PMAN sensitivity kernel includes the geodetically imaged main dike, whereas PM.ROSA is too far west of the dike”: - perhaps add: main dike **onset/ascent** whereases

p.13.

“The deep (> 9 km) and shallow (< 5 km) portions of melt ascent were aseismic.”: Perhaps you should add **initial melt ascent** or something like that and reference Fig.4 bluish dots – otherwise it can be confusing.

“...these shallow earthquakes and tremor-like bursts likely mark the flanks of the ascending dike.” Pls add reference to Fig 5 c) for the tremor bursts.

p14.

“The dike intruded parallel to the PdC-FZ and VT seismicity zone, east of the surface-mapped fault zone, where the fault is likely buried by lavas and scoria cones”

I feel like there is something missing in this sentence. “VT seismicity zone” is mentioned here for the first time here – I assume it refers to the redish seismicity in the PdC-FZ, but that seismicity surely reflects a dyke intrusion at depth? So perhaps you can rephrase, to something like highlighting a narrow zone of seismicity.

This comment is of large significance: “the 2022 São Jorge seismic-volcanic unrest shows seismicity confined to one flank of the dike with no activity within or to the east of the dike.” Here the dike refers to the geodetically modelled dyke – please state that clearly in the text wherever that is the case – as there often a discrepancy between dikes constrained by geodetic models or seismicity.

p.17

“ultra-thin lineaments” : change to very? thin lineaments – ultra is quite extreme

“may indicate that, following the main dike ascending to the base of the island edifice, a laterally propagating branch of magmatic fluids^{71,79,80} may have initiated at the western edge of the vertically-stalled intrusion, at 9-10 km depth⁶” - this explanation is quite confusing – I’m unsure of what it means – do you perhaps mean that there is a magma pocket in this area left over from the initial ascent?

“suggesting long-lived upward fluid migration along propagating fractures⁷⁵ in the PdC-FZ” Are you suggesting propagation of fluids along faults or fault-propagation?

“The PdC-FZ likely facilitated fluid channelling⁸¹ through the lowermost crust near the crust-mantle boundary.” – can you add the value in brackets last?

p.19

“Our results suggest that pre-existing faults can have opposing effects on magma propagation.” Can you build under this sentence earlier in the discussion – it comes a bit out of the blue, as you mention the fault zone multiple times as facilitating the intrusion.

“Our results suggest that pre-existing faults can have opposing effects on magma propagation. The PdC-FZ facilitated rapid vertical melt ascent, but may also have caused magma to stall by allowing lateral devolatilisation and melt escape, thereby increasing viscosity and reducing pressure within the main intrusion.” : you need to add here something along the lines, that this is one scenario, another one could be that there wasn’t enough magma pressure etc. to drive the dyke to the surface.

Fig. 8 : Can you add the Moho depth as a number somewhere in the text? Then the discussions and the dimension of Fig8 will be clearer.

Methods:

Did you try using QM for the entire period? How reliable is EQ transformer in the first days with few stations?